# Bridging Offline Reinforcement Learning and Imitation Learning: A Tale of Pessimism

**Paria Rashidinejad**
Department of EECS
UC Berkeley
Berkeley, CA, 94709
paria.rashidinejad@berkeley.edu

**Banghua Zhu**
Department of EECS
UC Berkeley
Berkeley, CA, 94709
banghua@berkeley.edu

**Cong Ma**
Department of Statistics
University of Chicago
Chicago, IL, 60637
congm@uchicago.edu

**Jiantao Jiao**
Department of EECS
UC Berkeley
Berkeley, CA, 94709
jiantao@berkeley.edu

**Stuart Russell**
Department of EECS
UC Berkeley
Berkeley, CA, 94709
russell@berkeley.edu

## Abstract

Offline (or batch) reinforcement learning (RL) algorithms seek to learn an optimal policy from a fixed dataset without active data collection. Based on the composition of the offline dataset, two main methods are used: imitation learning which is suitable for expert datasets, and vanilla offline RL which often requires uniform coverage datasets. From a practical standpoint, datasets often deviate from these two extremes and the exact data composition is usually unknown. To bridge this gap, we present a new offline RL framework that smoothly interpolates between the two extremes of data composition, hence unifying imitation learning and vanilla offline RL. The new framework is centered around a weak version of the concentrability coefficient that measures the deviation of the behavior policy from the expert policy alone. Under this new framework, we ask: can one develop an algorithm that achieves a minimax optimal rate adaptive to unknown data composition? To address this question, we consider a lower confidence bound (LCB) algorithm developed based on pessimism in the face of uncertainty in offline RL. We study finite-sample properties of LCB as well as information-theoretic limits in multi-armed bandits, contextual bandits, and Markov decision processes (MDPs). Our analysis reveals surprising facts about optimality rates. In particular, in both contextual bandits and RL, LCB achieves a faster rate of $1/N$ for nearly-expert datasets compared to the usual rate of $1/\sqrt{N}$ in offline RL, where $N$ is the batch dataset sample size. In contextual bandits, we prove that LCB is adaptively optimal for the entire data composition range, achieving a smooth transition from imitation learning to offline RL. We further show that LCB is almost adaptively optimal in tabular MDPs.

## 1 Introduction

Reinforcement learning (RL) algorithms have recently achieved tremendous empirical success including beating Go champions [45, 46] and surpassing professionals in Atari games [30, 31], to name a few. Most success stories, however, are in the realm of online RL in which active data collection is necessary. This online paradigm falls short of leveraging previously-collected datasets and dealing with scenarios where online exploration is not possible [10]. To tackle these issues, offline

35th Conference on Neural Information Processing Systems (NeurIPS 2021).

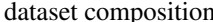

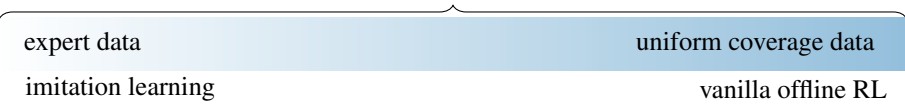

| dataset composition | |
|---|---|
| expert data | uniform coverage data |
| imitation learning | vanilla offline RL |

Figure 1: Dataset composition range for offline RL problems.

(or batch) reinforcement learning [24, 26] arises in which the agent aims at achieving competence by exploiting a batch dataset without access to online exploration. This paradigm is useful in a diverse array of application domains such as healthcare [51, 15, 35], autonomous driving [57, 4, 36], and recommendation systems [47, 12, 49].

The key component of offline RL is a pre-collected dataset from an unknown stochastic environment. Broadly speaking, there exist two types of *data composition* for which offline RL algorithms have shown promising empirical and theoretical success; see Figure 1 for an illustration.

**Expert data.** One end of the spectrum includes datasets collected by following an expert policy. For such datasets, imitation learning algorithms (e.g., behavior cloning [40]) are shown to be effective in achieving a small sub-optimality w.r.t. the expert policy. Recently, Rajaraman et al. [39] showed that the behavior cloning algorithm achieves the minimal sub-optimality of $1/N$ in episodic Markov decision processes (MDPs), where $N$ is the sample size in the expert dataset.

**Uniform coverage data.** On the other end of the spectrum lies the datasets with uniform coverage, which aim to cover *all* states and actions, even the states never visited or actions never taken by satisfactory policies. Most vanilla offline RL algorithms are only suited in this region and are shown to diverge—both empirically [11, 22] and theoretically [2, 7]—for narrower datasets [10, 20], such as those collected via human demonstrations or hand-crafted policies. In this regime, a widely-adopted requirement is the bounded *uniform concentrability coefficient* which assumes that the ratio of the state-action occupancy density of *any policy* and the data distribution is bounded uniformly over all states and actions [32, 9, 6, 52]. Another common assumption is uniformly lower bounded data distribution on all states and actions [43, 1], which ensures all states and actions are visited with sufficient probabilities. Algorithms developed for this regime are demonstrated to achieve a $1/\sqrt{N}$ sub-optimality competing with the optimal policy [53, 16, 50].

## 1.1 Motivating questions

Both of these two ends impose strong assumptions on the dataset: at one extreme, we hope for a solely expert-driven dataset; at the other extreme, we require the dataset to cover every, even sub-optimal, actions. In practice, there are numerous scenarios where the dataset deviates from these two extremes, which has motivated new offline RL benchmark datasets with different data compositions [10, 20]. With this need in mind, the first and foremost question is regarding offline RL formulations:

**Question 1 (Formulation)** *Can we propose an offline RL framework that accommodates the entire data composition range?*

We answer this question affirmatively by proposing a new formulation for offline RL that smoothly interpolates between two regimes: expert data and data with uniform coverage. More specifically, we characterize the data composition in terms of the ratio between the state-action occupancy density of an optimal policy $d^\star(s,a)$[1] and that of the data distribution $\mu(s,a)$, i.e., we define $C^\star$ to be the smallest constant that satisfies $d^\star(s,a)/\mu(s,a) \leq C^\star$ for all $s \in \mathcal{S}$ and $a \in \mathcal{A}$; see Definition 1 for a precise characterization.

In words, $C^\star$ can be viewed as a measure of the deviation between the data distribution and the distribution induced by the optimal policy. $C^\star = 1$ describes the expert datasets as by definition, the behavior policy is identical to the optimal policy. In contrast, when $C^\star > 1$, the dataset is no longer purely expert-driven: it could contain "spurious" samples—states and actions that are not visited by the optimal policy. As another example, when the data distribution is lower bounded by $\mu_{\min}$ over all states and actions, $C^\star$ is upper bounded by $\mu_{\min}^{-1}$.

---

[1]Our developments can accommodate arbitrary competing policies, however, we restrict ourselves to the optimal policy for ease of presentation.

Assuming a finite $C^\star$ is the weakest concentrability requirement [42, 13, 52] that is currently enjoyed only by some online algorithms such as CPI [18]. $C^\star$ imposes a much weaker assumption in contrast to other concentrability requirements which involve taking a maximum over all policies; see [42] for a hierarchy of different concentrability definitions. We would like to immediately point out that existing works on offline RL either do not specify the dependency of sub-optimality on data coverage [17, 55], or do not have a batch data coverage assumption that accommodates the entire data spectrum [54, 19]. For instance, Yin et al. [54] requires a uniformly lower bounded data distribution that traces an optimal policy, which implies that optimal actions should be included in states not visited by the optimal policy. Furthermore, this characterization of data coverage does not recover imitation learning: even if the behavior policy is exactly equal to the optimal policy, data distribution lower bound can be arbitrarily small. Further discussion of related work is presented in Appendix A.

With this formulation in mind, a natural next step is designing offline RL algorithms that can handle various data compositions, i.e., for all $C^\star \geq 1$. Recently, efforts have been made toward reducing the offline dataset requirements based on a shared intuition: the agent should act conservatively and avoid states and actions less covered in the offline dataset. Based on this intuition, a variety of model-based [55, 19, 56] and model-free [22, 33, 11, 34, 25, 37, 44, 14, 28, 23, 3] offline RL algorithms are proposed that achieve promising empirical results. However, it is observed empirically that existing model-free methods perform better when the dataset is nearly expert-driven whereas existing model-based methods perform better when the dataset is randomly-collected [55, 5, 56].

It remains unclear whether a single algorithm exists that performs well regardless of data composition—an important challenge from a practical perspective [21, 10, 20]. More importantly, the knowledge of the dataset composition may not be available *a priori* to assist in selecting the right algorithm. In practice, imitation learning often succeeds with very few samples in contrast to offline RL [41]. Unifying offline RL and imitation learning via a single algorithm is thus beneficial, as it can result in tremendous sample savings, in case the dataset has good coverage on an expert policy. This motivates the second question on the algorithm design:

**Question 2 (Adaptive algorithm design)** *Can we design algorithms that can achieve minimal sub-optimality when facing different dataset compositions (i.e., different $C^\star$)? Furthermore, can this be achieved in an adaptive manner, i.e., without knowing $C^\star$ beforehand?*

To answer the second question, we analyze a *pessimistic* variant of a value-based method in which we first form a lower confidence bound (LCB) for the value function of a policy using the batch data and then seek to find a policy that maximizes the LCB. The idea of pessimism has appeared in the literature of risk minimization [48, 8]. A similar algorithm design has recently appeared in [17]. It turns out that such a simple algorithm—fully agnostic to the data composition—achieves *almost* optimal performance in multi-armed bandits and MDPs, and optimally solves the offline learning problem in contextual bandits.

**Results summary.** Figure 2 summarizes our theoretical findings. For multi-armed bandits, we prove that LCB achieves a $\sqrt{C^\star/N}$ sub-optimality for any $C^\star \geq 1$. Yet, we prove lower bounds showing that LCB cannot be adaptively optimal for any $C^\star \geq 1$ if the knowledge of $C^\star$ is not available. For contextual bandits with at least two contexts, we prove that LCB enjoys a rate of $\sqrt{(C^\star - 1)/N} + 1/N$, which translates to a fast rate of $1/N$ when $C^\star \approx 1$ akin to the performance of behavioral cloning and smoothly transitions from $1/N$ to $1/\sqrt{N}$ as $C^\star$ increases. This rate matches the information theoretic limit, showing adaptive optimality of LCB in contextual bandits. Establishing the $C^\star - 1$ dependency requires a novel analysis based on a careful policy sub-optimality decomposition and directly analyzing the probability of taking wrong actions. For MDPs, we similarly show that LCB achieves a fast rate of $1/N$ for $C^\star \approx 1$ and a rate of $\sqrt{C^\star/N}$ for larger values of $C^\star$. We conjecture that LCB upper bound also has the form $\sqrt{(C^\star - 1)/N}$. We verify this conjecture in a simple example and show that establishing the $C^\star - 1$ dependency in MDPs requires a delicate analysis that accounts for the value gap between optimal and sub-optimal actions.

## 2 Background and problem formulation

**Notation.** The probability simplex over set $\mathcal{X}$ is denoted by $\Delta(\mathcal{X})$. We write $x \lesssim y$ when there exists $c > 0$ such that $x \leq cy$ and write $x \asymp y$ if $c_1, c_2 > 0$ exist such that $c_1|x| \leq |y| \leq c_2|x|$. We write $x \vee y$ to denote the supremum of $x$ and $y$. We write $f(x) = O(g(x))$ if $M > 0, x_0$ exist such

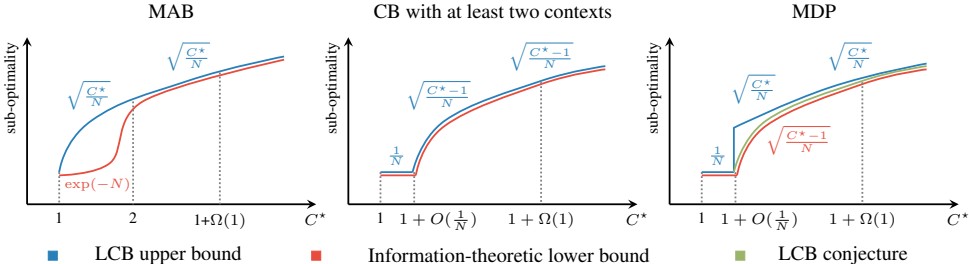

Figure 2: The sub-optimality upper bounds and information-theoretic lower bounds for the LCB-based algorithms. In all setting, $C^\star$ is unknown to the algorithm.

that $|f(x)| \le Mg(x)$ for all $x \ge x_0$. We use $\widetilde{O}(\cdot)$ to be the big-$O$ notation ignoring logarithmic factors. We write $f(x) = \Omega(g(x))$ if $M > 0, x_0$ exist such that $|f(x)| \ge Mg(x)$ for all $x \ge x_0$.

## 2.1 Markov decision processes

An infinite-horizon discounted MDP is described by a tuple $M = (\mathcal{S}, \mathcal{A}, P, R, \rho, \gamma)$, where $\mathcal{S}$ is a finite state space with $S = |\mathcal{S}|$, $\mathcal{A}$ is a finite action space, $P : \mathcal{S} \times \mathcal{A} \mapsto \Delta(\mathcal{S})$ is a transition matrix, $R : \mathcal{S} \times \mathcal{A} \mapsto \Delta([0, 1])$ encodes a family of reward distributions with $r : \mathcal{S} \times \mathcal{A} \mapsto [0, 1]$ as the expected reward function, $\rho : \mathcal{S} \mapsto \Delta(\mathcal{S})$ is the initial distribution, and $\gamma \in [0, 1)$ is a discount factor.

A stationary deterministic policy $\pi : \mathcal{S} \mapsto \mathcal{A}$ is a function that maps a state to an action. Correspondingly, the (normalized) state-action discounted occupancy measure $d^\pi : \mathcal{S} \times \mathcal{A} \mapsto [0, 1]$ is defined as $d^\pi(s, a) := (1-\gamma) \sum_{t=0}^{\infty} \gamma^t \, \mathbb{P}_t(s_t = s, a_t = a; \pi)$ for all $s \in \mathcal{S}, a \in \mathcal{A}$, where $\mathbb{P}_t(s_t = s, a_t = a; \pi)$ denotes the probability of $s_t = s, a_t = a$ after executing policy $\pi$ and starting from $s_0 \sim \rho(\cdot)$.

The value function $V^\pi : \mathcal{S} \mapsto \mathbb{R}$ of the policy $\pi$ is defined as $V^\pi(s) := \mathbb{E}[\sum_{t=0}^{\infty} \gamma^t r_t \mid s_0 = s, a_t = \pi(s_t) \text{ for all } t \ge 0]$ for $s \in \mathcal{S}$. The quality function (or Q-function) $Q^\pi : \mathcal{S} \times \mathcal{A} \to \mathbb{R}$ of policy $\pi$ is defined analogously $Q^\pi(s, a) := \mathbb{E}[\sum_{t=0}^{\infty} \gamma^t r_t \mid s_0 = s, a_0 = a, a_t = \pi(s_t) \text{ for all } t \ge 1]$. It is convenient to define a scalar summary of the performance of a policy $\pi$ as $J(\pi) := \mathbb{E}_{s \sim \rho}[V^\pi(s)]$. It is well known [38] that a stationary deterministic policy $\pi^\star$ exists that simultaneously maximizes $V^\pi(s)$ for all $s \in \mathcal{S}$, and hence maximizing the expected value $J(\pi)$. We use shorthands $V^\star := V^{\pi^\star}$ and $Q^\star := Q^{\pi^\star}$ to denote the optimal value function and Q-function.

## 2.2 Offline data and offline RL

The current paper focuses on offline RL, where the agent cannot interact with the MDP and instead is given a *batch dataset* $\mathcal{D}$ consisting of tuples $(s, a, r, s')$, where $r \sim R(\cdot \mid s, a)$ and $s' \sim P(\cdot \mid s, a)$. For simplicity, we assume $(s, a)$ pairs are generated i.i.d. according to a data distribution $\mu$ over $\mathcal{S} \times \mathcal{A}$, which is *unknown* to the agent.[2] We denote by $N(s, a) \ge 0$ the number of times $(s, a)$ is observed in $\mathcal{D}$ and by $N = |\mathcal{D}|$ the number of samples. The goal of offline RL is to find a policy $\hat{\pi}$ —based on $\mathcal{D}$— so as to minimize the expected sub-optimality with respect to the optimal policy $\pi^\star$, i.e., $\mathbb{E}_{\mathcal{D}}[J(\pi^\star) - J(\hat{\pi})]$, where the expectation is taken with respect to the randomness in the dataset.

## 2.3 Dataset coverage assumption

**Definition 1 (Single policy concentrability)** *Given a policy $\pi$, define $C^\pi$ to be the smallest constant that satisfies $d^\pi(s, a)/\mu(s, a) \le C^\pi$ for all $s \in \mathcal{S}$ and $a \in \mathcal{A}$.*

In words, $C^\pi$ characterizes the *distribution shift* between the occupancy measure induced by $\pi$ and data distribution $\mu$. For a stationary deterministic optimal policy, $C^\star := C^{\pi^\star}$ is the "best" *concentrability coefficient* definition [42, 13, 2, 52] which is often much smaller than the widely-used uniform concentrability coefficient $C := \max_\pi C^\pi$ which takes the maximum over all policies. A small $C^\pi$ implies that data distribution covers $(s, a)$ pairs visited by policy $\pi$, whereas a small $C$

---
[2]The i.i.d. assumption is motivated by the data randomization performed in experience replay [31].

requires the coverage of $(s, a)$ visited by all policies. A similar notion of concentratbility coefficient in $\ell_2$ norm instead of $\ell_\infty$ norm has appeared in the literature of off-policy evaluation [27, 29].

# 3 A warm-up: LCB in multi-armed bandits

We begin with the simplest example of an MDP, the multi-armed bandit (MAB) model, where $S = 1$ and $\gamma = 0$. For MABs, the offline dataset simplifies to $\mathcal{D} = \{(a_i, r_i)\}_{i=1}^N$, where $a_i \sim \mu(\cdot)$ and $r_i \sim R(a_i)$. Competing with an optimal arm $a^\star$, the data coverage assumption becomes $1/\mu(a^\star) \leq C^\star$. The goal of offline learning in MAB is to select an arm $\hat{a}$ that minimizes the expected sub-optimality $\mathbb{E}_{\mathcal{D}}[J(\pi^\star) - J(\hat{\pi})] = \mathbb{E}_{\mathcal{D}}[r(a^\star) - r(\hat{a})]$, where $r(a)$ is the expected reward of arm $a$.

## 3.1 Why does the best empirical arm fail?

A natural choice for solving the offline learning problem is to select the arm with the highest empirical average reward, i.e., $\hat{a} := \arg\max_a \hat{r}(a)$. Though intuitive, the empirical best arm is quite sensitive to the arms with a small $N(a)$: a less-explored sub-optimal arm might have a high empirical mean just by chance (due to large variance) and overwhelm the true optimal arm. The following proposition formalizes this intuition; see Appendix B.1 for a proof.

**Proposition 1 (Failure of the best empirical arm)** *For any $\epsilon < 0.3$, $N \geq 500$, there exists a bandit problem with two arms such that for $\hat{a} = \arg\max_a \hat{r}(a)$, one has $\mathbb{E}_{\mathcal{D}}[r(a^\star) - r(\hat{a})] \geq \epsilon$.*

## 3.2 LCB: The benefit of pessimism

Given the failure of best empirical arm, one soon realizes that it is not sensible to put every arm on an equal footing: one should be pessimistic about the true mean reward of the arms pulled less often. Strategically, pessimism can be deployed by first constructing a penalty function $b(a)$ that shrinks as $N(a)$ increases and then returning $\hat{a} \in \arg\max_a \ \hat{r}(a) - b(a)$. When $b(a)$ captures a confidence level about the empirical reward, $\hat{r}(a) - b(a)$ can be viewed as a lower confidence bound (LCB) on the true mean reward $r(a)$. Algorithm 1 shows one instance of the LCB, in which the penalty function originates from Hoeffding's inequality. The following theorem captures the performance of this algorithm in the MAB setting. The proof can be found in Appendix B.2.

**Theorem 1 (LCB sub-optimality, MAB)** *For a MAB, assume that $1/\mu(a^\star) \leq C^\star$ for some $C^\star \geq 1$. Provided that $N \geq 8C^\star \log N$, arm $\hat{a}$ returned by Algorithm 1 with $\delta = 1/N$ obeys*

$$\mathbb{E}_{\mathcal{D}}[r(a^\star) - r(\hat{a})] \lesssim \min\left(1, \sqrt{\frac{C^\star \log(2N|\mathcal{A}|)}{N}}\right). \tag{1}$$

Applying the above guarantee to the failure instance given in Proposition 1, one sees that LCB secures a sub-optimality of $\widetilde{O}(1/\sqrt{N})$, which beats the best empirical arm. This is because the LCB approach applies larger penalties to the arms with a small number of samples, which helps to rule them out.

## 3.3 Is LCB optimal for solving offline multi-armed bandits?

Given the performance bound (1), it is natural to ask whether LCB is optimal for solving offline MAB problems. To address this question, we resort to the usual minimax criterion. Define the following MAB family: $\mathsf{MAB}(C^\star) = \{(\mu, R) \mid 1/\mu(a^\star) \leq C^\star\}$, which includes all possible pairs of $\mu$ and $R$ such that the data coverage assumption $1/\mu(a^\star) \leq C^\star$ holds. We define the worst-case risk of any estimator $\hat{a}$ to be $\sup_{(\mu,R)\in\mathsf{MAB}(C^\star)} \mathbb{E}_{\mathcal{D}}[r(a^\star) - r(\hat{a})]$. An estimator $\hat{a}$ is a measureable function of the dataset $\mathcal{D}$ collected under the MAB instance $\mu$ and $R$. The following theorem shows that LCB is optimal up to a logarithmic factor when $C^\star \geq 2$; see Appendix B.3 for the proof.

**Theorem 2 (Information-theoretic limit, MAB)** *For $C^\star \geq 2$, one has*

$$\inf_{\hat{a}} \sup_{(\mu,R)\in\mathsf{MAB}(C^\star)} \mathbb{E}_{\mathcal{D}}[r(a^\star) - r(\hat{a})] \gtrsim \min\left(1, \sqrt{\frac{C^\star}{N}}\right). \tag{2}$$

*For $C^\star \in (1, 2)$, one has*

$$\inf_{\hat{a}} \sup_{(\mu, R) \in \mathsf{MAB}(C^\star)} \mathbb{E}_{\mathcal{D}}[r(a^\star) - r(\hat{a})] \gtrsim \exp\left(-(2 - C^\star) \log\left(\tfrac{2}{C^\star - 1}\right) \cdot N\right).$$

### 3.4 Imitation learning in bandit: the most played arm achieves a better rate

Theorem 2 reveals that when $C^\star \geq 2$, the best possible expected sub-optimality is $\sqrt{C^\star / N}$, which is achieved by LCB. On the other hand, in the case of $C^\star \in [1, 2)$, which corresponds to $\mu(a^\star) > 1/2$, we can simply use imitation learning to improve the rate by picking the most frequently selected arm in the dataset, i.e., $\hat{a} = \arg\max_a N(a)$. The performance guarantee of the most played arm is stated in the following proposition. The proof is deferred to Appendix B.4.

**Proposition 2 (Sub-optimality of the most played arm)** *Assume that $1/\mu(a^\star) \leq C^\star$ for some $C^\star \in [1, 2)$. For $\hat{a} = \arg\max_a N(a)$, we have*

$$\mathbb{E}_{\mathcal{D}}[r(a^\star) - r(\hat{a})] \leq \exp\left(-N \cdot \mathsf{KL}\left(\mathrm{Bern}\left(\tfrac{1}{2}\right) \,\middle\|\, \mathrm{Bern}\left(\tfrac{1}{C^\star}\right)\right)\right). \tag{3}$$

When $C^\star \in [1, 2)$, the most played arm achieves an exponential rate in $N$, whereas the upper bound for LCB is only $1/\sqrt{N}$. On the other hand, the most played arm algorithm completely fails when $C^\star > 2$, while LCB secures the rate $1/\sqrt{N}$. In terms of $C^\star$ dependence, the KL divergence above evaluates to $\log(C^\star / 2) + \log(1/(C^\star - 1))/2$. As $C^\star \to 1$, the rate increases to the order of $1/(C^\star - 1)^N$, matching the lower bound in Theorem 2.

### 3.5 Non-adaptivity of LCB

One may wonder whether LCB can achieve optimal rate under both cases of $C^\star \in [1, 2)$ and $C^\star \geq 2$. Unfortunately, we show in the following theorem that regardless of the parameter $\delta$ in Algorithm 1, LCB cannot be optimally adaptive in both regimes. The proof is deferred to Appendix B.5.

**Theorem 3 (Non-adaptivity of LCB, MAB)** *Let $C^\star = 1.5$. There exists a two-armed bandit instance $(\mu_0, R_0) \in \mathsf{MAB}(C^\star)$ such that Algorithm 1 with $L := \sqrt{\log(2|\mathcal{A}|/\delta)/2}$ satisfies*

$$\mathbb{E}_{\mathcal{D}}[r(a^\star) - r(\hat{a})] \gtrsim \min\left(\tfrac{\sqrt{L}}{N}, \tfrac{1}{\sqrt{N}}\right) \cdot \exp(-32L).$$

*On the other hand, when $C^\star = 6$, there exists $(\mu_1, R_1) \in \mathsf{MAB}(C^\star)$ such that*

$$\mathbb{E}_{\mathcal{D}}[r(a^\star) - r(\hat{a})] \gtrsim \min\left(1, \sqrt{\tfrac{L}{N}}\right).$$

Intuitively, a larger $L$ means that we put greater weight on penalty instead of empirical average. As $L \to \infty$, the LCB algorithm recovers the most played arm algorithm; while as $L \to 0$, the LCB algorithm recovers the best empirical arm algorithm. When $C^\star \in (1, 2)$, to achieve an exponential rate similar to the most played arm (Theorem 2), we need to select $\delta$ such that $L \gtrsim N^\alpha$ for $\alpha > 0$. However, under this choice of $L$, the algorithm fails to achieve $1/\sqrt{N}$ rate when $C^\star \geq 6$, which can be achieved by setting $\delta = 1/N$ (and thus $L = \log(2|\mathcal{A}|N)$) based on Theorem 1. Hence, it is impossible for LCB to achieve optimal rate in both $C^\star \in (1, 2)$ and $C^\star \geq 2$ regimes simultaneously.

## 4 LCB in contextual bandits

We take the analysis one step further by studying offline learning in contextual bandits (CBs). CB is a special case of MDP described in Section 2.1 with $\gamma = 0$. In CB setting, the batch dataset is $\mathcal{D} = \{(s_i, a_i, r_i)\}_{i=1}^N$ and the coverage assumption simplifies to $\max_s \rho(s)/\mu(s, \pi^\star(s)) \leq C^\star$. The offline learning objective in CB is to find a policy $\hat{\pi}$ based on $\mathcal{D}$ that minimizes the expected sub-optimality $\mathbb{E}_{\mathcal{D}}[J(\pi^\star) - J(\hat{\pi})] = \mathbb{E}_{\mathcal{D}, \rho}[r(s, \pi^\star(s)) - r(s, \hat{\pi}(s))]$.

### 4.1 LCB algorithm and its performance guarantee

The pessimism principle introduced for MAB can be naturally extended to CB by subtracting a penalty function $b(s, a)$ from the empirical rewards $\hat{r}(s, a)$ and returning $\hat{\pi}(s) \in \arg\max_a \hat{r}(s, a) - b(s, a)$ for every state $s$. The following theorem establishes an upper bound on the expected sub-optimality of the policy returned by Algorithm 1; see Appendix C.1 for a complete proof.

---

**Algorithm 1** LCB for bandits and contextual bandits

---

1: **Inputs:** Batch dataset $\mathcal{D} = \{(s_i, a_i, r_i)\}_{i=1}^N$, and confidence level $\delta$.
2: **for** $s \in \mathcal{S}, a \in \mathcal{A}$ **do**
3:     **if** $N(s, a) = 0$ **then** Set $\hat{r}(s, a) \hookleftarrow 0$.
4:     **else** Set $\hat{r}(s, a) \hookleftarrow \frac{1}{N(s,a)} \sum_{i=1}^N r_i \, \mathbb{1}\{(s_i, a_i) = (s, a)\}$.
5:     Compute the penalty $b(s, a) = \sqrt{\frac{2000 \log(2S|\mathcal{A}|/\delta)}{N(s,a) \vee 1}}$.
6: **Return:** $\hat{\pi}(s) \in \arg\max_a \hat{r}(s, a) - b(s, a)$ for each $s \in \mathcal{S}$.

---

**Theorem 4 (LCB sub-optimality, CB)** *For a CB with $S \geq 2$, assume $\max_s \rho(s)/\mu(s, \pi^\star(s)) \leq C^\star$, for some $C^\star \geq 1$. The policy $\hat{\pi}$ returned by Algorithm 1 with $\delta = 1/N$ obeys*

$$\mathbb{E}_{\mathcal{D}}[J(\pi^\star) - J(\hat{\pi})] \lesssim \min\left(1, \widetilde{O}\left(\sqrt{\frac{S(C^\star - 1)}{N}} + \frac{S}{N}\right)\right).$$

The sub-optimality in Theorem 4 consists of two terms. The first term has the usual statistical estimation rate of $1/\sqrt{N}$. The second term is due to *missing mass*, which captures the suboptimality incurred in states for which an optimal arm is never observed in the dataset. Importantly, the dependency of the first term on data composition is $C^\star - 1$. When $C^\star$ is close to one, LCB enjoys a faster rate of $1/N$, reminiscent of the behavioral cloning rate. Furthermore, the convergence rate smoothly transitions from $1/N$ to $1/\sqrt{N}$ as $C^\star$ increases.

### 4.2 Optimality of LCB for solving offline contextual bandits

We now establish an information-theoretic lower bound for the contextual bandit setup described above. Define the following family of CB problems $\mathsf{CB}(C^\star) := \{(\rho, \mu, R) \mid \max_s \rho(s)/\mu(s, \pi^\star(s)) \leq C^\star\}$. Let $\hat{\pi} : \mathcal{S} \mapsto \mathcal{A}$ be an arbitrary estimator of the best arm $\pi(s)$ for any state $s$, which is a measurable function of the data. For the worst-case risk of $\hat{\pi}$ defined as $\sup_{(\rho,\mu,R) \in \mathsf{CB}(C^\star)} \mathbb{E}_{\mathcal{D}}[J(\pi^\star) - J(\hat{\pi})]$, we have the following minimax lower bound:

**Theorem 5 (Information-theoretic limit, CB)** *Assume that $S \geq 2$. For any $C^\star \geq 1$, one has*

$$\inf_{\hat{\pi}} \sup_{(\rho,\mu,R) \in \mathsf{CB}(C^\star)} \mathbb{E}_{\mathcal{D}}[J(\pi^\star) - J(\hat{\pi})] \gtrsim \min\left(1, \sqrt{\frac{S(C^\star - 1)}{N}} + \frac{S}{N}\right).$$

The proof is provided in Appendix C.2. Comparing with Theorem 4, one sees that LCB enjoys a near-optimal rate in CB with $S \geq 2$ regardless of $C^\star$. This is in stark contrast to the MAB case.

On a closer inspection, in the $C^\star \in [1, 2)$ regime, there is a clear separation between the information-theoretic difficulty of offline learning in MAB, which has an exponential rate in $N$, and CB with at least 2 states, which has a $1/N$ rate. The reason behind this separation is the missing mass rate when $S \geq 2$. Informally, when there is only one state, the probability that an optimal action is never observed in the dataset decays exponentially. On the other hand, when there are more than one states, the probability that an optimal action is never observed in at least one state has a $1/N$ rate.

Assume hypothetically that we know $C^\star \in (1, 2)$. Under this circumstance, one might wonder whether simply picking the most played arm in every state achieves a fast rate, analogous to MAB. Strikingly, the answer is negative as the following proposition shows that the most played arm fails to achieve a vanishing rate when $C^\star \in (1, 2)$. The proof of this theorem is deferred to Appendix C.3.

**Proposition 3 (Failure of the most played arm, CB)** *For any $C^\star \in (1, 2)$, there exists a contextual bandit problem $(\rho, \mu, R) \in \mathsf{CB}(C^\star)$ such that for the policy $\hat{\pi}(s) = \arg\max_a N(s, a)$,*

$$\lim_{N \to \infty} \mathbb{E}_{\mathcal{D}}[J(\pi^\star) - J(\hat{\pi})] \geq C^\star - 1.$$

## 5 LCB in Markov decision processes

### 5.1 Offline value iteration with LCB

Now we are ready to instantiate the LCB principle to the full-fledged MDP case. Our algorithm design builds upon the classic value iteration algorithm. As we do not have access to the true expected

---

**Algorithm 2** Offline value iteration with LCB (VI-LCB)

---
1: **Inputs:** Batch dataset $\mathcal{D}$, discount factor $\gamma$, and confidence level $\delta$.
2: Set $T := \frac{\log N}{1-\gamma}$, $L := 2000 \log(2(T+1)S|\mathcal{A}|/\delta)$, $V_{\max} = (1-\gamma)^{-1}$.
3: Split $\mathcal{D}$ into $T+1$ sets $\mathcal{D}_t = \{(s_i, a_i, r_i, s_i')\}_{i=1}^m$ for $t \in \{0, 1, \ldots, T\}$.
4: Set $m_t(s,a) := \sum_{i=1}^m \mathbb{1}\{(s_i, a_i) = (s,a)\}$ based on dataset $\mathcal{D}_t$ for $t \in \{0, 1, \ldots, T\}$.
5: Initialize $Q_0(s,a) = 0$, $V_0(s) = 0$ and set $\pi_0(s) = \arg\max_a m_0(s,a)$, for $a \in \mathcal{A}$ and $s \in \mathcal{S}$.
6: **for** $t = 1, \ldots, T$ **do**
7:      **for** $(s,a) \in (\mathcal{S}, \mathcal{A})$ **do**
8:          **if** $m_t(s,a) = 0$ **then** Set $r_t(s,a) = 0$ and $P^t(\cdot \mid s,a)$ to be a random probability vector.
9:          **else** Set $P^t(\cdot \mid s,a)$ to be empirical transitions and $r_t(s,a)$ be empirical rewards.
10:          Compute penalty $b_t(s,a) := V_{\max} \cdot \sqrt{\frac{L}{m_t(s,a)\vee 1}}$.
11:          Set $Q_t(s,a) \leftarrow r_t(s,a) - b_t(s,a) + \gamma \sum_{s'} P^t(s' \mid s,a) V_{t-1}(s')$.
12:      Compute $V_t^{\mathrm{mid}} \leftarrow \max_a Q_t(s,a)$ and $\pi_t^{\mathrm{mid}}(s) \in \arg\max_a Q_t(s,a)$.
13:      **for** $s \in \mathcal{S}$ **do**
14:          **if** $V_t^{\mathrm{mid}}(s) \leq V_{t-1}(s)$ **then** $V_t(s) \leftarrow V_{t-1}(s)$ and $\pi_t(s) \leftarrow \pi_{t-1}(s)$.
15:          **else** $V_t(s) \leftarrow V_t^{\mathrm{mid}}(s)$ and $\pi_t(s) \leftarrow \pi_t^{\mathrm{mid}}(s)$.
16: **Return** $\hat{\pi} := \pi_T$.

---

rewards $r$ and transitions $P$, we replace them with the empirical counterparts $\hat{r}$ and $\hat{P}$. Furthermore, mimicking the LCB algorithmic design for MABs and CBs, we subtract a penalty function $b(s,a)$ from the $Q$ update as the finishing touch, which yields the value iteration algorithm with LCB:

$$Q(s,a) \leftarrow \hat{r}(s,a) - b(s,a) + \gamma \sum_{s'} \hat{P}(s' \mid s,a) V(s'), \qquad \text{for all } (s,a), \qquad (4)$$

$$V(s) \leftarrow \max_a Q(s,a), \qquad \text{for all } s. \qquad (5)$$

Algorithm 2 uses the update rule (4) as its key component as well as a few other tricks:

- **Data splitting.** Instead of using the full dataset $\mathcal{D}$ to form $\hat{r}$ and $\hat{P}$, Algorithm 2 splits $\mathcal{D}$ and uses different samples in each update (4). This procedure is not needed in practice, however, it alleviates the dependency issues in the analysis, which removes an extra factor of $S$ in the sample complexity.

- **Monotonic update.** Algorithm 2 updates value function $V$ and policy $\pi$ only when the corresponding value estimate is larger than that in the previous iteration. The key benefit of the monotonic update is to shave a $1/(1-\gamma)$ factor in the sample complexity; see [43] for further discussions.

Now we turn to the performance guarantee of VI-LCB, whose proof is given in Appendix D.6.

**Theorem 6 (LCB sub-optimality, MDP)** *For a MDP, assume that $\max_{s,a} d^\star(s,a)/\mu(s,a) \leq C^\star$. Then, for all $C^\star \geq 1$, policy $\hat{\pi}$ returned by Algorithm 2 with $\delta = 1/N$ achieves*

$$\mathbb{E}_{\mathcal{D}} \left[ J(\pi^\star) - J(\hat{\pi}) \right] \lesssim \min\left( \frac{1}{1-\gamma}, \widetilde{O}\left( \sqrt{\frac{SC^\star}{(1-\gamma)^5 N}} \right) \right). \qquad (6)$$

*In addition, if $1 \leq C^\star \leq 1 + \frac{L\log(N)}{200(1-\gamma)N}$, we have a tighter performance upper bound*

$$\mathbb{E}_{\mathcal{D}} \left[ J(\pi^\star) - J(\hat{\pi}) \right] \lesssim \min\left( \frac{1}{1-\gamma}, \widetilde{O}\left( \frac{S}{(1-\gamma)^4 N} \right) \right). \qquad (7)$$

The upper bound shows that for all $C^\star \geq 1$, we can guarantee a rate of $\widetilde{O}(\sqrt{SC^\star/((1-\gamma)^5 N)})$, which is similar to the rate of CB when the $C^\star = 1 + \Omega(1)$ by taking $\gamma = 0$. When $C = 1 + \widetilde{O}(1/N)$, we have a rate $S/((1-\gamma^4)N)$, which also recovers the result in the CB case. However, in the regime of $C^\star \in [1 + \widetilde{\Omega}(1/N), 1 + O(1)]$, while CB enjoys $\sqrt{S(C^\star - 1)/N}$ rate, we fail to give the same dependence on $C^\star$ in MDP; see Section 6 for further discussion on sub-optimality in this regime.

## 5.2 Information-theoretic lower bound for offline RL in MDPs

To capture the statistical limits of offline learning in MDPs, as before we define the following family of instances $\mathsf{MDP}(C^\star) := \{(\rho, \mu, P, R) \mid \max_{s,a} \frac{d^\star(s,a)}{\mu(s,a)} \leq C^\star\}$. We have the following minimax lower bound for offline policy learning in MDPs, with the proof deferred to Appendix D.7.

**Theorem 7 (Information-theoretic limit, MDP)** *For any $C^\star \geq 1, \gamma \geq 0.5$, one has*

$$\inf_{\hat{\pi}} \sup_{(\rho,\mu,P,R)\in\mathsf{MDP}(C^\star)} \mathbb{E}_\mathcal{D}[J(\pi^\star) - J(\hat{\pi})] \gtrsim \min\left(\frac{1}{1-\gamma}, \frac{S}{(1-\gamma)^2 N} + \sqrt{\frac{S(C^\star-1)}{(1-\gamma)^3 N}}\right).$$

**Imitation learning and offline learning.** Similar to the CB lower bound, the statistical limit in Theorem 7 involves two terms. The first term captures the imitation learning regime under which a fast rate $1/N$ is expected, while the second term deals with the large $C^\star$ regime with a rate $1/\sqrt{N}$. More interestingly, the dependence on $C^\star$ appears to be $C^\star - 1$, which is different from the performance upper bound of VI-LCB in Theorem 6. We will comment more on this in the coming section.

**Dependence on the effective horizon.** Comparing the upper bound in Theorem 6 with the lower bound in Theorem 7, one sees that the sample complexity of VI-LCB is loose by an extra $1/(1 - \gamma)^2$ factor in sample complexity. We believe that this extra factor can be shaved by replacing the Hoeffding-based penalty to a Bernstein-based one and using variance reduction similar to [43].

# 6 Proof techniques and conjecture

To prove the crude rate of $\sqrt{C^\star/N}$ for any $C^\star \geq 1$ in all three MAB, CB, and MDP settings, it is sufficient to bound the sub-optimality by an expectation over penalty followed by the coverage assumption. Since the penalty is proportional to $1/\sqrt{N}$, this technique only yields a $1/\sqrt{N}$ rate.

**The $\sqrt{(C^\star - 1)/N} + S/N$ rate in CB.** We carefully decompose the sub-optimality and characterize the probability of choosing sub-optimal arms to prove the tighter bound in CB, which closes the gap between upper and lower bounds. We achieve this goal by directly analyzing the policy sub-optimality via a gradual decomposition of the sub-optimality of $\hat{\pi}$ as illustrated in Figure 3.

$$\mathbb{E}_\mathcal{D}[J(\pi^\star) - J(\hat{\pi})] \begin{cases} N(s, \pi^\star(s)) = 0 \; \to T_1 \\ \\ N(s, \pi^\star(s)) \geq 1 \begin{cases} \mathbb{1}\{\mathcal{E}^c\} \to T_2 \\ \\ \mathbb{1}\{\mathcal{E}\} \begin{cases} \rho(s) < \frac{2C^\star L}{N} \to T_3 \\ \\ \rho(s) \geq \frac{2C^\star L}{N} \begin{cases} \mu(s, \pi^\star(s)) < 10\overline{\mu}(s) \; \to T_4 \\ \mu(s, \pi^\star(s)) \geq 10\overline{\mu}(s) \; \to T_5 \end{cases} \end{cases} \end{cases} \end{cases}$$

Figure 3: Decomposition of the sub-optimality of the policy $\hat{\pi}$ returned by Algorithm 1.

In the first level of decomposition, we separate the error based on whether $N(s, \pi^\star(s))$ is zero for a certain state $s$. When $N(s, \pi^\star(s)) = 0$, there is absolutely no basis for the LCB approach to figure out the correct action $\pi^\star(s)$. Fortunately, this type of error, incurred by *missing mass*, can be bounded by $T_1 \lesssim \frac{C^\star S}{N}$.

The second level of decomposition hinges on the following clean/good event: $\mathcal{E} := \{\forall s, a : |r(s,a) - \hat{r}(s,a)| \leq b(s,a)\}$. In words, the event $\mathcal{E}$ captures the scenario in which the penalty function provides valid confidence bounds for every state-action pair. Standard concentration arguments tell us that $\mathcal{E}$ takes place with high probability, i.e., the term $T_2$ in the figure is no larger than $\delta$. By setting $\delta$ small, say $1/N$, we are allowed to concentrate on the case when $\mathcal{E}$ holds.

The third level of decomposition relies on the observation that states with small weights (i.e., $\rho(s)$ is small) have negligible effects on the sub-optimality $J(\pi^\star) - J(\hat{\pi})$. More specifically, the aggregated contribution $T_3$ from the states with $\rho(s) \lesssim \frac{C^\star L}{N}$ is upper bounded by $T_3 \lesssim \frac{C^\star S L}{N}$. This allows us to focus on the states with large weights. We record an immediate consequence of large $\rho(s)$ and the data coverage assumption, that is $\mu(s, \pi^\star(s)) \geq \rho(s)/C^\star \asymp L/N$.

Now comes the most important part of the error decomposition, which is not present in the MAB analysis. We decompose the error based on whether the optimal action has a higher data probability $\mu(s, \pi^\star(s))$ than the total probability of sub-optimal actions $\overline{\mu}(s) := \sum_{a \neq \pi^\star(s)} \mu(s, a)$. In particular,

when $\mu(s, \pi^\star(s)) < 10\overline{\mu}(s)$, we can repeat the analysis of MAB and show that $T_4 \lesssim \sqrt{\frac{S(C^\star - 1)L}{N}}$. Here, the appearance of $C^\star - 1$, as opposed to $C^\star$ is due to the restriction $\mu(s, \pi^\star(s)) < 10\overline{\mu}(s)$. One can verify that $\mu(s, \pi^\star(s)) < 10\overline{\mu}(s)$ together with the data coverage assumption ensures that $\sum_{s:\rho(s) \geq 2C^\star L/N, \mu(s,\pi^\star(s)) < 10\overline{\mu}(s)} \rho(s) \lesssim C^\star - 1$. On the other hand, when $\mu(s, \pi^\star(s)) \geq 10\overline{\mu}(s)$, i.e., when the optimal action is more likely to be seen in the dataset, the penalty function $b(s, \pi^\star(s))$ associated with the optimal action would be much smaller than those of the sub-optimal actions. Thanks to the LCB approach, the optimal action will be chosen with high probability, i.e., $T_5 \lesssim 1/N^{10}$.

$C^\star$ **dependency in MDPs.** Ignoring the dependency on $1/(1-\gamma)$, by comparing Theorems 6 and 7, one realizes that VI-LCB is optimal both when $C^\star \geq 1 + \Theta(1)$ and $C^\star \leq 1 + \Theta(1/N)$. However, in the middle region, the upper and lower bounds differ in their dependency on $C^\star$. We *conjecture* that VI-LCB is optimal even this regime and the current gap is an artifact of our analysis.

**Conjecture 1 (Adaptive optimality of LCB)** *The LCB approach, together with value iteration is adaptively optimal for solving offline MDPs for all ranges of $C^\star$.*

**Technical hurdle.** It turns out that closing the gap in MDPs is significantly more challenging due to error propagation. Naively applying the decomposition in the CB case fails to achieve the $C^\star - 1$ dependence in the regime $C^\star \in [1 + \Omega(1/N), 1 + O(1)]$. Major difficulties arise in controlling case (i), where $\mu(s, \pi^\star(s)) \gg \sum_{a \neq \pi^\star(s)} \mu(s, a)$. Recall that VI-LCB picks the right action if

$$r_t(s, \pi^\star(s)) - \sqrt{\frac{L}{m_t(s, \pi^\star(s)) \vee 1}} + \gamma P^t(\cdot|s, \pi^\star(s)) \cdot V_{t-1} > r_t(s, a) - \sqrt{\frac{L}{m_t(s, a) \vee 1}} + \gamma P^t(\cdot|s, a) \cdot V_{t-1},$$

for all $a \neq \pi^\star(s)$. The presence of the previous values $V_{t-1}$ drastically changes the picture: even if we know that $m_t(s, \pi^\star(s)) \gg m_t(s, a)$, the current analysis does not guarantee the above inequality. It is likely that the value gap $g(s) := Q^\star(s, \pi^\star(s)) - Q^\star(s, a)$ affects whether VI-LCB chooses the optimal action. How to study the interplay between the gap and the policy chosen by VI-LCB forms the main obstacle to obtaining tight performance guarantees when $C^\star \in [1 + \Omega(1/N), 1 + O(1)]$.

**A confirmation from an episodic MDP.** In Appendix E.6, we present an episodic example to demonstrate that (1) an episodic variant of VI-LCB achieves the optimal dependency on $C^\star$ and hence closes the gap between upper and lower bounds, and (2) a tight analysis of the sub-optimality is rather intricate and depends on a delicate decomposition based on the gap $Q^\star(s, \pi^\star(s)) - Q^\star(s, a)$. As a preview, our example is an episodic MDP with $H = 3$. To tackle case (i) above, we decompose the error based on whether $g(s)$ is small. If $g(s)$ is small for state $s$, the contribution to the sub-optimality is well controlled. Otherwise, we manage to show that VI-LCB selects the right action with high probability. What is more interesting and surprising is that the right threshold for value gap is given by $\sqrt{(C^\star - 1)/N}$. Ultimately, this allows us to achieve the optimal dependency on $C^\star$.

## 7  Discussion

We propose a new batch RL framework based on the single policy concentrability coefficient $C^\star$ that smoothly interpolates the two extremes of data composition encountered in practice, namely the expert data and uniform coverage data. Under this new framework, we pursue the statistically optimal algorithms that can even be implemented without the knowledge of the data composition. More specifically, focusing on the lower confidence bound (LCB) approach inspired by the principle of pessimism, we find that LCB is adaptively minimax optimal for addressing the offline learning problems in most settings. Under the new framework, there exist numerous avenues for future study. One interesting direction is to provide a tighter bound for LCB in MDP for the regime where a significant fraction of the data comes from the optimal policy (Conjecture 1). Furthermore, it would be important to extend this work to function approximation setting. We expect to see our characterization of offline RL via single-policy concentrability to be extended to the function approximation setting and used in the development of new offline RL algorithms that only require partial coverage. Another interesting direction for future work is to analyze whether alternative conservative methods such as value regularization can achieve adaptivity and/or minimax optimality.

## Acknowledgements

The authors are grateful to Nan Jiang, Aviral Kumar, Yao Liu, and Zhaoran Wang for helpful discussions and suggestions. PR was partially supported by the Open Philanthropy Foundation and the Leverhulme Trust. BZ and JJ were partially supported by NSF Grants IIS-1901252, CCF-1909499, and DMS-2023505.

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
