# Supplementary Material for "Bridging Offline Reinforcement Learning and Imitation Learning: A Tale of Pessimism"

## Contents

# A  Related work

In this section we review additional related works. In Section A.1, we discuss various assumptions on the batch dataset that have been proposed in the literature. In Section A.2, we review conservative methods in offline RL. We conclude this section by comparing existing lower bounds with the ones presented in this paper.

## A.1  Assumptions on batch dataset

One of the main challenges in offline RL is the insufficient coverage of the dataset caused by lack of online exploration (Wang et al., 2020a; Zanette, 2020; Szepesvári, 2010) and in particular the *distribution shift* in which the occupancy density of the behavior policy and the one induced by the learned policy are different. This effect can be characterized using concentrability coefficients (Munos, 2007) which impose bounds on the density ratio (importance weights).

Most concentrability requirements imposed in existing offline RL involve taking a supremum of the density ratio over all state-action pairs and all policies, i.e., $\max_\pi C^\pi$ (Scherrer, 2014; Chen and Jiang, 2019; Jiang, 2019; Wang et al., 2019; Liao et al., 2020; Liu et al., 2019; Zhang et al., 2020a) and some definitions are more complex and stronger assuming a bounded ratio per time step (Szepesvári and Munos, 2005; Munos, 2007; Antos et al., 2008; Farahmand et al., 2010; Antos et al., 2007). A more stringent definition originally proposed by Munos (2003) also imposes exploratoriness on state marginals. This definition is recently used by Xie and Jiang (2020) to develop an efficient offline RL algorithm with general function approximation and only realizability. The MABO algorithm proposed by Xie and Jiang (2020) and the related algorithms by Feng et al. (2019) and Uehara et al. (2020) use a milder definition based on a *weighted* norm of density ratios as opposed to the infinity norm. In contrast, to compete with an optimal policy, we only require coverage over states and actions visited by that policy, which is referred to as the "best" concentrability coefficient (Scherrer, 2014; Geist et al., 2017; Agarwal et al., 2020b; Xie and Jiang, 2020).

Another related assumption is the uniformly lower bounded data distribution. For example, some works consider access to a generative model with an equal number of samples on all state-action pairs (Sidford et al., 2018a,b; Agarwal et al., 2020a; Li et al., 2020). As discussed before, this assumption is significantly stronger than assuming $C^\star$ is bounded. Furthermore, one can modify the analysis of the LCB algorithm to show optimal data composition dependency in this case as well.

Relaxation of the concentrability assumption is possible by allowing the ratio to hold only for a subset $\mathcal{C}$ of state-action pairs and characterizing the sub-optimality incurred by $(s, a) \in \mathcal{C}$ via a missing mass analysis dependent on a constant $\xi$ such that $\sum_{(s,a) \notin \mathcal{C}} d^\star(s, a) \leq \xi$.

## A.2  Conservatism in offline RL

In practice, such high coverage assumptions on batch dataset also known as data diversity (Levine et al., 2020) often fail to hold (Gulcehre et al., 2020; Agarwal et al., 2020c; Fu et al., 2020). Several methods have recently emerged to address such strong data requirements. The first category involves policy regularizers or constraints to ensure closeness between the learned policy and the behavior policy (Fujimoto et al., 2019b; Wu et al., 2019; Jaques et al., 2019; Peng et al., 2019; Siegel et al., 2020; Wang et al., 2020b; Kumar et al., 2019; Fujimoto et al., 2019a; Ghasemipour et al., 2020; Nachum et al., 2019b; Zhang et al., 2020b; Nachum et al., 2019a; Zhang et al., 2020c). These methods are most suited when the batch dataset is nearly-expert (Wu et al., 2019; Fu et al., 2020) and sometimes require the knowledge of the behavior policy.

Another category includes the value-based methods. Kumar et al. (2020) propose conservative Q-learning through value regularization and demonstrate empirical success. Liu et al. (2020) propose a variant of fitted Q-iteration with a conservative update called MSB-QI. This algorithm effectively requires the data distribution to be uniformly lower bounded on the state-action pairs visited by any competing policy. Moreover, the sub-optimality of MSB-QI has a $1/(1-\gamma)^4$ horizon dependency compared to ours which is $1/(1-\gamma)^{2.5}$. The last category involves learning pessimistic models such as Kidambi et al. (2020), Yu et al. (2020) and Yu et al. (2021) all of which demonstrate empirical success.

From a theoretical perspective, the recent work Jin et al. (2020) studies pessimism in offline RL in episodic MDPs and function approximation setting. The authors present upper and lower bounds for linear MDPs with a suboptimality gap of $dH$, where $d$ is the feature dimension and $H$ is the horizon. Specialized to the tabular case, this gap is equal to $SAH$, compared to ours which is only $H$. Furthermore, this work does not study the adaptivity of pessimism to data composition.

[PR: Need to change the paragraph below. + read csaba paper] Another recent work by Yin et al. (2021) studies pessimism in tabular MDP setting and proves matching upper and lower bounds. However, their approach requires a uniform lower bound on the data distribution that traces an optimal policy. This assumption is stronger than ours; for example, it requires optimal actions to be included in the states not visited by an optimal policy. Furthermore, this characterization of data coverage does not recover the imitation learning setting: if the behavior policy is exactly equal to the optimal policy, data distribution lower bound can still be small.

## A.3 Information-theoretic lower bounds

There exists a large body of literature providing information-theoretic lower bounds for RL under different settings; see e.g., Dann and Brunskill (2015); Krishnamurthy et al. (2016); Jiang et al. (2017); Jin et al. (2018); Azar et al. (2013); Lattimore and Hutter (2012); Domingues et al. (2020); Duan et al. (2020); Zanette (2020); Wang et al. (2020a). In the generative model setting with uniform samples, Azar et al. (2013) proves a lower bound on value sub-optimality which is later extended to policy sub-optimality by Sidford et al. (2018a). For the offline RL setting, Kidambi et al. (2020) prove a lower bound only considering the data and policy occupancy support mismatch without dependency on sample size. Jin et al. (2020) gives a lower bound for linear MDP setting but which does not give a tight dependency on parameters when specialized to the tabular setting. In Yin et al. (2020, 2021), a hard MDP is constructed with a dependency on the data distribution lower bound. In contrast, our lower bounds depend on $C^\star$, which has not been studied in the past, and holds for the entire data spectrum. In the imitation learning setting, (Xu et al., 2020) considers discounted MDP setting and shows a lower bound on the performance of the behavior cloning algorithm. We instead present an information-theoretic lower bound for any algorithm for $C^\star = 1$ which is based on adapting the construction of Rajaraman et al. (2020) to the discounted case.

# B  Proofs for multi-armed bandits

In Section B.1, we prove Proposition 1 that demonstrates the failure of the best empirical arm when solving offline MABs. Section B.2 is devoted to the proof of Theorem 1, which supplies the performance upper bound of the LCB approach. This upper bound is accompanied by a minimax lower bound given in Section B.3. In the end, we provably show the lack of adaptivity of the LCB approach in Section B.4.

## B.1 Proof of Proposition 1

We start by introducing the bandit instance under consideration. Set $|\mathcal{A}| = 2$, $a^\star = 1$, $\mu(1) = (N-1)/N$, and $\mu(2) = 1/N$. As for the reward distributions, for the optimal arm $a^\star = 1$, we let $R(1) = 2\epsilon$ almost surely. In contrast, for arm 2 we set

$$R(2) = \begin{cases} 2.1\epsilon, & \text{w.p. } 0.5, \\ 0, & \text{w.p. } 0.5. \end{cases}$$

It is easy to check that indeed $a^\star = 1$ is the optimal arm to choose. Our goal is to show that for this particular bandit problem, given $N$ offline data from $\mu$ and $R$, the empirical best arm $\hat{a}$ will perform poorly with high probability.

To see this, consider the following event

$$\mathcal{E}_1 := \{N(2) = 1\}.$$

We have

$$\mathbb{P}(\mathcal{E}_1) = N \cdot \mu(1)^{N-1} \cdot \mu(2) = (1 - 1/N)^{N-1}.$$

As long as $N$ is sufficiently large (say $N \geq 500$), we have $\mathbb{P}(\mathcal{E}_1) \geq 0.36$ for any $0 \leq n \leq N$, and thus $\mathbb{P}(\mathcal{E}_1) \geq 0.36$.

Now we are in position to develop a performance lower bound for the empirical best arm $\hat{a}$. By construction, we have $r(1) - r(2) = 0.95\epsilon$. Therefore the sub-optimality is given by

$$\begin{aligned} \mathbb{E}_{\mathcal{D}}[r(a^\star) - r(\hat{a})] &= 0.95\epsilon \cdot \mathbb{P}(\hat{a} \neq a^\star) \\ &\geq 0.95\epsilon \cdot \mathbb{P}(\mathcal{E}_1 \cap \hat{r}(2) = 2.1\epsilon) \\ &\geq 0.95\epsilon \cdot 0.18 > 0.1\epsilon. \end{aligned}$$

Rescaling the value of $\epsilon$ finishes the proof.

## B.2 Proof of Theorem 1

Before embarking on the main proof, we record two useful lemmas. The first lemma sandwiches the true mean reward by the empirical one and the penalty function, which directly follows from Hoeffding's inequality and a union bound. For completeness, we provide the proof at the end of this subsection.

**Lemma 1.** *With probability at least $1 - \delta$, we have*

$$\hat{r}(a) - b(a) \leq r(a) \leq \hat{r}(a) + b(a), \quad \text{for all } 1 \leq a \leq |\mathcal{A}|. \tag{1}$$

The second one is a simple consequence of the Chernoff bound for binomial random variables.

**Lemma 2.** *With probability at least $1 - \exp(-N\mu(a^\star)/8)$, one has*

$$N(a^\star) \geq \frac{1}{2}N\mu(a^\star). \tag{2}$$

Denote by $\mathcal{E}$ the event that both relations (1) and (2) hold. Conditioned on $\mathcal{E}$, one has

$$r(a^\star) \leq \hat{r}(a^\star) + b(a^\star) = \hat{r}(a^\star) - b(a^\star) + 2b(a^\star).$$

In view of the definition of $\hat{a}$, we have $\hat{r}(a^\star) - b(a^\star) \leq \hat{r}(\hat{a}) - b(\hat{a})$, and hence

$$r(a^\star) \leq \hat{r}(\hat{a}) - b(\hat{a}) + 2b(a^\star) \leq r(\hat{a}) + 2b(a^\star),$$

where the last inequality holds under the event $\mathcal{E}$ (in particular the bound (1) on $\hat{a}$). Now we are left with the term $b(a^\star)$. It suffices to lower bound $N(a^\star)$. Note that the event $\mathcal{E}$ (cf. the lower bound (2)) ensures that

$$N(a^\star) \geq \frac{1}{2} N\mu(a^\star) \geq \frac{N}{2C^\star} > 0.$$

As a result, we conclude

$$b(a^\star) = \sqrt{\frac{\log(2|\mathcal{A}|/\delta)}{2N(a^\star)}} \leq \sqrt{\frac{\log(2|\mathcal{A}|/\delta)}{N\mu(a^\star)}},$$

which further implies

$$r(a^\star) \leq r(\hat{a}) + 2\sqrt{\frac{\log(2|\mathcal{A}|/\delta)}{N\mu(a^\star)}} \tag{3}$$

whenever the event $\mathcal{E}$ holds. It is easy to check that under the assumption $N \geq 8C^\star \log(1/\delta)$, we have $\mathbb{P}(\mathcal{E}) \geq 1 - 2\delta$. This finishes the proof of the high probability claim.

In the end, we can compute the expected sub-optimality as

$$\mathbb{E}_{\mathcal{D}}[r(a^\star) - r(\hat{a})] = \mathbb{E}_{\mathcal{D}}[(r(a^\star) - r(\hat{a}))\,\mathbb{1}\{\mathcal{E}\}] + \mathbb{E}_{\mathcal{D}}[(r(a^\star) - r(\hat{a}))\,\mathbb{1}\{\mathcal{E}^c\}]$$

$$\leq 2\sqrt{\frac{\log(2|\mathcal{A}|/\delta)}{N\mu(a^\star)}}\mathbb{P}(\mathcal{E}) + \mathbb{P}(\mathcal{E}^c).$$

Here the inequality uses the bound (3) and the fact that $r(a^\star) - r(\hat{a}) \leq 1$. We continue bounding the sub-optimality by

$$\mathbb{E}_{\mathcal{D}}[r(a^\star) - r(\hat{a})] \leq 2\sqrt{\frac{\log(2|\mathcal{A}|/\delta)}{N\mu(a^\star)}} + 2\delta \leq 2\sqrt{\frac{C^\star \log(2|\mathcal{A}|/\delta)}{N}} + 2\delta.$$

Here the last relation uses $\mu(a^\star) \geq 1/C^\star$. Taking $\delta = 1/N$ completes the proof.

*Proof of Lemma 1.* Consider a fixed action $a$. If $N(a) = 0$, one trivially has $\hat{r}(a) - b(a) = -1 \leq r(a) \leq \hat{r}(a) + b(a) = 1$. When $N(a) > 0$, applying Hoeffding's inequality, one sees that

$$\mathbb{P}\left(|\hat{r}(a) - r(a)| \geq \sqrt{\frac{\log(2|\mathcal{A}|/\delta)}{2N(a)}} \mid N(a)\right) \leq \frac{\delta}{|\mathcal{A}|}.$$

Since this claim holds for all possible $N(a)$, we have for any fixed action $a$

$$\mathbb{P}\left(|\hat{r}(a) - r(a)| \geq b(a)\right) \leq \frac{\delta}{|\mathcal{A}|}.$$

A further union bound over the action space yields the advertised claim. □

## B.3 Proof of Theorem 2

We separate the proof into two cases: $C^\star \geq 2$ and $C^\star \in (1, 2)$. For both cases, our lower bound proof relies on the classic Le Cam's two-point method (Yu, 1997; Le Cam, 2012). In essence, we construct two MAB instances in the family $\mathsf{MAB}(C^\star)$ with different optimal rewards that are difficult to distinguish given the offline dataset.

**The case of $C^\star \geq 2$.** We consider a simple two-armed bandit. For the behavior policy, we set $\mu(2) = 1/C^\star$ and $\mu(1) = 1 - 1/C^\star$. Since we are constructing lower bound instances, it suffices to consider Bernoulli distributions supported on $\{0, 1\}$. In particular, we consider the following two possible sets for the Bernoulli means

$$f_1 = (\frac{1}{2}, \frac{1}{2} - \delta); \qquad f_2 = (\frac{1}{2}, \frac{1}{2} + \delta),$$

with $\delta \in [0, 1/4]$. Indeed, $(\mu, f_1), (\mu, f_2) \in \mathsf{MAB}(C^\star)$ with the proviso that $C^\star \geq 2$. Denote the loss/sub-optimality of an estimator $\hat{a}$ to be

$$\mathcal{L}(\hat{a}; f) := r(a^\star) - r(\hat{a}), \tag{4}$$

where the optimal action $a^\star$ implicitly depends on the reward distribution $f$. Clearly, for any estimator $\hat{a}$, we have

$$\mathcal{L}(\hat{a}; f_1) + \mathcal{L}(\hat{a}; f_2) \geq \delta.$$

Therefore Le Cam's method tells us that

$$\inf_{\hat{a}} \sup_{(\mu, R) \in \mathsf{MAB}(C^\star)} \mathbb{E}_{\mathcal{D}}[r(a^\star) - r(\hat{a})] \geq \inf_{\hat{a}} \sup_{f \in f_1, f_2} \mathbb{E}_{\mathcal{D}}[\mathcal{L}(\hat{a}; f)] \geq \frac{\delta}{4} \cdot \exp(-\mathsf{KL}(\mathbb{P}_{\mu \otimes f_1} \| \mathbb{P}_{\mu \otimes f_2})).$$

Here $\mathsf{KL}(\mathbb{P}_{\mu \otimes f_1} \| \mathbb{P}_{\mu \otimes f_2})$ denotes the KL divergence between the two MAB instances with $N$ samples. Direct calculations yield

$$\mathsf{KL}(\mathbb{P}_{\mu \otimes f_1} \| \mathbb{P}_{\mu \otimes f_2}) \leq \frac{N \mathsf{KL}(\mathbb{P}_{f_1} \| \mathbb{P}_{f_2})}{C^\star} \leq \frac{N(2\delta)^2}{C^\star(1/4 - \delta^2)} \leq 200 N \delta^2 / C^\star.$$

Here we use the fact that for two Bernoulli distribution, $\mathsf{KL}(\mathrm{Bern}(p) \| \mathrm{Bern}(q)) \leq (p - q)^2 / [q(1 - q)]$ and that $\delta \in [0, 1/4]$. Taking

$$\delta = \min\left\{\frac{1}{4}, \sqrt{\frac{C^\star}{N}}\right\}$$

yields the desired lower bound for $C^\star \geq 2$.

**The case of $C^\star \in (1, 2)$.** Recall that when $C^\star \geq 2$, we construct the same behavior distribution $\mu$ for two different reward distributions $f_1, f_2$. In contrast, in the case of $C^\star \in [1, 2)$, we construct instances that are different in both the reward distributions as well as the behavior distribution. More specifically, let $\mu_1(1) = 1/C^\star$, $\mu_1(2) = 1 - 1/C^\star$, $f_1 = (\frac{1}{2} + \delta, \frac{1}{2})$ for some $\delta > 0$ which will be specified later. Similarly, we let $\mu_2(1) = 1 - 1/C^\star$, $\mu_2(2) = 1/C^\star$, $f_2 = (\frac{1}{2}, \frac{1}{2} + \delta)$. It is straightforward to check that $(\mu_1, f_1), (\mu_2, f_2) \in \mathsf{MAB}(C^\star)$. Clearly, for any estimator $\hat{a}$, we have

$$\mathcal{L}(\hat{a}; f_1) + \mathcal{L}(\hat{a}; f_2) \geq \delta.$$

Again, applying Le Cam's method, we have

$$\inf_{\hat{a}} \sup_{(\mu, R) \in \mathsf{MAB}(C^\star)} \mathbb{E}_{\mathcal{D}}[r(a^\star) - r(\hat{a})] \geq \frac{\delta}{4} \cdot \exp(-\mathsf{KL}(\mathbb{P}_{\mu_1 \otimes f_1} \| \mathbb{P}_{\mu_2 \otimes f_2})). \tag{5}$$

Note that

$$
\begin{aligned}
\mathsf{KL}(\mathbb{P}_{\mu_1 \otimes f_1} \| \mathbb{P}_{\mu_2 \otimes f_2}) &\leq N \cdot \Big( \frac{\frac{1}{2} + \delta}{C^\star} \log(\frac{1 + 2\delta}{C^\star - 1}) + \frac{\frac{1}{2} - \delta}{C^\star} \log(\frac{1 - 2\delta}{C^\star - 1}) \\
&\qquad + \frac{1 - \frac{1}{C^\star}}{2} \log(\frac{C^\star - 1}{1 + 2\delta}) + \frac{1 - \frac{1}{C^\star}}{2} \log(\frac{C^\star - 1}{1 - 2\delta}) \Big) \\
&= N \cdot \Big( \Big( \frac{1 + \delta}{C^\star} - \frac{1}{2} \Big) \log \Big( \frac{1 + 2\delta}{C^\star - 1} \Big) + \Big( \frac{1 - \delta}{C^\star} - \frac{1}{2} \Big) \log \Big( \frac{1 - 2\delta}{C^\star - 1} \Big) \Big).
\end{aligned}
$$

Taking $\delta = \frac{2 - C^\star}{2}$, we get $\mathsf{KL}(\mathbb{P}_{\mu_1 \otimes f_1} \| \mathbb{P}_{\mu_2 \otimes f_2}) \leq N \cdot \frac{2 - C^\star}{C^\star} \cdot \log \Big( \frac{2}{C^\star - 1} \Big)$. Thus we know that

$$\inf_{\hat{a}} \sup_{(\mu, R) \in \mathsf{MAB}(C^\star)} \mathbb{E}_{\mathcal{D}}[r(a^\star) - r(\hat{a})] \gtrsim \exp \Big( -(2 - C^\star) \cdot \log \Big( \frac{2}{C^\star - 1} \Big) \cdot N \Big). \tag{6}$$

This finishes the proof of the lower bound for $C^\star \in (1, 2)$.

## B.4 Proof of Proposition 2

To begin with, we have $\mathbb{E}[r(a^\star) - r(\hat{a})] \leq \mathbb{P}(\hat{a} \neq a^\star)$, where we have used the fact that the rewards are bounded between 0 and 1. Thus it is sufficient to control $\mathbb{P}(\hat{a} \neq a^\star)$, which obeys

$$\mathbb{P}(\hat{a} \neq a^\star) = \mathbb{P}(\exists a \neq a^\star, N(a) \geq N(a^\star)) \leq \mathbb{P}(N - N(a^\star) \geq N(a^\star)) = \mathbb{P}(N(a^\star) \leq \frac{N}{2}).$$

Applying the Chernoff bound for binomial random variables yields

$$\mathbb{P}(N(a^\star) \leq \frac{N}{2}) \leq \exp \Big( -N \cdot \mathsf{KL} \Big( \mathsf{Bern} \Big( \frac{1}{2} \Big) \, \Big\| \, \mathsf{Bern} \Big( \frac{1}{C^\star} \Big) \Big) \Big).$$

Taking the previous steps collectively to arrive at the desired conclusion.

## B.5 Proof of Theorem 3

We prove the case when $C^\star = 1.5$ and when $C^\star = 6$ separately.

**The case when $C^\star = 1.5$.** We begin by introducing the MAB problem.

**The bandit instance.** Consider a two-armed bandit problem with the optimal arm denoted by $a^\star$ and the sub-optimal arm $a$. We set $\mu(a^\star) = 1/C^\star$, and $\mu(a) = 1 - 1/C^\star$ in accordance with the requirement $1/\mu(a^\star) \leq C^\star$. We consider the following reward distributions: the optimal arm $a^\star$ has a deterministic reward equal to $1/2$ whereas the sub-optimal arm has a reward distribution of $\mathsf{Bern}(1/2 - g)$ for some $g \in (0, 1/3)$, which will be specified momentarily. It is straightforward to check that the arm $a^\star$ is indeed optimal and the MAB problem $(\mu, R)$ belongs to $\mathsf{MAB}(C^\star)$.

**Lower bounding the performance of LCB.**   For the two-armed bandit problem introduced above, we have

$$\mathbb{E}_{\mathcal{D}}[r(a^\star) - r(\hat{a})] = g \cdot \mathbb{P}(\text{LCB chooses arm } a)$$

$$= g \sum_{k=0}^{N} \mathbb{P}(\text{LCB chooses arm } a \mid N(a) = k)\mathbb{P}(N(a) = k)$$

$$\geq g \sum_{k=N\mu(a)/2}^{2N\mu(a)} \mathbb{P}(\text{LCB chooses arm } a \mid N(a) = k)\mathbb{P}(N(a) = k), \qquad (7)$$

where we restrict ourselves to the event

$$\mathcal{E} := \{\tfrac{1}{2}N\mu(a) \leq N(a) \leq 2N\mu(a)\}.$$

It turns out that when $1 \leq k \leq 2N\mu(a)$, one has

$$\mathbb{P}(\text{LCB chooses arm } a \mid N(a) = k) \geq \frac{1}{\sqrt{4N\mu(a)}} \cdot \exp\left(-\frac{(g\sqrt{2N\mu(a)} + \sqrt{L})^2}{\frac{1}{4} - g^2}\right). \qquad (8)$$

Combine inequalities (7) and (8) to obtain

$$\mathbb{E}_{\mathcal{D}}[r(a^\star) - r(\hat{a})] \geq g\frac{1}{\sqrt{4N\mu(a)}} \cdot \exp\left(-\frac{(g\sqrt{2N\mu(a)} + \sqrt{L})^2}{\frac{1}{4} - g^2}\right)\mathbb{P}(\mathcal{E}).$$

Setting $g = \min\{1/3, \sqrt{L/(2N\mu(a))}\}$ yields

$$\mathbb{E}_{\mathcal{D}}[r(a^\star) - r(\hat{a})] \geq \frac{\min\left(\sqrt{L/(2N\mu(a))}, \frac{1}{3}\right)}{\sqrt{4N\mu(a)}} \cdot \exp\left(-32L\right)\mathbb{P}(\mathcal{E})$$

$$\geq \min\left(\frac{\sqrt{L}}{8N\mu(a)}, \frac{1}{12\sqrt{N\mu(a)}}\right) \cdot \exp\left(-32L\right),$$

where the last inequality uses Chernoff's bound, i.e., $\mathbb{P}(\mathcal{E}) \geq 1 - 2\exp(-N\mu(a)/8) \geq \frac{1}{2}$. Substituting the definition of $L$ and $\mu(a)$ completes the proof.

*Proof of the lower bound* (8). By the definition of LCB, we have

$$\mathbb{P}(\text{LCB chooses arm } a \mid N(a) = k) = \mathbb{P}\left(1/2 - \sqrt{L/N(a^\star)} \leq \hat{r}(a) - \sqrt{L/N(a)} \mid N(a) = k\right)$$

$$\geq \mathbb{P}\left(\hat{r}(a) \geq 1/2 + \sqrt{L/N(a)} \mid N(a) = k\right)$$

$$\geq \frac{1}{\sqrt{2k}} \cdot \exp\left(-k \cdot \mathsf{KL}\left(\frac{1}{2} - \sqrt{\frac{L}{k}}\Big\|\frac{1}{2} + g\right)\right)$$

$$\geq \frac{1}{\sqrt{2k}} \cdot \exp\left(-\frac{k(g + \sqrt{\frac{L}{k}})^2}{\frac{1}{4} - g^2}\right).$$

Here, the penultimate inequality comes from a lower bound for Binomial tails ([Robert, 1990](#)) and the last inequality uses the elementary fact that $\mathsf{KL}(p\|q) \le (p-q)^2/q(1-q)$. One can easily see that the probability lower bound is decreasing in $k$ and hence when $N(a) = k \le 2N\mu(a)$, we have

$$\mathbb{P}(\text{LCB chooses the arm } a \mid N(a) = k) \ge \frac{1}{\sqrt{4N\mu(a)}} \cdot \exp\left(-\frac{(g\sqrt{2N\mu(a)} + \sqrt{L})^2}{\frac{1}{4} - g^2}\right).$$

This completes the proof. $\qquad\square$

**The case when $C^\star = 6$.** We now prove the lower bound for the case of $C^\star = 6$.

**The bandit instance.** Consider a two-armed bandit problem with $\mu(a^\star) = \frac{1}{C^\star}$ for the optimal arm and $\mu(a) = 1 - \frac{1}{C^\star}$ for the sub-optimal arm, which satisfies the concentrability requirement. We set the following reward distributions: the optimal arm $a^\star$ is distributed according to $\mathsf{Bern}(1/2)$ and the sub-optimal arm has a deterministic reward equal to $1/2 - g$ for some $g \in (0, 1/2)$, which will be specified momentarily. It is immediate that $a^\star$ is optimal in this construction and that the MAB problem $(\mu, R)$ belongs to $\mathsf{MAB}(C^\star)$.

**Lower bounding the performance of LCB.** Similar arguments as before give

$$\mathbb{E}_{\mathcal{D}}[r(a^\star) - r(\hat{a})] \ge g \sum_{k=N\mu(a^\star)/2}^{2N\mu(a^\star)} \mathbb{P}(\text{LCB chooses arm } a \mid N(a^\star) = k)\mathbb{P}(N(a^\star) = k), \qquad (9)$$

where we restrict ourselves to the event (with abuse of notation)

$$\mathcal{E} := \{\frac{1}{2}N\mu(a^\star) \le N(a^\star) \le 2N\mu(a^\star)\}.$$

By the definition of LCB, when $C^\star = 6$ and $\frac{1}{2}N\mu(a^\star) \le k \le 2N\mu(a^\star) \le \frac{1}{3}N$, one has

$$\begin{aligned}
\mathbb{P}(\text{LCB chooses arm } a \mid N(a^\star) = k) &= \mathbb{P}\left(\hat{r}(a^\star) - \sqrt{L/N(a^\star)} \le \frac{1}{2} - g - \sqrt{L/N(a)} \mid N(a^\star) = k\right) \\
&= \mathbb{P}\left(\hat{r}(a^\star) \le 1/2 - g + \sqrt{L/k} - \sqrt{L/(N-k)} \mid N(a^\star) = k\right) \\
&\ge \mathbb{P}\left(\hat{r}(a^\star) \le 1/2 - g + \sqrt{3L/N} - \sqrt{3L/(2N)} \mid N(a^\star) = k\right) \\
&> \mathbb{P}\left(\hat{r}(a^\star) \le 1/2 - g + \sqrt{\frac{L}{4N}} \mid N(a^\star) = k\right).
\end{aligned}$$

We set $g = \min\{\sqrt{L/(4N)}, 1/2\}$. Under this choice of $g$, we always have

$$\mathbb{P}(\text{LCB chooses arm } a \mid N(a^\star) = k) \ge \frac{1}{2}. \qquad (10)$$

Combine the inequalities (9) and (10) to obtain

$$\mathbb{E}_{\mathcal{D}}[r(a^\star) - r(\hat{a})] \ge g \cdot \frac{1}{2} \cdot \mathbb{P}(\mathcal{E}) \ge \frac{\min(1, \sqrt{L/N})}{8}.$$

# C   Proofs for contextual bandits

In Section C.1, we prove the sub-optimality guarantee of the LCB approach for contextual bandits stated in Theorem 4. In Section C.2 we prove Theorem **??**—a minimax lower bound for contextual bandits. In the end, we prove the failure of the most played arm approach in Section C.3.

## C.1   Proof of Theorem 4

We prove a stronger version of Theorem 4: Fix a deterministic expert policy $\pi$ that is not necessarily optimal. We assume that

$$\max_s \frac{\rho(s)}{\mu(s, \pi(s))} \leq C^{\pi}.$$

Setting $\delta = 1/N$, the policy $\hat{\pi}$ returned by Algorithm 1 obeys

$$\mathbb{E}_{\mathcal{D}}[J(\pi) - J(\hat{\pi})] \lesssim \min\left(1, \widetilde{O}\left(\sqrt{\frac{S(C^{\pi} - 1)}{N}} + \frac{S}{N}\right)\right).$$

The statement in Theorem 4 can be recovered when we take $\pi = \pi^{\star}$.

We begin with defining a good event

$$\mathcal{E} := \{\forall s, a : |r(s, a) - \hat{r}(s, a)| \leq b(s, a)\}, \tag{11}$$

on which the penalty function $b(s, a)$ provides a valid upper bound on the reward estimation error $r(s, a) - \hat{r}(s, a)$. With this definition in place, we state a key decomposition of the sub-optimality of the LCB method:

$$\mathbb{E}_{\mathcal{D}}\left[\sum_s \rho(s)\left[r(s, \pi(s)) - r(s, \hat{\pi}(s))\right]\right]$$

$$= \mathbb{E}_{\mathcal{D}}\left[\sum_s \rho(s)\left[r(s, \pi(s)) - r(s, \hat{\pi}(s))\right] \mathbb{1}\{N(s, \pi(s)) = 0\}\right] =: T_1$$

$$+ \mathbb{E}_{\mathcal{D}}\left[\sum_s \rho(s)\left[r(s, \pi(s)) - r(s, \hat{\pi}(s))\right] \mathbb{1}\{N(s, \pi(s)) \geq 1\}\mathbb{1}\{\mathcal{E}\}\right] := T_2$$

$$+ \mathbb{E}_{\mathcal{D}}\left[\sum_s \rho(s)\left[r(s, \pi(s)) - r(s, \hat{\pi}(s))\right] \mathbb{1}\{N(s, \pi(s)) \geq 1\}\mathbb{1}\{\mathcal{E}^c\}\right] := T_3.$$

In words, the term $T_1$ corresponds to the error induced by missing mass, i.e., when the expert action $\pi(s)$ is not seen in the data $\mathcal{D}$. The second term $T_2$ denotes the error when the good event $\mathcal{E}$ takes place. The last term $T_3$ denotes the sub-optimality incurred under the complement event $\mathcal{E}^c$.

To avoid cluttered notation, we denote $L := 2000\sqrt{2\log(S|\mathcal{A}|N)}$ such that $b(s, a) = \sqrt{L/N(s, a)}$ when $N(s, a) \geq 1$. These three error terms obey the following upper bounds, whose proofs are provided in subsequent subsections:

$$T_1 \leq \frac{4SC^{\pi}}{9N}; \tag{12a}$$

$$T_2 \lesssim \frac{SC^{\pi}}{N}L + \sqrt{\frac{S(C^{\pi} - 1)}{N}L} + \frac{1}{N^9}; \tag{12b}$$

$$T_3 \leq \frac{1}{N}. \tag{12c}$$

Combining the above three bounds together with the fact that $\mathbb{E}_{\mathcal{D}}[J(\pi) - J(\hat{\pi})] \le 1$ yields that

$$\mathbb{E}_{\mathcal{D}}[J(\pi) - J(\hat{\pi})] \lesssim \min\left(1, \widetilde{O}\left(\sqrt{\frac{S(C^\pi - 1)}{N}} + \frac{SC^\pi}{N}\right)\right).$$

Note that if $C^\pi \ge 2$, the first term $\sqrt{\frac{S(C^\pi - 1)}{N}}$ always dominates. Conversely, if $C^\pi < 2$, we can omit the extra $C^\pi$ in the second term $\frac{SC^\pi}{N}$. This gives the desired claim in Theorem 4.

### C.1.1 Proof of the bound (12a) on $T_1$

Since $r(s, \pi(s)) - r(s, \hat{\pi}(s)) \le 1$ for any $\hat{\pi}(s)$, one has

$$T_1 \le \mathbb{E}_{\mathcal{D}}\left[\sum_s \rho(s)\mathbb{1}\{N(s, \pi(s)) = 0\}\right] = \sum_s \rho(s)\mathbb{P}(N(s, \pi(s)) = 0)$$

$$= \sum_s \rho(s)(1 - \mu(s, \pi(s)))^N.$$

Recall the assumption that $\max_s \frac{\rho(s)}{\mu(s, \pi(s))} \le C^\pi$. We can continue the upper bound of $T_1$ to obtain

$$T_1 \le \sum_s C^\pi \mu(s, \pi(s))(1 - \mu(s, \pi(s)))^N \le \sum_s C^\pi \frac{4}{9N} = \frac{4}{9N}SC^\pi.$$

Here, the last inequality holds since $\max_{x\in[0,1]} x(1 - x)^N \le 4/(9N)$.

### C.1.2 Proof of the bound (12b) on $T_2$

For any state $s \in \mathcal{S}$, define the total mass on sub-optimal actions to be

$$\bar{\mu}(s) := \sum_{a:a\neq\pi(s)} \mu(s, a).$$

We can then partition the state space into the following three disjoint sets:

$$\mathcal{S}_1 := \left\{s \mid \rho(s) < \frac{2C^\pi L}{N}\right\}, \tag{13a}$$

$$\mathcal{S}_2 := \left\{s \mid \rho(s) \ge \frac{2C^\pi L}{N}, \mu(s, \pi(s)) \ge 10\bar{\mu}(s)\right\}, \tag{13b}$$

$$\mathcal{S}_3 := \left\{s \mid \rho(s) \ge \frac{2C^\pi L}{N}, \mu(s, \pi(s)) < 10\bar{\mu}(s)\right\}. \tag{13c}$$

The set $\mathcal{S}_1$ includes the states that are "less important" in evaluating the performance of LCB. The set $\mathcal{S}_2$ captures the states for which the expert action $\pi(s)$ is drawn more frequently under the behavior distribution $\mu$.

With this partition at hand, we can decompose the term $T_2$ accordingly:

$$T_2 = \sum_{s\in\mathcal{S}_1} \rho(s)\mathbb{E}_{\mathcal{D}}\left[[r(s, \pi(s)) - r(s, \hat{\pi}(s))]\,\mathbb{1}\{N(s, \pi(s)) \ge 1\}\mathbb{1}\{\mathcal{E}\}\right] =: T_{2,1}$$

$$+ \sum_{s\in\mathcal{S}_2} \rho(s)\mathbb{E}_{\mathcal{D}}\left[[r(s, \pi(s)) - r(s, \hat{\pi}(s))]\,\mathbb{1}\{N(s, \pi(s)) \ge 1\}\mathbb{1}\{\mathcal{E}\}\right] =: T_{2,2}$$

$$+ \sum_{s\in\mathcal{S}_3} \rho(s)\mathbb{E}_{\mathcal{D}}\left[[r(s, \pi(s)) - r(s, \hat{\pi}(s))]\,\mathbb{1}\{N(s, \pi(s)) \ge 1\}\mathbb{1}\{\mathcal{E}\}\right] =: T_{2,3}.$$

The proof is completed by observing the following three upper bounds:

$$T_{2,1} \leq \frac{2SC^\pi L}{N}; \qquad T_{2,2} \lesssim \frac{1}{N^9}; \qquad T_{2,3} \lesssim \sqrt{\frac{C^\pi SL}{N}} \min\{1, 10(C^\pi - 1)\} \lesssim \sqrt{\frac{(C^\pi - 1)SL}{N}}.$$

**Proof of the bound on $T_{2,1}$.** We again use the basic fact that

$$[r(s, \pi(s)) - r(s, \hat{\pi}(s))] \, \mathbb{1}\{N(s, \pi(s)) \geq 1\} \mathbb{1}\{\mathcal{E}\} \leq 1$$

to reach

$$T_{2,1} \leq \sum_{s \in \mathcal{S}_1} \rho(s) \leq \frac{2SC^\pi L}{N},$$

where the last inequality hinges on the definition (13a) of $\mathcal{S}_1$, namely for any $s \in \mathcal{S}_1$, one has $\rho(s) < \frac{2C^\pi L}{N}$.

**Proof of the bound on $T_{2,2}$.** Fix a state $s \in \mathcal{S}_2$, we define the following two sets of actions:

$$\mathcal{A}_1(s) := \{a \mid r(s, a) < r(s, \pi(s)), \mu(s, a) \leq L/(200N)\},$$
$$\mathcal{A}_2(s) := \{a \mid r(s, a) < r(s, \pi(s)), \mu(s, a) > L/(200N)\}.$$

Further define $A(s, a)$ to be the event that $\hat{r}(s, \pi(s)) - b(s, \pi(s)) < \hat{r}(s, a) - b(s, a)$. Clearly one has $r(s, \pi(s)) - r(s, \hat{\pi}(s)) \leq \mathbb{1}\{\cup_{a \in \mathcal{A}_1(s) \cup \mathcal{A}_2(s)} A(s, a)\}$. Consequently, we can write the following decomposition:

$$\mathbb{E}_{\mathcal{D}}\left[[r(s, \pi(s)) - r(s, \hat{\pi}(s))] \, \mathbb{1}\{N(s, \pi(s)) \geq 1\} \, \mathbb{1}\{\mathcal{E}\}\right]$$
$$\leq \mathbb{P}(\exists a, r(s, a) < r(s, \pi(s)), A(s, a), N(s, \pi(s)) \geq 1)$$
$$\leq \mathbb{P}(\exists a \in \mathcal{A}_1(s), A(s, a), N(s, \pi(s)) \geq 1) =: p_1(s)$$
$$+ \mathbb{P}(\exists a \in \mathcal{A}_2(s), A(s, a), N(s, \pi(s)) \geq 1) =: p_2(s).$$

As a result, $T_{2,2}$ obeys

$$T_{2,2} \leq \sum_{s \in \mathcal{S}_2} \rho(s) p_1(s) + \sum_{s \in \mathcal{S}_2} \rho(s) p_2(s), \qquad (14)$$

which satisfy the bounds

$$\sum_{s \in \mathcal{S}_2} \rho(s) p_1(s) \lesssim \frac{1}{N^{10}}, \quad \text{and} \quad \sum_{s \in \mathcal{S}_2} \rho(s) p_2(s) \lesssim \frac{1}{N^9}.$$

Taking these two bounds collectively leads us to the desired conclusion. In what follows, we focus on the proving the aforementioned two bounds.

**Proof of the bound on $\sum_{s \in \mathcal{S}_2} \rho(s) p_1(s)$.** Fix a state $s \in \mathcal{S}_2$. In view of the data coverage assumption, one has

$$\mu(s, \pi(s)) \geq \frac{\rho(s)}{C^\pi} \geq \frac{2L}{N}. \qquad (15)$$

In contrast, for any $a \in \mathcal{A}_1(s)$, we have

$$\mu(s, a) \leq \frac{L}{200N}. \qquad (16)$$

Therefore one has $\mu(s, \pi(s)) \gg \mu(s, a)$ for any non-expert action $a$. As a result, the optimal action is selected more frequently than the sub-optimal ones. It turns out that under such circumstances, the LCB algorithm picks the right action with high probability. We make this intuition precise below.

The bounds (15) and (16) together with Chernoff's bound give

$$\mathbb{P}\left(N(s, a) \leq \frac{5L}{200}\right) \geq 1 - \exp\left(-\frac{L}{200}\right);$$

$$\mathbb{P}\left(N(s, \pi(s)) > L\right) \geq 1 - \exp\left(-\frac{L}{4}\right).$$

These allow us to obtain an upper bound for the function $\hat{r} - b$ evaluated at sub-optimal actions and a lower bound on $\hat{r}(s, \pi(s)) - b(s, \pi(s))$. More precisely, if $N(s, a) = 0$, we know that $\hat{r}(s, a) = -1$; when $1 \leq N(s, a) \leq \frac{5L}{200}$, we have

$$\hat{r}(s, a) - b(s, a) \leq 1 - \sqrt{\frac{L}{5L/200}} \leq -5.$$

Now we turn to lower bounding the function $\hat{r} - b$ evaluated at the optimal action. When $N(s, \pi(s)) > L$, one has

$$\hat{r}(s, \pi(s)) - b(s, \pi(s)) > -\sqrt{\frac{L}{N(s, \pi(s))}} = -1.$$

To conclude, if both $N(s, a) \leq \frac{5L}{200}$ and $N(s, \pi(s)) \geq L$ hold, we must have $\hat{r}(s, a) - b(s, a) < \hat{r}(s, \pi(s)) - b(s, \pi(s))$. Therefore we can deduce that

$$\sum_{s \in \mathcal{S}_2} \rho(s) p_1(s) = \sum_{s \in \mathcal{S}_2} \rho(s) \mathbb{P}(\exists a \in \mathcal{A}_1(s), A(s, a), N(s, \pi(s)) \geq 1)$$

$$\leq (|\mathcal{A}| - 1) \exp\left(-\frac{L}{200}\right) + \exp\left(-\frac{1}{4}L\right)$$

$$\leq |\mathcal{A}| \exp\left(-\frac{L}{200}\right)$$

$$\lesssim \frac{1}{N^{10}}.$$

The last inequality comes from the choice of $L = 2000 \log(2S|\mathcal{A}|N)$.

**Proof of the bound on $\sum_{s \in \mathcal{S}_2} \rho(s) p_2(s)$.** Before embarking on the proof of $\sum_{s \in \mathcal{S}_2} \rho(s) p_2(s) \lesssim \frac{1}{N^9}$, it is helpful to pause and gather a few useful properties of $(s, a)$ with $s \in \mathcal{S}_2$, $a \in \mathcal{A}_2(s)$:

1. $\rho(s) \geq \frac{2C^\pi L}{N}$ and hence $\mu(s, \pi(s)) \geq \frac{2L}{N}$ by the definition of $C^\pi$;

2. $\frac{L}{200N} \leq \mu(s, a) \leq \frac{1}{10}\mu(s, \pi(s))$;

3. $\sum_{a \in \mathcal{A}_2} \mu(s, a) \leq \frac{1}{10}\mu(s, \pi(s))$;

4. $|\mathcal{A}_2(s)| \leq 200N/L$.

In addition, we define a high probability event on which the sample sizes $N(s,a)$ concentrate around their respective means $N\mu(s,a)$:

$$\mathcal{E}_2(s) := \Bigg\{ \frac{1}{2} N\mu(s, \pi(s)) \leq N(s, \pi(s)) \leq 2N\mu(s, \pi(s)),$$

$$\forall a \in \mathcal{A}_2(s), \frac{1}{2} N\mu(s, a) \leq N(s, a) \leq 2N\mu(s, a) \Bigg\},$$

which—in view of the Chernoff bound and the union bound—obeys

$$\mathbb{P}(\mathcal{E}_2(s)) \geq 1 - 1/N^9. \tag{17}$$

With these preparations in place, we can derive

$$\begin{aligned}
p_2(s) &= \mathbb{P}(\exists a \in \mathcal{A}_2, A(s, a), N(s, \pi(s)) \geq 1) \\
&\leq \mathbb{P}(\mathcal{E}_2^c(s)) + \mathbb{P}(\exists a \in \mathcal{A}_2, A(s, a), N(s, \pi(s)) \geq 1, \mathcal{E}_2(s)) \\
&\leq \mathbb{P}(\mathcal{E}_2^c(s)) + \sum_{a \in \mathcal{A}_2} \mathbb{P}(A(s, a), N(s, \pi(s)) \geq 1, \mathcal{E}_2(s)) \\
&\lesssim \frac{1}{N^9} + \frac{|\mathcal{A}_2|}{N^{10}} \lesssim \frac{1}{N^9},
\end{aligned}$$

where the last line arises from the bound

$$\mathbb{P}(A(s, a), N(s, \pi(s)) \geq 1, \mathcal{E}_2(s)) \lesssim \frac{1}{N^{10}}, \tag{18}$$

and the cardinality upper bound $|\mathcal{A}_2(s)| \lesssim N$. This completes the bound on $\sum_{s \in \mathcal{S}_2} p(s)$.

*Proof of the bound* (18). On the event $\mathcal{E}_2(s)$, one must have $N(s, a) \geq 1$ and $N(s, \pi(s)) \geq 1$. Therefore, we can define

$$\epsilon := \sqrt{\frac{L}{N(s, a)}} - \sqrt{\frac{L}{N(s, \pi(s))}} \quad \text{and} \quad \Delta = r(s, \pi(s)) - r(s, a),$$

and obtain the following bound on the conditional probability

$$\mathbb{P}\left( \hat{r}(s, a) - \sqrt{\frac{L}{N(s, a)}} \geq \hat{r}(s, \pi(s)) - \sqrt{\frac{L}{N(s, \pi(s))}} \,\Bigg|\, N(s, \pi(s)), N(s, a), \mathcal{E}_2 \right)$$

$$\leq \exp\left( -2 \frac{N(s, a) N(s, \pi(s)) (\epsilon + \Delta)^2}{N(s, a) + N(s, \pi(s))} \,\Bigg|\, N(s, \pi(s)), N(s, a), \mathcal{E}_2 \right),$$

where the inequality arises from Lemma 13. Note that under event $\mathcal{E}_2(s)$ and the property $\mu(s, a) \leq \frac{1}{10} \mu(s, \pi(s))$, we have $N(s, \pi(s)) \geq 4N(s, a)$ and thus $\epsilon \geq \frac{1}{2} \sqrt{\frac{L}{N(s,a)}}$. This allows us to further upper bound the probability as

$$\mathbb{P}\left( \hat{r}(s, a) - \sqrt{\frac{L}{N(s, a)}} \geq \hat{r}(s, \pi(s)) - \sqrt{\frac{L}{N(s, \pi(s))}} \,\Bigg|\, N(s, \pi(s)), N(s, a), \mathcal{E}_2 \right)$$

$$\leq \exp\left( -N(s, a)(\epsilon + \Delta)^2 \right)$$

$$\leq \exp\left( -\left( \frac{1}{2} \sqrt{L} + \sqrt{N(s, a)} \Delta \right)^2 \right)$$

$$\leq \exp\left( -\frac{1}{4} L \right) \lesssim \frac{1}{N^{10}},$$

under the choice of $L = 2000 \log(2S|\mathcal{A}|N)$. Since this upper bound holds for any configuration of $N(s, a)$ and $N(s, \pi(s))$, one has the desired claim. $\qquad\square$

**Proof of the bound on $T_{2,3}$.** On the good event $\mathcal{E}$, we know that

$$
\begin{aligned}
r(s, \pi(s)) - r(s, \hat{\pi}(s)) &\leq r(s, \pi(s)) - [\hat{r}(s, \hat{\pi}(s)) - b(s, \hat{\pi}(s))] \\
&\leq r(s, \pi(s)) - [\hat{r}(s, \pi(s)) - b(s, \pi(s))] \\
&\leq 2b(s, \pi(s)).
\end{aligned}
$$

Here the middle line arises from the definition of the LCB algorithm, i.e., $\hat{\pi}(s) \in \arg\max_a \hat{r}(s, a) - b(s, a)$ for each $s$. Substitute this upper bound into the definition of $T_2$ to obtain

$$
\begin{aligned}
T_{2,3} &\leq 2 \sum_{s \in \mathcal{S}_3} \rho(s) \mathbb{E}_{\mathcal{D}} \left[ b(s, \pi(s)) \mathbb{1}\{N(s, \pi(s)) \geq 1\} \mathbb{1}\{\mathcal{E}\} \right] \\
&= 2 \sum_{s \in \mathcal{S}_3} \rho(s) \mathbb{E}_{\mathcal{D}} \left[ \sqrt{\frac{L}{N(s, \pi(s))}} \mathbb{1}\{N(s, \pi(s)) \geq 1\} \mathbb{1}\{\mathcal{E}\} \right] \\
&\leq 2\sqrt{L} \sum_{s \in \mathcal{S}_3} \rho(s) \mathbb{E}_{\mathcal{D}} \left[ \sqrt{\frac{1}{N(s, \pi(s)) \vee 1}} \mathbb{1}\{N(s, \pi(s)) \geq 1\} \right],
\end{aligned}
$$

where we have used the definition of $b(s, a)$. Lemma 14 tells us that there exists a universal constant $c > 0$ such that

$$
\mathbb{E}_{\mathcal{D}} \left[ \sqrt{\frac{1}{N(s, \pi(s)) \vee 1}} \mathbb{1}\{N(s, \pi(s)) \geq 1\} \right] \leq \frac{c}{\sqrt{N\mu(s, \pi(s))}}.
$$

As a result, we reach the conclusion that

$$
T_{2,3} \leq 2\sqrt{L} \sum_{s \in \mathcal{S}_3} \rho(s) \frac{c}{\sqrt{N\mu(s, \pi(s))}}.
$$

In view of the assumption $\max_s \rho(s)/\mu(s, \pi(s)) \leq C^{\pi}$, one further has

$$
T_{2,3} \leq 2c\sqrt{\frac{C^{\pi} L}{N}} \sum_{s \in \mathcal{S}_3} \sqrt{\rho(s)} \leq 2c\sqrt{\frac{C^{\pi} L}{N}} \sqrt{S} \sqrt{\sum_{s \in \mathcal{S}_3} \rho(s)},
$$

with the last inequality arising from Cauchy-Schwarz's inequality. The desired bound on $T_{2,3}$ follows from the following simple fact regarding $\sum_{s \in \mathcal{S}_3} \rho(s)$:

$$
\sum_{s \in \mathcal{S}_3} \rho(s) \leq \min\{1, 10(C^{\pi} - 1)\}. \tag{19}
$$

*Proof of the inequality* (19). The upper bound 1 is trivial to see. To achieve the other upper bound, we first use the assumption $\max_s \rho(s)/\mu(s, \pi(s)) \leq C^{\pi}$ to see

$$
\sum_{s \in \mathcal{S}_3} \rho(s) \leq \sum_{s \in \mathcal{S}_3} C^{\pi} \mu(s, \pi(s)) \leq 10 C^{\pi} \sum_{s \in \mathcal{S}_3} \bar{\mu}(s).
$$

Here the last relation follows from the definition of $\mathcal{S}_3$. Note that

$$
\sum_{s \in \mathcal{S}_3} \bar{\mu}(s) \leq \sum_s \bar{\mu}(s) = 1 - \sum_s \mu(s, \pi(s)) \leq 1 - \frac{1}{C^{\pi}},
$$

where we have reused the assumption $\max_s \rho(s)/\mu(s, \pi(s)) \leq C^{\pi}$. Taking the previous two inequalities collectively yields the final claim. $\qquad\square$

### C.1.3 Proof of the bound (12c) on $T_3$

It is not hard to see that

$$\sum_s \rho(s) \left[ r(s, \pi(s)) - r(s, \hat{\pi}(s)) \right] \mathbb{1}\{N(s, \pi(s)) \geq 1\} \leq 1,$$

which further implies

$$T_3 \leq \mathbb{E}_{\mathcal{D}} \left[ \mathbb{1}\{\mathcal{E}^c\} \right] = \mathbb{P}(\mathcal{E}^c).$$

It then boils down to upper bounding the probability $\mathbb{P}(\mathcal{E}^c)$. The proof is similar in spirit to that of Lemma 1.

Fix a state-action pair $(s, a)$. If $N(s, a) = 0$, one clearly has $-1 = \hat{r}(s, a) - b(s, a) \leq r(s, a) \leq \hat{r}(s, a) + b(s, a) = 1$. Therefore we concentrate on the case when $N(s, a) \geq 1$. Apply the Hoeffding's inequality to see that for any $\delta_1 \in (0, 1)$, one has

$$\mathbb{P}\left( |\hat{r}(s, a) - r(s, a)| \geq \sqrt{\frac{\log(2/\delta_1)}{2N(s, a)}} \mid N(s, a) \right) \leq \delta_1.$$

In particular, setting $\delta_1 = \delta/(S|\mathcal{A}|)$ yields

$$\mathbb{P}\left( |\hat{r}(s, a) - r(s, a)| \geq \sqrt{\frac{\log(2S|\mathcal{A}|/\delta)}{2N(s, a)}} \mid N(s, a) \right) \leq \frac{\delta}{S|\mathcal{A}|}, \tag{20}$$

Recall that $b(s, a)$ is defined such that when $N(s, a) \geq 1$,

$$b(s, a) = \sqrt{\frac{2000 \log(2S|\mathcal{A}|/\delta)}{N(s, a)}}.$$

Since the inequality (20) holds for any $N(s, a)$, we have for any fixed $(s, a)$,

$$\mathbb{P}\left( |\hat{r}(s, a) - r(s, a)| \geq b(s, a) \right) \leq \frac{\delta}{S|\mathcal{A}|}.$$

Taking a union bound over $\mathcal{S} \times \mathcal{A}$ leads to the conclusion that $\mathbb{P}(\mathcal{E}^c) \leq \delta$, and hence $T_3 \leq \delta$. Taking $\delta = 1/N$ gives the advertised result.

### C.2 Proof of Theorem 5

We prove the lower bound differently for the following regimes: $C^\star = 1$, $C^\star \geq 2$, and $C^\star \in (1, 2)$. When $C^\star = 1$, the offline RL problem reduces to the imitation learning problem in contextual bandits, whose lower bound has been shown in the paper Rajaraman et al. (2020). When $C^\star \in (1, 2)$ or $C^\star \geq 2$, we generalize the lower bound given for the multi-armed bandits with different choices of initial distributions. In what follows, we detail the proofs for each regime.

**The case when $C^\star = 1$.** When $C^\star = 1$, one has $d^\star(s, a) = \mu(s, a)$ for any $(s, a)$ pair. This recovers the imitation learning problem, where the rewards are also included in the dataset. Thus the lower bound proved in Lemma 6 is applicable, which comes from a modified version of Theorem 6 in the paper Rajaraman et al. (2020):

$$\inf_{\hat{\pi}} \sup_{(\rho, \mu, R) \in \mathsf{CB}(1)} \mathbb{E}_{\mathcal{D}}[J(\pi^\star) - J(\hat{\pi})] \gtrsim \min\left( 1, \frac{S}{N} \right). \tag{21}$$

**The case when $C^\star \geq 2$.** Fix a contextual bandit instance $(\rho, \mu, R)$, define the loss/sub-optimality of an estimated policy $\pi$ to be

$$\mathcal{L}(\pi; (\rho, \mu, R)) := J(\pi^\star) - J(\hat{\pi}).$$

We intend to show that when $C^\star \geq 2$,

$$\inf_{\hat{\pi}} \sup_{(\rho, \mu, R) \in \mathsf{CB}(C^\star)} \mathbb{E}_{\mu \otimes R}[\mathcal{L}(\pi; (\rho, \mu, R))] \gtrsim \min\left(1, \sqrt{\frac{SC^\star}{N}}\right). \tag{22}$$

Our proof follows the standard recipe of proving minimax lower bounds, namely, we first construct a family of hard contextual bandit instances, and then apply Fano's inequality to obtain the desired lower bound.

**Construction of hard instances.** Consider a CB with state space $\mathcal{S} := \{1, 2, \ldots, S\}$. Set the initial distribution $\rho_0(s) = 1/S$ for any $s \in \mathcal{S}$. Each state $s \in \mathcal{S}$ is associated with two actions $a_1$ and $a_2$. The behavior distribution for each $s, a$ is specified below

$$\mu_0(s, a_1) = \frac{1}{S} - \frac{1}{SC^\star} \qquad \text{and} \qquad \mu_0(s, a_2) = \frac{1}{SC^\star}.$$

It is easy to check that for any reward distribution $R$, one has $(\rho_0, \mu_0, R) \in \mathsf{CB}(C^\star)$. It remains to construct a set of reward distributions that are nearly indistinguishable from the data. To achieve this goal, we leverage the Gilbert-Varshamov lemma (cf. Lemma 15) to obtain a set $\mathcal{V} \subseteq \{-1, 1\}^S$ that obeys (1) $|\mathcal{V}| \geq \exp(S/8)$ and (2) $\|\boldsymbol{v}_1 - \boldsymbol{v}_2\|_1 \geq S/2$ for any $\boldsymbol{v}_1 \boldsymbol{v}_2 \in \mathcal{V}$ with $\boldsymbol{v}_1 \neq \boldsymbol{v}_2$. With this set $\mathcal{V}$ in place, we can continue to construct the following set of Bernoulli reward distributions

$$\mathcal{R} := \left\{ \left\{ \mathsf{Bern}\left(\frac{1}{2}\right), \mathsf{Bern}\left(\frac{1}{2} + v_s \delta\right) \right\}_{s \in \mathcal{S}} \mid \boldsymbol{v} \in \mathcal{V} \right\}.$$

Here $\delta \in (0, 1/3)$ is a parameter that will be specified later. Each element $\boldsymbol{v} \in \mathcal{V}$ is mapped to a reward distribution such that for the state $s$, the reward distribution associated with $(s, a_2)$ is $\mathsf{Bern}(\frac{1}{2} + v_s \delta)$. In view of the second property of the set $\mathcal{V}$, one has for any policy $\pi$ and any two different reward distributions $R_1, R_2 \in \mathcal{R}$,

$$\mathcal{L}(\pi; (\rho_0, \mu_0, R_1)) + \mathcal{L}(\pi; (\rho_0, \mu_0, R_2)) \geq \frac{\delta}{4}.$$

**Application of Fano's inequality.** Now we are ready to apply Fano's inequality, that is

$$\inf_{\hat{\pi}} \sup_{(\rho_0, \mu_0, R) \mid R \in \mathcal{R}} \mathbb{E}_{\mu_0 \otimes R}[\mathcal{L}(\pi; (\rho_0, \mu_0, R))] \geq \frac{\delta}{8}\left(1 - \frac{N \max_{i \neq j} \mathsf{KL}\left(\mu \otimes R_i \| \mu \otimes R_j\right) + \log 2}{\log |\mathcal{R}|}\right).$$

It then remains to control $\max_{i \neq j} \mathsf{KL}\left(\mu \otimes R_i \| \mu \otimes R_j\right)$ and $\log |\mathcal{R}|$. For the latter quantity, we have

$$\log |\mathcal{R}| = \log |\mathcal{V}| \geq S/8,$$

where the inequality comes from the first property of the set $\mathcal{V}$. With regards to the KL divergence, one has

$$\max_{i \neq j} \mathsf{KL}\left(\mu \otimes R_i \| \mu \otimes R_j\right) \leq S \cdot \frac{1}{SC^\star} \cdot 16\delta^2 = \frac{16\delta^2}{C^\star}.$$

As a result, we conclude that as long as

$$\frac{200N\delta^2}{SC^\star} \le 1,$$

one has

$$\inf_{\hat{\pi}} \sup_{(\rho_0, \mu_0, R)|R \in \mathcal{R}} \mathcal{L}(\pi; (\rho_0, \mu_0, R)) \gtrsim \delta.$$

To finish the proof, we can set $\delta = \sqrt{\frac{SC^\star}{200N}}$ when $\sqrt{\frac{SC^\star}{200N}} < \frac{1}{3}$, and $\delta = \frac{1}{3}$ otherwise. This yields the desired lower bound (22).

**The case when $C^\star \in (1, 2)$.** We intend to show that

$$\inf_{\hat{\pi}} \sup_{(\rho, \mu, R) \in \mathsf{CB}(C^\star)} \mathbb{E}[\mathcal{L}(\pi; (\rho, \mu, R))] \gtrsim \min\left(C^\star - 1, \sqrt{\frac{S(C^\star - 1)}{N}}\right). \tag{23}$$

The proof is similar to that of the previous case, with the difference lying in the construction of $\rho_0$ and $\mu_0$.

**Construction of hard instances.** Consider a CB with state space $\mathcal{S} := \{0, 1, 2, \ldots, S\}$ and action space $\mathcal{A} := \{a_1, a_2\}$. Set the initial distribution $\rho_0(s) = (C^\star - 1)/S$ for any $1 \le s \le S$ and $\rho_0(0) = 2 - C^\star$. Each state $1 \le s \le S$ is associated with two actions $a_1$ and $a_2$ such that

$$\mu_0(s, a_1) = \mu_0(s, a_2) = \frac{C^\star - 1}{SC^\star}.$$

In contrast, for $s = 0$, one has a single action $a_1$ with $\mu_0(0, a_1) = \frac{2 - C^\star}{C^\star}$. Similar to the above case, we have for any reward distribution $R$, that $(\rho_0, \mu_0, R) \in \mathsf{CB}(C^\star)$.

We deploy essentially the same family $\mathcal{R}$ of reward distributions as before with an additional reward of $R(0, a_1) \equiv 0$ on state $s = 0$. As a result, one can show that for any policy $\pi$ and any two different reward distributions $R_1, R_2 \in \mathcal{R}$,

$$\mathcal{L}(\pi; (\rho_0, \mu_0, R_1)) + \mathcal{L}(\pi; (\rho_0, \mu_0, R_2)) \ge \frac{\delta}{4}(C^\star - 1).$$

**Application of Fano's inequality.** Fano's inequality tells us that

$$\inf_{\hat{\pi}} \sup_{(\rho_0, \mu_0, R)|R \in \mathcal{R}} \mathbb{E}[\mathcal{L}(\pi; (\rho_0, \mu_0, R))] \ge \frac{\delta}{8}\left(1 - \frac{N \max_{i \ne j} \mathsf{KL}(\mu \otimes R_i \| \mu \otimes R_j) + \log 2}{S/8}\right).$$

In the current case, we have

$$\max_{i \ne j} \mathsf{KL}(\mu \otimes R_i \| \mu \otimes R_j) \le S \cdot \frac{C^\star - 1}{SC^\star} \cdot 16\delta^2 = \frac{16(C^\star - 1)}{C^\star}\delta^2.$$

As before, setting

$$\delta = \min\left(\sqrt{\frac{SC^\star}{200(C^\star - 1)N}}, \frac{1}{3}\right)$$

yields the lower bound

$$\inf_{\hat{\pi}} \sup_{(\rho_0, \mu_0, R)|R \in \mathcal{R}} \mathbb{E}[\mathcal{L}(\pi; (\rho_0, \mu_0, R))] \gtrsim \min\left(C^\star - 1, \sqrt{\frac{SC^\star(C^\star - 1)}{N}}\right) \gtrsim \min\left(C^\star - 1, \sqrt{\frac{S(C^\star - 1)}{N}}\right).$$

**Putting the pieces together.** We are now in position to summarize and simplify the three established lower bounds (21), (22), and (23).

When $C^\star = 1$, the claim in Theorem **??** is identical to the bound (21).

When $C^\star \geq 2$, we have from the bound (22) that

$$\inf_{\hat{\pi}} \sup_{(\rho,\mu,R)\in\mathsf{CB}(C^\star)} \mathbb{E}[\mathcal{L}(\pi;(\rho,\mu,R))] \gtrsim \min\left(1,\sqrt{\frac{SC^\star}{N}}\right) \asymp \min\left(1,\sqrt{\frac{S(C^\star-1)}{N}}\right).$$

Further notice that

$$\sqrt{\frac{S(C^\star-1)}{N}} \geq \sqrt{\frac{S}{N}} \geq \min\left(1,\frac{S}{N}\right).$$

The claimed lower bound in Theorem **??** arises.

In the end, when $C^\star \in (1,2)$, we know from the bounds (21) and (23) that

$$\inf_{\hat{\pi}} \sup_{(\rho,\mu,R)\in\mathsf{CB}(C^\star)} \mathbb{E}[\mathcal{L}(\pi;(\rho,\mu,R))] \gtrsim \max\left\{\min\left(1,\frac{S}{N}\right), \min\left(C^\star-1,\sqrt{\frac{S(C^\star-1)}{N}}\right)\right\}.$$

Elementary calculations reveal that

$$\max\left\{\min\left(1,\frac{S}{N}\right), \min\left(C^\star-1,\sqrt{\frac{S(C^\star-1)}{N}}\right)\right\} \asymp \min\left(1,\sqrt{\frac{S(C^\star-1)}{N}}+\frac{S}{N}\right),$$

which completes the proof.

### C.3 Proof of Proposition 3

We design the hard instance with state space $\{s_0, s_1\}$ and action space $\{a_0, a_1\}$. Only under state $(s_0, a_0)$ we can possibly get non-zero reward, and all other state-action pairs give 0 rewards. We set $d^\star(s_0) = d^\star(s_0, a_0) = C^\star - 1 - \epsilon$, $d^\star(s_1) = 2 - C^\star + \epsilon$ for some small $\epsilon > 0$. The constraints introduced by concentrability are $\mu(s_0, a_0) \geq (C^\star - 1 - \epsilon)/C^\star$, $\mu(s_1) \geq (2 - C^\star + \epsilon)/C^\star$.

We set $\mu(s_0, a_0) = (C^\star - 1 - \epsilon)/C^\star$, $\mu(s_0, a_1) = (C^\star - 1)/C^\star$, $\mu(s_1) = (2 - C^\star + \epsilon)/C^\star$. One can verify that $d^\star, \mu$ are valid probability distributions and the concentrability assumption still holds.

In this case, since $\mu(s_0, a_0) < \mu(s_0, a_1)$, the algorithm fails to identify the optimal arm $a_0$ as $N \to \infty$. This incurs the following expected sub-optimality

$$\lim_{N\to\infty} \mathbb{E}_{\mathcal{D}}[J(\pi^\star) - J(\hat{\pi})] = d^\star(s_0) = C^\star - 1 - \epsilon.$$

Setting $\epsilon \to 0$ gives us the conclusion.

## D  Proofs for MDPs

We begin by presenting additional notation on discounted MDPs including some convenient matrix notation as well as Bellman and Bellman-like equations. We then prove certain properties for Algorithm 2, namely pessimism and contraction results (Proposition 1) and a value difference telescoping result (cf. Lemma 4). Next, we prove the LCB sub-optimality Theorem 6. In the end, we prove the minimax lower bound followed by an analysis of imitation learning with an alternative data coverage assumption.

## D.1 Additional notation

**Matrix notation.** We present the algorithm and results in this section with the help of some matrix notation for MDPs. For a function $f : \mathcal{X} \mapsto \mathbb{R}$, we overload the notation and write $f \in \mathbb{R}^{|\mathcal{S}|}$ to denote a vector with elements $f(x)$, e.g., $V, Q$, and $r$. We write $P \in \mathbb{R}^{S|\mathcal{A}| \times S}$ to represent the probability transition matrix whose $(s, a)$-th row denoted by $P_{s,a}$ is a probability vector representing $P(\cdot \mid s, a)$. We use $P^\pi \in \mathbb{R}^{S|\mathcal{A}| \times S|\mathcal{A}|}$ to denote a transtion matrix induced by policy $\pi$ whose $(s, a) \times (s', a')$ element is equal to $P(s'|s, a)\pi(a'|s')$. We write $\rho^\pi \in \mathbb{R}^{S|\mathcal{A}|}$ to denote the initial distribution induced by policy $\pi$ whose $(s, a)$ element is equal to $\rho(s)\pi(a|s)$. For two $n$-dimensional vectors $x$ and $y$, we use $x \cdot y = x^\top y$ to denote their inner product and $x \leq y$ to denote an element-wise inequality $x_i \leq y_i$ for all $i \in \{1, \ldots, n\}$.

**Bellman and Bellman-like equations.** Given a discounted MDP, the Bellman value operator $\mathcal{T}_\pi$ associated with a policy $\pi$ is defined as

$$\mathcal{T}_\pi V := r_\pi + \gamma P_\pi V. \tag{24}$$

It is well-known that $V^\pi$ is the unique solution to $\mathcal{T}_\pi V = V$, which is known as the Bellman equation.
[PR: Define $d_\pi$] In addition to $V^\pi$, other quantities in an MDP also follow a Bellman-like equation, which we briefly review here. For discounted occupancy measures, simple algebra gives

$$d_\pi = (1 - \gamma)\rho + \gamma d_\pi P_\pi \quad \Rightarrow \quad d_\pi = (1 - \gamma)\rho(I - \gamma P_\pi)^{-1}, \tag{25}$$

$$d^\pi = (1 - \gamma)\rho^\pi + \gamma d^\pi P^\pi \quad \Rightarrow \quad d^\pi = (1 - \gamma)\rho^\pi(I - \gamma P^\pi)^{-1}. \tag{26}$$

## D.2 Proof architecture

For the general case of $C^\star \geq 1$, we first define the clean event of interest as below.

$$\mathcal{E}_{\mathrm{MDP}} := \left\{ \forall s, a, t : \left| r(s, a) - r_t(s, a) + \gamma \left( P_{s,a} - P_{s,a}^t \right) \cdot V_{t-1} \right| \leq b_t(s, a) \right\}. \tag{27}$$

In words, on the event $\mathcal{E}_{\mathrm{MDP}}$, the penalty function $b_t(s, a)$ well captures the statistical fluctuations of the Q-function estimate $r_t(s, a) + \gamma P_{s,a}^t \cdot V_{t-1}$. The following lemma shows that this event happens with high probability. The proof is postponed to Appendix D.3.

**Lemma 3** (Clean event probability, MDP). *One has $\mathbb{P}(\mathcal{E}_{MDP}) \geq 1 - \delta$.*

In the above lemma, concentration of $V_t$ is only needed instead of any value function $V$ such as required in the work Yu et al. (2020). For the latter to hold, one needs to introduce another factor of $\sqrt{S}$ by taking a union bound. We avoid a union bound by exploiting the independence of $P_{s,a}^t$ and $V_t$ obtained by randomly splitting the dataset. This is key to obtaining an optimal dependency on the state size $S$.

Under the clean event, we can show that the monotonically increasing value function $V_t$ always lower bounds the value of the corresponding policy $\pi_t$, along with a recursive inequality on the sub-optimality of $Q_{t+1}$ w.r.t. $Q^\star$ to penalty and sub-optimality of the previous step.

**Proposition 1** (Contraction properties of Algorithm 2). *Let $\pi$ be an arbitrary policy. On the event $\mathcal{E}_{MDP}$, one has for all $s \in \mathcal{S}, a \in \mathcal{A}$, and $t \in \{1, \ldots, T\}$:*

$$V_{t-1} \leq V_t \leq V^{\pi_t} \leq V^\star, \quad Q_t \leq r + \gamma P V_{t-1}, \quad \text{and} \quad Q^\pi - Q_t \leq \gamma P^\pi(Q^\pi - Q_{t-1}) + 2b_t.$$

By recursively applying the last inequality, we can derive a value difference lemma. The following lemma relates the sub-optimality to the penalty term $b_t$, of which we have good control:

**Lemma 4** (Value difference for Algorithm 2). *Let $\pi$ be an arbitrary policy. On the event $\mathcal{E}_{MDP}$, one has for all $t \in \{1, \ldots, T\}$*

$$J(\pi) - J(\pi_t) \leq \frac{\gamma^t}{1-\gamma} + 2 \sum_{i=1}^{t} \mathbb{E}_{\nu_{t-i}^\pi}[b_i(s,a)].$$

*Here, $\nu_k^\pi := \gamma^k \rho^\pi (P^\pi)^k$ for $k \geq 0$.*

The proof is provided in Appendix D.5. The value difference bound has two terms: the first term is due to convergence error of value iteration and the second term is the error caused by subtracting penalties $b_i(s,a)$ in each iteration $i$ from the rewards. By plugging in $b_i$ and choosing $t$ appropriately we can get the desired performance guarantee.

## D.3   Proof of Lemma 3

The proof is similar to that of Lemma 1. For completeness, we include it here.

From the algorithmic design, it is clear (in particular the $Q$ update and the monotonic improvement step) that

$$V_t(s) \in [0, V_{\max}], \qquad \text{for all } s \in \mathcal{S} \text{ and } t \geq 0.$$

As a result, for a fixed tuple $(s, a, t)$, if $m_t(s, a) = 0$, one has

$$\left| r(s,a) + \gamma P_{s,a} \cdot V_t - r_t(s,a) - \gamma P_{s,a}^t \cdot V_{t-1} \right| \leq 1 + \gamma V_{\max} = V_{\max} \leq b_t(s,a).$$

When $m_t(s, a) \geq 1$, exploiting the independence between $V_t$ and $P_{s,a}^t$ and using Hoeffding's inequality to obtain

$$\mathbb{P}\left( \left| r(s,a) + \gamma P_{s,a} \cdot V_t - r_t(s,a) - \gamma P_{s,a}^t \cdot V_{t-1} \right| \geq V_{\max} \sqrt{L/m_t(s,a)} \mid m_t(s,a) \right) \leq 2 \exp\left(-2L\right).$$

Since the above inequality holds for any $m_t(s, a)$, one necessarily has

$$\mathbb{P}\left( \left| r(s,a) + \gamma P_{s,a} \cdot V_t - r_t(s,a) - \gamma P_{s,a}^t \cdot V_{t-1} \right| \geq b_t(s,a) \right) \leq 2 \exp\left(-2L\right).$$

Taking a union bound over $s, a$ and $t \in \{0, \ldots, T\}$ and setting $\delta_1 = \frac{\delta}{2S|\mathcal{A}|(T+1)}$ finishes the proof.

## D.4   Proof of Proposition 1

We prove the claims one by one.

**Proof of $V_{t-1} \leq V_t$.**   The first claim $V_{t-1} \leq V_t$ is directly implied by the monotonic improvement in Algorithm 2.

**Proof of $V_t \leq V^{\pi_t}$.**   For the second claim $V_t \leq V^{\pi_t}$, it suffices to prove that $V_t \leq \mathcal{T}_{\pi_t} V_t$. Indeed, $V_t \leq \mathcal{T}_{\pi_t} V_t$ together with the monotonicity of the Bellman's operator yield the conclusion $V_t \leq V^{\pi_t}$. In what follows, we prove $V_t \leq \mathcal{T}_{\pi_t} V_t$ via induction.

The base case $V_0 \leq \mathcal{T}_{\pi_0} V_0$ holds due to zero initialization. Hence from now on, we assume $V_k \leq \mathcal{T}_{\pi_k} V_k$ for $0 \leq k \leq t-1$ and intend to prove $V_t \leq \mathcal{T}_{\pi_t} V_t$. We split the proof into two cases.

- If $V_{t-1}(s) \geq \max_a \{r_{t-1}(s,a) - b_{t-1}(s,a) + \gamma P^{t-1}_{s,a} \cdot V_{t-1}\}$, the algorithm sets $V_t(s) = V_{t-1}(s)$ and $\pi_t(s) = \pi_{t-1}(s)$. Consequently, we have

$$V_t(s) = V_{t-1}(s) \leq (\mathcal{T}_{\pi_{t-1}} V_{t-1})(s) \leq (\mathcal{T}_{\pi_t} V_t)(s),$$

where the first inequality arises from the induction hypothesis and the last one holds since $V_{t-1} \leq V_t$ and $\pi_t(s) = \pi_{t-1}(s)$.

- If instead, the algorithm sets $Q_t(s,a) = r_t(s,a) - b_t(s,a) + \gamma P^t_{s,a} \cdot V_{t-1}$ with $\pi_t(s) = \arg\max_a Q_t(s,a)$ and $V_t(s) = Q_t(s, \pi_t(s))$, then we have

$$\begin{aligned}
(\mathcal{T}_{\pi_t} V_t)(s) =& r(s, \pi_t(s)) + \gamma P_{s, \pi_t(s)} \cdot V_t \\
\geq& r(s, \pi_t(s)) + \gamma P_{s, \pi_t(s)} \cdot V_{t-1} \\
=& r_t(s, \pi_t(s)) - b_t(s, \pi_t(s)) + \gamma P^t_{s, \pi_t(s)} \cdot V_{t-1} \\
& + b_t(s, \pi_t(s)) + r(s, \pi_t(s)) - r_t(s, \pi_t(s)) + \gamma (P_{s, \pi_t(s)} - P^t_{s, \pi_t(s)}) \cdot V_{t-1} \\
=& V_t(s) + b_t(s, \pi_t(s)) + r(s, \pi_t(s)) - r_t(s, \pi_t(s)) + \gamma (P_{s, \pi_t(s)} - P^t_{s, \pi_t(s)}) \cdot V_{t-1} \\
\geq& V_t(s),
\end{aligned}$$

where the first inequality is due to $V_{t-1} \leq V_t$ and the last inequality holds under the clean event $\mathcal{E}_{\mathrm{MDP}}$.

This finishes the proof of $V_t \leq \mathcal{T}_{\pi_t} V_t$ and hence $V_t \leq V^{\pi_t}$. The claim $V^{\pi_t} \leq V^\star$ is trivial to see.

**Proof of $Q_t \leq r + \gamma P V_{t-1} \leq r + \gamma P V_t$.** Since $V_t \geq V_{t-1}$, we have

$$\begin{aligned}
r(s,a) + \gamma P_{s,a} \cdot V_t \geq\ & r(s,a) + \gamma P_{s,a} \cdot V_{t-1} \\
=\ & r_t(s,a) - b_t(s,a) + \gamma P^t_{s,a} \cdot V_{t-1} \\
& + b_t(s,a) + r(s,a) - r_t(s,a) + \gamma (P_{s,a} - P^t_{s,a}) \cdot V_{t-1} \\
\geq\ & Q_t(s,a),
\end{aligned}$$

where the last inequality holds under $\mathcal{E}_{\mathrm{MDP}}$.

**Proof of $Q^\pi - Q_t \leq \gamma P^\pi (Q^\pi - Q_{t-1}) + 2b_t$.** Let $Q(:, \pi) \in \mathbb{R}^S$ be a vector with elements $Q^\pi(s, \pi(s))$. By definition, one has

$$\begin{aligned}
& Q^\pi(s,a) - Q_t(s,a) \\
& = r(s,a) + \gamma P_{s,a} \cdot V^\pi - r_t(s,a) + b_t(s,a) - \gamma P^t_{s,a} \cdot V_{t-1} \\
& = \gamma P_{s,a} \cdot V^\pi - \gamma P_{s,a} \cdot V_{t-1} + b_t(s,a) + r(s,a) - r_t(s,a) + \gamma (P_{s,a} - P^t_{s,a}) \cdot V_{t-1} \\
& \leq \gamma P_{s,a} \cdot (Q^\pi(:, \pi) - Q_{t-1}(:, \pi)) + b_t(s,a) + r(s,a) - r_t(s,a) + \gamma (P_{s,a} - P^t_{s,a}) \cdot V_{t-1} \\
& \leq \gamma P_{s,a} \cdot (Q^\pi(:, \pi) - Q_{t-1}(:, \pi)) + 2b_t(s,a).
\end{aligned}$$

Here, the first inequality comes from the fact that $V_{t-1} \geq \max_a Q_{t-1}(:, a) \geq Q_t(:, \pi)$ and the last inequality again holds under $\mathcal{E}_{\mathrm{MDP}}$.

## D.5 Proof of Lemma 4

In view of Proposition 1, one has $V_t \leq V^{\pi_t}$. Therefore we obtain

$$\mathbb{E}_\rho \left[ V^\pi(s) - V^{\pi_t}(s) \right] \leq \mathbb{E}_\rho \left[ V^\pi(s) - V_t(s) \right] \leq \mathbb{E}_\rho \left[ V^\pi(s) - V_t^{\text{mid}}(s) \} \right],$$

where the last inequality arises from the monotonicity imposed by Algorithm 2. Note that $V_t^{\text{mid}}(s) = Q_t(s, \pi_t^{\text{mid}}(s))$ and that $\pi_t^{\text{mid}}$ is greedy with respect to $Q_t$. We can continue the upper bound as

$$\mathbb{E}_\rho \left[ V^\pi(s) - V^{\pi_t}(s) \right] \leq \mathbb{E}_\rho \left[ Q^\pi(s, \pi(s)) - Q_t(s, \pi_t^{\text{mid}}) \} \right] \leq \mathbb{E}_\rho \left[ Q^\pi(s, \pi(s)) - Q_t(s, \pi(s)) \right].$$

Rewriting using the matrix notation gives

$$\mathbb{E}_\rho \left[ V^\pi(s) - V^{\pi_t}(s) \right] \leq \mathbb{E}_\rho \left[ Q^\pi(s, \pi(s)) - Q_t(s, \pi(s)) \right] = \rho^\pi (Q^\pi - Q_t). \tag{28}$$

Now we are ready to apply the third claim in Proposition 1 to deduce that on the event $\mathcal{E}_{\text{MDP}}$:

$$Q^\pi - Q_t \leq \gamma P^\pi (Q^\pi - Q_{t-1}) + 2b_t \leq \gamma P^\pi \left[ \gamma P^\pi (Q^\pi - Q_{t-2}) + 2b_{t-1} \right] + 2b_t$$

$$\leq \cdots$$

$$\leq \gamma^t (P^\pi)^t (Q^\pi - Q_0) + 2 \sum_{j=1}^t (\gamma P^\pi)^{t-j} b_j$$

$$\leq \frac{\gamma^t}{1-\gamma} \mathbf{1} + 2 \sum_{j=1}^t (\gamma P^\pi)^{t-j} b_j.$$

Here $\mathbf{1}$ denotes the all-one vector with dimension $S|\mathcal{A}|$, and the last inequality arises from the fact that $Q^\pi - Q_0 = Q^\pi \leq (1-\gamma)^{-1} \mathbf{1}$. Multiplying both sides the of the equation above by $\rho^\pi$, we conclude that

$$\rho^\pi (Q^\pi - Q_t) \leq \frac{\gamma^t}{1-\gamma} + 2 \sum_{j=1}^t \rho^\pi (\gamma P^\pi)^{t-j} b_j = \frac{\gamma^t}{1-\gamma} + 2 \sum_{j=1}^t v_{t-j}^\pi b_j, \tag{29}$$

where we use the definition of $v_k^\pi = \rho^\pi (\gamma P^\pi)^k$. Combine the inequalities (28) and (29) to reach the desired result.

## D.6 Proof of Theorem 6

Similar to the proof given for contextual bandits, we prove a stronger result than Theorem 6. Fix any deterministic expert policy $\pi$. Assume that the data coverage assumption holds, that is

$$\max_{s,a} \frac{d^\pi(s,a)}{\mu(s,a)} \leq C^\pi.$$

Then for all $C^\pi \geq 1$, Algorithm 2 with $\delta = 1/N$ achieves

$$\mathbb{E}_\mathcal{D} \left[ J(\pi) - J(\hat{\pi}) \right] \lesssim \min \left( \frac{1}{1-\gamma}, \widetilde{O} \left( \sqrt{\frac{SC^\pi}{(1-\gamma)^5 N}} \right) \right). \tag{30}$$

In addition, if $1 \leq C^\pi \leq 1 + \frac{L \log(N)}{200(1-\gamma)N}$, then we have a tighter performance upper bound

$$\mathbb{E}_\mathcal{D} \left[ J(\pi) - J(\hat{\pi}) \right] \lesssim \min \left( \frac{1}{1-\gamma}, \widetilde{O} \left( \frac{S}{(1-\gamma)^4 N} \right) \right). \tag{31}$$

The result in Theorem 6 can be recovered by taking $\pi = \pi^\star$.

We split the proof into two cases: (1) the general case when $C^\pi \geq 1$ and (2) the regime where $C^\pi \leq 1 + L/(200m)$.

**The general case when $C^\pi \geq 1$.** The proof of the general case follows similar steps as those in the proof of Theorem 4. We first decompose the expected sub-optimality into three terms:

$$\mathbb{E}_\mathcal{D}\left[\sum_s \rho(s)[V^\pi(s) - V^{\pi_T}(s)]\right]$$

$$= \mathbb{E}_\mathcal{D}\left[\sum_s \rho(s)[V^\pi(s) - V^{\pi_T}(s)] \, \mathbb{1}\{\exists t \leq T, m_t(s, \pi(s)) = 0\}\right] =: T_1$$

$$+ \mathbb{E}_\mathcal{D}\left[\sum_s \rho(s)[V^\pi(s) - V^{\pi_T}(s)] \, \mathbb{1}\{\forall t \leq T, m_t(s, \pi(s)) \geq 1\} \, \mathbb{1}\{\mathcal{E}_{\mathrm{MDP}}\}\right] =: T_2$$

$$+ \mathbb{E}_\mathcal{D}\left[\sum_s \rho(s)[V^\pi(s) - V^{\pi_T}(s)] \, \mathbb{1}\{\forall t \leq T, m_t(s, \pi(s)) \geq 1\} \, \mathbb{1}\{\mathcal{E}_{\mathrm{MDP}}^c\}\right] =: T_3.$$

Similar to before, the first term $T_1$ captures the sub-optimality incurred by the missing mass on the expert action $\pi(s)$. The second term $T_2$ is the sub-optimality under the clean event $\mathcal{E}_{\mathrm{MDP}}$, while the last one $T_3$ denotes the sub-optimality suffered under the complement event $\mathcal{E}_{\mathrm{MDP}}^c$, on which the empirical average of Q-function falls outside the constructed confidence interval.

As we will show in subsequent sections, these error terms satisfy the following upper bounds:

$$T_1 \leq \frac{4SC^\pi(T+1)^2}{9(1-\gamma)^2 N}; \tag{32a}$$

$$T_2 \leq \frac{\gamma^T}{1-\gamma} + 32\frac{1}{(1-\gamma)^2}\sqrt{\frac{LSC^\pi(T+1)}{N}}; \tag{32b}$$

$$T_3 \leq V_{\max}\delta. \tag{32c}$$

Setting $\delta = 1/N$, $T = \log N/(1-\gamma)$ and noting that $\gamma^T \leq 1/N$ yield that

$$\mathbb{E}_\mathcal{D}\left[J(\pi) - J(\hat{\pi})\right] \lesssim \left(\sqrt{\frac{SC^\pi}{(1-\gamma)^5 N}} + \frac{SC^\pi}{(1-\gamma)^4 N}\right).$$

Note that we always have $\mathbb{E}_\mathcal{D}\left[J(\pi) - J(\hat{\pi})\right] \leq \frac{1}{1-\gamma}$. In the interesting regime of $\frac{SC^\pi}{(1-\gamma)^3 N} \leq 1$, the first term above always dominates. This gives the desired claim (30).

**The case when $C^\pi \leq 1 + L/(200m)$.** Under this circumstance, the following lemma proves useful.

**Lemma 5.** *For any deterministic policy $\hat{\pi}$, one has*

$$J(\pi) - J(\hat{\pi}) \leq V_{\max}^2 \mathbb{E}_{s \sim d_\pi}\left[\mathbb{1}\{\hat{\pi}(s) \neq \pi(s)\}\right]. \tag{33}$$

*Proof.* In view of the performance difference lemma in Kakade and Langford (2002, Lemma 6.1), one has

$$J(\pi) - J(\hat{\pi}) = \frac{1}{1-\gamma}\mathbb{E}_{s \sim d_\pi}\left[Q^{\hat{\pi}}(s, \pi(s)) - Q^{\hat{\pi}}(s, \hat{\pi}(s))\right]$$

$$= \frac{1}{1-\gamma}\mathbb{E}_{s \sim d_\pi}\left[\left[Q^{\hat{\pi}}(s, \pi(s)) - Q^{\hat{\pi}}(s, \hat{\pi}(s))\right]\mathbb{1}\{\hat{\pi}(s) \neq \pi(s)\}\right]$$

$$\leq V_{\max}^2 \mathbb{E}_{s \sim d_\pi}\left[\mathbb{1}\{\hat{\pi}(s) \neq \pi(s)\}\right].$$

Here the last line uses the fact that $Q^{\hat{\pi}}(s, \pi(s)) - Q^{\hat{\pi}}(s, \hat{\pi}(s)) \leq V_{\max}$. □

Lemma 5 links the sub-optimality of a policy to its disagreement with the optimal policy. With Lemma 5 at hand, we can continue to decompose the expected sub-optimality into:

$$\mathbb{E}_{\mathcal{D}}\left[\sum_s \rho(s)[V^\pi(s) - V^{\pi_T}(s)]\right]$$

$$\leq V_{\max}^2\, \mathbb{E}_{\mathcal{D}}[\mathbb{E}_{s\sim d_\pi}[\mathbb{1}\{\pi_T(s) \neq \pi(s)\}]]$$

$$= V_{\max}^2\, \mathbb{E}_{\mathcal{D}}[\mathbb{E}_{s\sim d_\pi}[[\mathbb{1}\{\pi_T(s) \neq \pi(s)\}\,\mathbb{1}\{\exists t \leq T, m_t(s, \pi(s)) = 0\}]] =: T_1'$$

$$\quad + V_{\max}^2\, \mathbb{E}_{\mathcal{D}}[\mathbb{E}_{s\sim d_\pi}[[\mathbb{1}\{\pi_T(s) \neq \pi(s)\}\,\mathbb{1}\{\forall t \leq T, m_t(s, \pi(s)) \geq 1\}]] =: T_2'$$

We bound each term according to

$$T_1' \leq \frac{4SC^\pi(T+1)^2}{9(1-\gamma)^2 N}; \tag{34a}$$

$$T_2' \lesssim \frac{SC^\pi LT}{(1-\gamma)^2 N} + \frac{ST^{10}}{(1-\gamma)^2 N^9}. \tag{34b}$$

The claimed bound (31) follows by taking $\delta = 1/N$ and $T = \log N/(1-\gamma)$.

### D.6.1  Proof of the bound (32a) on $T_1$ and the bound (34a) on $T_1'$

Since for any $s \in \mathcal{S}$, $V^\pi(s) - V^{\pi_T}(s) \leq V_{\max}$ one has

$$T_1 \leq V_{\max}\, \mathbb{E}_{\mathcal{D}}\left[\sum_s \rho(s)\,\mathbb{1}\{\exists t \leq T, m_t(s, \pi(s)) = 0\}\right] = V_{\max}\sum_s \rho(s)\mathbb{P}\left(\exists t \leq T, m_t(s, \pi(s)) = 0\right).$$

The definition of the normalized occupancy measure (??) entails $\rho(s) \leq d^\pi(s, \pi(s))$ and thus

$$\frac{\rho(s)}{\mu(s, \pi(s))} \leq \frac{1}{1-\gamma} \cdot \frac{d^\pi(s, \pi(s))}{\mu(s, \pi(s))} \leq \frac{C^\pi}{1-\gamma}.$$

Here the last relation follows from the data coverage assumption. Combine the above two inequalities to see that

$$T_1 \leq V_{\max}\sum_s \frac{C^\pi}{1-\gamma}\mu(s, \pi(s))\mathbb{P}\left(\exists t \leq T, m_t(s, \pi(s)) = 0\right)$$

$$= \frac{C^\pi}{(1-\gamma)^2}\sum_s \mu(s, \pi(s))\,\mathbb{P}\left(\exists t \leq T, m_t(s, \pi(s)) = 0\right)$$

$$\leq \frac{C^\pi}{(1-\gamma)^2}\sum_{t=0}^T\sum_s \mu(s, \pi(s))\,\mathbb{P}\left(m_t(s, \pi(s)) = 0\right),$$

where in the penultimate line, we identify $V_{\max}$ with $1/(1-\gamma)$, and the last relation is by the union bound. Direct calculations yield

$$\mathbb{P}\left(m_t(s, \pi(s)) = 0\right) = (1 - \mu(s, \pi(s)))^m,$$

which further implies

$$T_1 \leq \frac{C^\pi(T+1)}{(1-\gamma)^2}\sum_s \mu(s, \pi(s))(1 - \mu(s, \pi(s)))^m \leq \frac{4C^\pi S(T+1)}{9(1-\gamma)^2 m} = \frac{4C^\pi S(T+1)^2}{9(1-\gamma)^2 N}.$$

Here, we have used $\max_{x\in[0,1]} x(1-x)^m \leq 4/(9m)$ and the fact that $m = N/(T+1)$.

The bound (34a) on $T_1'$ follows from exactly the same argument as above, except that we replace $\rho$ with $d^\pi$.

### D.6.2 Proof of the bound (32b) on $T_2$

Lemma 4 asserts that on the clean event $\mathcal{E}_{\mathrm{MDP}}$, one has

$$
\begin{aligned}
T_2 &\leq \frac{\gamma^T}{1-\gamma} + 2\sum_{t=1}^T \mathbb{E}_{\mathcal{D}, \nu_{T-t}^\pi} \left[ b_t(s, \pi(s)) \, \mathbb{1}\{m_t(s, \pi(s)) \geq 1\} \right] \\
&= \frac{\gamma^T}{1-\gamma} + 2\sum_{t=1}^T \mathbb{E}_{\mathcal{D}, \nu_{T-t}^\pi} \left[ V_{\max} \sqrt{\frac{L}{m_t(s, \pi(s))}} \, \mathbb{1}\{m_t(s, \pi(s)) \geq 1\} \right] \\
&\leq \frac{\gamma^T}{1-\gamma} + 2\sum_{t=1}^T \mathbb{E}_{\nu_{T-t}^\pi} \left[ 16 V_{\max} \sqrt{\frac{L}{m\mu(s, \pi(s))}} \right].
\end{aligned}
\tag{35}
$$

Here, we substitute in the definition of $b_t(s, a)$ in the middle line and the last inequality arises from Lemma 14 with $c_{1/2} \leq 16$.

By definition of $\nu_k^\pi = \rho^\pi (\gamma P^\pi)^k$, we have $\sum_{k=0}^\infty \nu_k^\pi = d^\pi/(1-\gamma)$. Therefore, one has

$$
\begin{aligned}
\sum_{t=1}^T \mathbb{E}_{\nu_{T-t}^\pi} \left[ \frac{1}{\sqrt{\mu(s, \pi(s))}} \right] &= \sum_{t=1}^T \sum_s \nu_{T-t}^\pi(s, \pi(s)) \frac{1}{\sqrt{\mu(s, \pi(s))}} \\
&= \sum_s \left[ \sum_{t=1}^T \nu_{T-t}^\pi(s, \pi(s)) \right] \frac{1}{\sqrt{\mu(s, \pi(s))}} \\
&\leq \sum_s \frac{d^\pi(s, \pi(s))}{1-\gamma} \frac{1}{\sqrt{\mu(s, \pi(s))}}.
\end{aligned}
$$

We then apply the concentrability assumption and the Cauchy–Schwarz inequality to deduce that

$$
\begin{aligned}
\sum_{t=1}^T \mathbb{E}_{\nu_{T-t}^\pi} \left[ \frac{1}{\sqrt{\mu(s, \pi(s))}} \right] &\leq \sqrt{\frac{C^\pi}{(1-\gamma)^2}} \sum_s \sqrt{d^\pi(s, \pi(s))} \\
&\leq \sqrt{\frac{C^\pi}{(1-\gamma)^2}} \sqrt{S} \sqrt{\sum_s d^\pi(s, \pi(s))} \\
&= \frac{\sqrt{SC^\pi}}{1-\gamma}.
\end{aligned}
$$

Substitute the above bound into the inequality (35) to arrive at the conclusion

$$
T_2 \leq \frac{\gamma^T}{1-\gamma} + 32 \frac{1}{(1-\gamma)^2} \sqrt{\frac{LSC^\pi}{m}}.
$$

The proof is completed by noting that $m = N/(T+1)$.

### D.6.3 Proof of the bound (32c) on $T_3$

It is easy to see that

$$
\sum_s \rho(s)[V^\pi(s) - V^{\pi_T}(s)] \, \mathbb{1}\{\forall s, t, m_t(s, \pi(s)) \geq 1\} \leq V_{\max},
$$

which further implies

$$
T_3 \leq V_{\max} \, \mathbb{E}_{\mathcal{D}}[\mathbb{1}\{\mathcal{E}_{\mathrm{MDP}}^c\}] = V_{\max} \, \mathbb{P}(\mathcal{E}_{\mathrm{MDP}}^c) \leq V_{\max} \delta.
$$

Here, the last bound relies on Lemma 3.

### D.6.4 Proof of the bound (34b) on $T_2'$

Partition the state space into the following two disjoint sets:

$$\mathcal{S}_1 := \left\{ s \mid d_\pi(s) < \frac{2C^\pi L}{m} \right\}, \tag{36a}$$

$$\mathcal{S}_2 := \left\{ s \mid d_\pi(s) \geq \frac{2C^\pi L}{m} \right\}, \tag{36b}$$

In words, the set $\mathcal{S}_1$ includes the states that are less important in evaluating the performance of LCB. We can then decompose the term $T_2'$ accordingly:

$$T_2' = V_{\max}^2 \sum_{s \in \mathcal{S}_1} d_\pi(s) \, \mathbb{E}_{\mathcal{D}}[\mathbb{1}\{\pi_T(s) \neq \pi(s)\} \, \mathbb{1}\{\forall t, m_t(s, \pi(s)) \geq 1\}] =: T_{2,1}$$

$$+ V_{\max}^2 \sum_{s \in \mathcal{S}_2} d_\pi(s) \, \mathbb{E}_{\mathcal{D}}[\mathbb{1}\{\pi_T(s) \neq \pi(s)\} \, \mathbb{1}\{\forall t, m_t(s, \pi(s)) \geq 1\}] =: T_{2,2}.$$

The proof is completed by observing the following two upper bounds:

$$T_{2,1} \leq \frac{2SC^\pi LT}{(1-\gamma)^2 N}, \qquad \text{and} \qquad T_{2,2} \lesssim \frac{S}{(1-\gamma)^2} \left(\frac{T}{N}\right)^9.$$

**Proof of the bound on $T_{2,1}$.** We again use the basic fact that

$$\mathbb{E}_{\mathcal{D}}[\mathbb{1}\{\pi_T(s) \neq \pi(s)\} \, \mathbb{1}\{\forall s, t, m_t(s, \pi(s)) \geq 1\}] \leq 1$$

to reach

$$T_{2,1} \leq V_{\max}^2 \sum_{s \in \mathcal{S}_1} d_\pi(s) \leq \frac{2SC^\pi L}{(1-\gamma)^2 m},$$

where the last inequality hinges on the definition of $\mathcal{S}_1$ given in (36a), namely for any $s \in \mathcal{S}_1$, one has $d_\pi(s) < \frac{2C^\pi L}{m}$. Identifying $m$ with $N/(T+1)$ concludes the proof.

**Proof of the bound on $T_{2,2}$.** Equivalently, we can write $T_{2,2}$ as

$$T_{2,2} = V_{\max}^2 \sum_{s \in \mathcal{S}_2} d_\pi(s) \mathbb{P}\left(\pi_T(s) \neq \pi(s), \; m_t(s, \pi(s)) \geq 1 \; \forall t\right).$$

By inspecting Algorithm 2, one can realize the following inclusion

$$\{\pi_T(s) \neq \pi(s)\} \subseteq \{\pi_0(s) \neq \pi(s)\} \cup \{\exists 0 \leq t \leq T-1 \text{ and } \exists a \neq \pi(s), Q_{t+1}(s, a) \geq Q_{t+1}(s, \pi(s))\}.$$

Indeed, if $\pi_0(s) = \pi(s)$ and for all $t$, $Q_{t+1}(s, \pi(s)) > \max_{a \neq \pi(s)} Q_{t+1}(s, a)$, LCB would select the expert action in the end, i.e., $\pi_T(s) = \pi(s)$. Therefore, we can upper bound $T_{2,2}$ as

$$T_{2,2} \leq V_{\max}^2 \sum_{s \in \mathcal{S}_2} d_\pi(s) \mathbb{P}\left(\pi_0(s) \neq \pi(s), m_t(s, \pi(s)) \geq 1 \; \forall t\right) =: \beta_1$$

$$+ V_{\max}^2 \sum_{s \in \mathcal{S}_2} d_\pi(s) \mathbb{P}\left(\exists t \leq T-1, \exists a \neq \pi(s), Q_{t+1}(s, a) \geq Q_{t+1}(s, \pi(s)), m_t(s, \pi(s)) \geq 1 \; \forall t\right) =: \beta_2.$$

In the sequel, we bound $\beta_1$ and $\beta_2$ in the reverse order.

**Bounding $\beta_2$.** Fix a state $s \in \mathcal{S}_2$. In view of the data coverage assumption, one has

$$\mu(s, \pi(s)) \geq \frac{1}{C^\pi} d_\pi(s) \geq \frac{1}{C^\pi} \frac{2C^\pi L}{m} = \frac{2L}{m}. \tag{37}$$

In contrast, for any $a \neq \pi(s)$, since $C^\pi \leq 1 + \frac{L}{200m}$, we have

$$\mu(s, a) \leq \sum_{a \neq \pi(s)} \mu(s, a) \leq 1 - \frac{1}{C^\pi} \leq \frac{L}{200m}, \tag{38}$$

where the middle inequality reuses the concentrability assumption. One has $\mu(s, \pi(s)) \gg \mu(s, a)$ for any non-expert action $a$. As a result, the expert action is pulled more frequently than the others. It turns out that under such circumstances, the LCB algorithm picks the expert action with high probability. We shall make this intuition precise below.

The bounds (37) and (38) together with Chernoff's bound give

$$\mathbb{P}\left(m_t(s, a) \leq \frac{5L}{200}\right) \geq 1 - \exp\left(-\frac{L}{200}\right);$$

$$\mathbb{P}\left(m_t(s, \pi(s)) \geq L\right) \geq 1 - \exp\left(-\frac{L}{4}\right).$$

These allow us to obtain an upper bound for the function $Q_{t+1}$ evaluated at non-expert actions and a lower bound on $Q_{t+1}(s, \pi(s))$. More precisely, when $m_t(s, a) \leq \frac{5L}{200}$, we have

$$Q_t(s, a) = r_t(s, a) - b_t(s, a) + \gamma P_{s,a}^t \cdot V_{t-1}$$

$$= r_t(s, a) - V_{\max} \sqrt{\frac{L}{m_t(s, a) \vee 1}} + \gamma P_{s,a}^t \cdot V_{t-1}$$

$$\leq 1 - V_{\max} \sqrt{\frac{L}{5L/200}} + \gamma V_{\max}$$

$$\leq -5V_{\max}.$$

Here we used the fact that $L \geq 70$. Now we turn to lower bounding the function $Q_t$ evaluated at the optimal action. When $m_t(s, \pi(s)) \geq L$, one has

$$Q_t(s, \pi(s)) = r_t(s, \pi(s)) - V_{\max} \sqrt{\frac{L}{m_t(s, \pi(s))}} + \gamma P_{s, \pi(s)}^t \cdot V_{t-1} \geq -V_{\max}.$$

To conclude, if both $m_t(s, a) \leq \frac{5L}{200}$ and $m_t(s, \pi(s)) \geq L$ hold, we must have $Q_t(s, a) < Q_t(s, \pi(s))$. Therefore we can deduce that

$$\mathbb{P}\left(\exists 0 \leq t \leq T \text{ and } \exists a \neq \pi(s), Q_t(s, a) \geq Q_t(s, \pi(s)), m_t(s, \pi(s)) \geq 1 \; \forall t\right)$$

$$\leq \sum_{0 \leq t \leq T} \mathbb{P}\left(\exists a \neq \pi(s), Q_t(s, a) \geq Q_t(s, \pi(s)), m_t(s, \pi(s)) \geq 1 \; \forall t\right)$$

$$\leq \sum_{0 \leq t \leq T-1} \left\{(|\mathcal{A}| - 1) \exp\left(-\frac{L}{200}\right) + \exp\left(-\frac{1}{4}L\right)\right\}$$

$$\leq T|\mathcal{A}| \exp\left(-\frac{L}{200}\right),$$

which further implies

$$\beta_2 \leq V_{\max}^2 \sum_{s \in \mathcal{S}_2} d_\pi(s) T |\mathcal{A}| \exp\left(-\frac{L}{200}\right)$$

$$\leq T V_{\max} |\mathcal{A}| \cdot \frac{1}{1-\gamma} \exp\left(-\frac{L}{200}\right)$$

$$\lesssim T m^{-9}.$$

**Bounding $\beta_1$.** In fact, the analysis of $\beta_2$ has revealed that with high probability, $\pi(s)$ is the most played arm among all actions. More precisely, we have

$$\beta_1 \leq V_{\max}^2 \sum_{s \in \mathcal{S}_2} d_\pi(s) \mathbb{P}\left(\pi_0(s) \neq \pi(s)\right)$$

$$\leq V_{\max}^2 \sum_{s \in \mathcal{S}_2} d_\pi(s) \left\{ \mathbb{P}\left(\max_a m_0(s,a) \geq \frac{5L}{200}\right) + \mathbb{P}\left(m_0(s, \pi(s)) \leq L\right) \right\}$$

$$\leq V_{\max}^2 |\mathcal{A}| \exp\left(-\frac{L}{200}\right) \lesssim \frac{1}{(1-\gamma)^2 m^{-9}}.$$

Combine the bounds on $\beta_1$ and $\beta_2$ to arrive at the claim on $T_{2,2}$.

### D.7 Proof of Theorem 7

Similar to the proof of the lower bound for contextual bandits, we split the proof into three cases: (1) $C^\star = 1$, (2) $C^\star \geq 2$, and (3) $C^\star \in (1, 2)$. For $C^\star = 1$, we adapt the lower bound from episodic imitation learning (Rajaraman et al., 2020) to the discounted case. For both $C^\star \in (1, 2)$ and $C^\star \geq 2$, we rely on the construction of the MDP in the paper Lattimore and Hutter (2012), which reduces the policy learning problem in MDP to a bandit problem. The key difference is that in our construction, we need to carefully design the initial distribution $\rho$ to incorporate the effect of $C^\star$ in the lower bound.

**The case when $C^\star = 1$.** In this case we have $\mu(s,a) = d^\star(s,a)$ for all $(s,a)$ pairs, which is the imitation learning setting. We adapt the lower bound given in Rajaraman et al. (2020) for episodic imitation learning to the discounted case and obtain the following lemma:

**Lemma 6.** *When $C^\star = 1$, one has*

$$\inf_{\hat{\pi}} \sup_{(\rho, \mu, P, R) \in \mathsf{MDP}(1)} \mathbb{E}_\mathcal{D}[J(\pi^\star) - J(\hat{\pi})] \gtrsim \min\left\{\frac{1}{1-\gamma}, \frac{S}{(1-\gamma)^2 N}\right\}. \tag{39}$$

We defer the proof to Appendix D.7.2, which follows exactly the analysis by Rajaraman et al. (2020) except for changing the setting from episodic to discounted.

**The case when $C^\star \geq 2$.** When $C^\star \geq 2$, we intend to show that

$$\inf_{\hat{\pi}} \sup_{(\rho, \mu, P, R) \in \mathsf{MDP}(C^\star)} \mathbb{E}_\mathcal{D}[J(\pi^\star) - J(\hat{\pi})] \gtrsim \min\left(\frac{1}{1-\gamma}, \sqrt{\frac{SC^\star}{(1-\gamma)^3 N}}\right). \tag{40}$$

We adopt the following construction of the hard MDP instance from the work Lattimore and Hutter (2012).

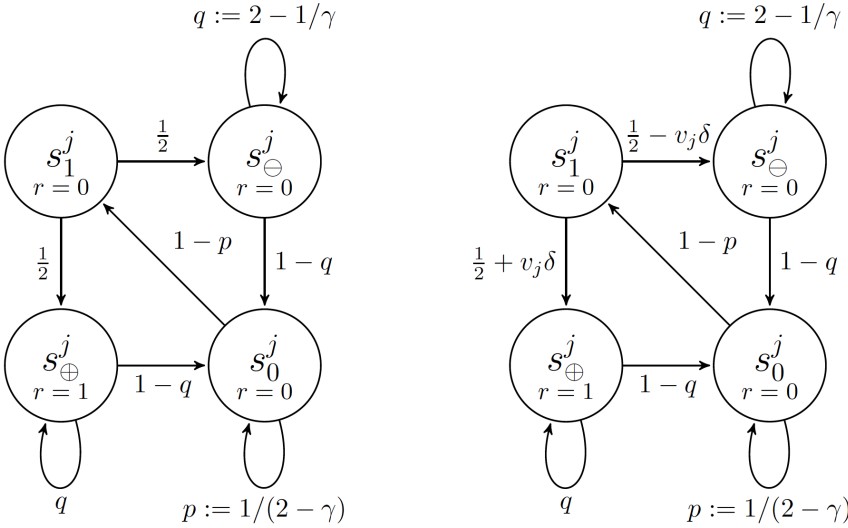

Figure 1: Illustration of one replica in the hard $\mathsf{MDP}_h$. The left plot shows the transition probabilities from $(s_1^j, a_1)$ and the right plot shows them from $(s_1^j, a_2)$.

**Construction of hard instances.** Consider the MDP which consists of $S/4$ replicas of MDPs in Figure 1 and an extra state $s_{-1}$. The total number of states is $S + 1$. For each replica, we have four states $s_0, s_1, s_\oplus, s_\ominus$. There is only one action, say $a_1$, in all the states except $s_1$, which has two actions $a_1, a_2$. The rewards are all deterministic. In addition, the transitions for states $s_0, s_\oplus, s_\ominus$ are shown in the diagram. More specifically, we have $\mathbb{P}(s_\oplus^j \mid s_1^j, a_1) = \mathbb{P}(s_\ominus^j \mid s_1^j, a_1) = 1/2$ and $\mathbb{P}(s_\oplus^j \mid s_1^j, a_2) = 1/2 + v_j \delta$, and $\mathbb{P}(s_\ominus^j \mid s_1^j, a_2) = 1/2 - v_j \delta$. Here $v_j \in \{-1, +1\}$ is the design choice associated with the $j$-th replica and $\delta \in [0, 1/4]$ will be specified later. Clearly, if $v_j = 1$, the optimal action at $s_1^j$ is $a_2$, otherwise, the optimal one is $a_1$. Under the extra state $s_{-1}$, there is only one action with reward 0 which transits to itself with probability 1. We use $s_i^j$ to denote state $i$ in $j$-th replica, where $j \in [S/4]$. Based on the description above, the only parameter in this MDP is the transition dynamics associated with the state $s_1^j$. We will later specify how to set these for each $s_1^j$. A single replica has the following important properties:

1. The probabilities $p, q$ are designed such that the three states $s_0, s_\ominus, s_\oplus$ are mostly absorbing, while any action in $s_1$ will lead to immediate transition to $s_\oplus$ or $s_\ominus$.

2. The state $s_\oplus$ is the only state that gives reward 1, which helps reduce the MDP problem to a bandit one: the MDP only depends on the choice of transition probabilities at state $s_1^j$; once a policy reaches state $s_1$ it should choose the action most likely to lead to state $\oplus$ whereupon it will either be rewarded or punished (visit state $\oplus$ or $\ominus$). Eventually, it will return to state 1 where the whole process repeats.

We also need to specify the initial distribution $\rho_0$ and the behavior distribution $\mu_0$. When $C^\star \geq 2$, we set the initial distribution $\rho_0$ to be uniformly distributed on the state $s_0$ in all the $S/4$

replicas, i.e., $\forall j \in [S/4], \rho_0(s_0^j) = 4/S$. From $d^\star = (1-\gamma)\rho(I - \gamma P_{\pi^\star})^{-1}$ we can derive $d^\star$ as follows:

$$d^\star(s_0^j) = \frac{8}{(2+\gamma)S}, \qquad d^\star(s_1^j) = \frac{8\gamma(1-\gamma)}{(2-\gamma)(2+\gamma)S} \in \left[\frac{1-\gamma}{S}, \frac{4(1-\gamma)}{S}\right],$$

$$d^\star(s_\oplus^j) = \frac{\gamma(\frac{1}{2}\mathbb{1}\{v_j = -1\} + (\frac{1}{2}+\delta)\mathbb{1}\{v_j = 1\})}{2(1-\gamma)} \cdot d^\star(s_1^j),$$

$$d^\star(s_\ominus^j) = \frac{\gamma(\frac{1}{2}\mathbb{1}\{v_j = 1\} + (\frac{1}{2}-\delta)\mathbb{1}\{v_j = -1\})}{2(1-\gamma)} \cdot d^\star(s_1^j), \qquad d^\star(s_{-1}) = 0.$$

This allows us to construct the behavior distribution $\mu_0$ as follows:

$$\mu_0(s_0^j) = \frac{d^\star(s_0^j)}{C^\star}, \qquad \mu_0(s_1^j, a_2) = \frac{d^\star(s_1^j)}{C^\star}, \qquad \mu_0(s_1^j, a_1) = d^\star(s_1^j) \cdot \left(1 - \frac{1}{C^\star}\right)$$

$$\mu_0(s_\oplus^j) = \frac{3}{4} \cdot \frac{\gamma}{2(1-\gamma)C^\star} \cdot d^\star(s_1^j), \qquad \mu_0(s_\ominus^j) = \frac{1}{2} \cdot \frac{\gamma}{2(1-\gamma)c^\star} \cdot d^\star(s_1^j),$$

$$\mu_0(s_{-1}) = 1 - \sum_j (\mu_0(s_0^j) + \mu_0(s_1^j) + \mu_0(s_\oplus^j) + \mu_0(s_\ominus^j))$$

It is easy to check that for any $v_j \in \{-1, 1\}$, $\delta \in [0, 1/4]$, one has $\mu_0(s_{-1}) > 0$, and more importantly

$$(\rho_0, \mu_0, P, R) \in \mathsf{MDP}(C^\star).$$

Since in this construction of MDP, the reward distribution is deterministic and fixed, and we only need to change the transition dynamics $P$, which is governed by the choice of $\delta$ and $v_{j_{1 \leq k \leq S/4}}$. Hence we write the loss/sub-optimality of a policy $\pi$ w.r.t. a particular design of $P$ as

$$\mathcal{L}(\pi; P) = J_P(\pi^\star) - J_P(\pi).$$

Our target then becomes

$$\inf_{\hat{\pi}} \sup_{(\rho_0, \mu_0, P, R) \in \mathsf{MDP}(C^\star)} \mathbb{E}[\mathcal{L}(\hat{\pi}; P)] \gtrsim \min\left(\frac{1}{1-\gamma}, \sqrt{\frac{SC^\star}{(1-\gamma)^3 N}}\right).$$

It remains to construct a set of transition probabilities (determined by $\delta$ and $\boldsymbol{v}$) that are nearly indistinguishable given the data. Similar to the construction in the lower bound for contextual bandits, we leverage the Gilbert-Varshamov lemma (cf. Lemma 15) to obtain a set $\mathcal{V} \subseteq \{-1, 1\}^{S/4}$ that obeys (1) $|\mathcal{V}| \geq \exp(S/32)$ and (2) $\|\boldsymbol{v}_1 - \boldsymbol{v}_2\|_1 \geq S/8$ for any $\boldsymbol{v}_1, \boldsymbol{v}_2 \in \mathcal{V}$ with $\boldsymbol{v}_1 \neq \boldsymbol{v}_2$. Each element $\boldsymbol{v} \in \mathcal{V}$ is mapped to a transition probability at $s_1^j$ such that the probability of transiting to $s_\oplus^j$ associated with $(s_1^j, a_2)$ is $\frac{1}{2} + v_j\delta$. We denote the resulting set of transition probabilities as $\mathcal{P}$. We record a useful characteristic of this family $\mathcal{P}$ of transition dynamics below, which results from the second property of the set $\mathcal{V}$.

**Lemma 7.** *For any policy $\pi$ and any two different transition probabilities $P_1, P_2 \in \mathcal{P}$, the following holds:*

$$\mathcal{L}(\pi; P_1) + \mathcal{L}(\pi; P_2) \geq \frac{\delta}{32(1-\gamma)}.$$

**Application of Fano's inequality.**    We are now ready to apply Fano's inequality, that is

$$\inf_{\hat{\pi}} \sup_{P \in \mathcal{P}} \mathbb{E}[\mathcal{L}(\hat{\pi}; P)] \geq \frac{\delta}{64(1-\gamma)}\left(1 - \frac{N \max_{i \neq j} \mathsf{KL}\left(\mu_0 \otimes P_i \| \mu_0 \otimes P_j\right) + \log 2}{\log|\mathcal{P}|}\right).$$

It remains to controlling $\max_{i \neq j} \mathsf{KL}\left(\mu_0 \otimes P_i \| \mu_0 \otimes P_j\right)$ and $\log|\mathcal{P}|$. For the latter quantity, we have

$$\log|\mathcal{P}| = \log|\mathcal{V}| \geq S/32,$$

where the inequality comes from the first property of the set $\mathcal{V}$. With regards to the KL divergence, one has

$$\max_{i \neq j} \mathsf{KL}\left(\mu_0 \otimes P_i \| \mu_0 \otimes P_j\right) \leq \frac{4(1-\gamma)}{SC^\star} \cdot \frac{S}{4} \cdot 16\delta^2 = \frac{16(1-\gamma)\delta^2}{C^\star},$$

since $\mu_0(s_1^j, a_2) \in [\frac{1-\gamma}{SC^\star}, \frac{4(1-\gamma)}{SC^\star}]$. As a result, we conclude that as long as

$$\frac{c_3(1-\gamma)N\delta^2}{SC^\star} \leq 1$$

for some universal constant $c_3$, one has

$$\inf_{\hat{\pi}} \sup_P \mathbb{E}[[\mathcal{L}(\hat{\pi}; P)] \gtrsim \frac{\delta}{1-\gamma}.$$

To finish the proof, we can set $\delta = \sqrt{\frac{SC^\star}{c_3(1-\gamma)N}}$ when $\sqrt{\frac{SC^\star}{c_3(1-\gamma)N}} < \frac{1}{4}$ and $\delta = \frac{1}{4}$ otherwise. This yields the desired lower bound (40).

**The case when $C^\star \in (1, 2)$.**    We intend to show that when $C^\star \in (1, 2)$,

$$\inf_{\hat{\pi}} \sup_{(\rho, \mu, P, R) \in \mathsf{MDP}(C^\star)} \mathbb{E}_\mathcal{D}[J(\pi^\star) - J(\hat{\pi})] \gtrsim \min\left(\frac{C^\star - 1}{1-\gamma}, \sqrt{\frac{S(C^\star - 1)}{(1-\gamma)^3 N}}\right). \tag{41}$$

The proof is similar to that of the previous case but with a different construction for $\rho_0$ and $\mu_0$.

**Construction of the hard instance.**    Let $\rho_0(s_0^j) = 4(C^\star - 1)/S$, $\rho_0(s_{-1}) = 2 - C^\star$. From $d^\star = (1-\gamma)\rho(I - \gamma P^{\pi^\star})^{-1}$ we can derive $d^\star$ as follows.

$$d^\star(s_0^j) = \frac{8(C^\star - 1)}{(2+\gamma)S}, \qquad d^\star(s_1^j) = \frac{8\gamma(1-\gamma)(C^\star - 1)}{(2-\gamma)(2+\gamma)S} \in \left[\frac{(1-\gamma)(C^\star - 1)}{S}, \frac{4(1-\gamma)(C^\star - 1)}{S}\right],$$

$$d^\star(s_\oplus^j) = \frac{\gamma(\frac{1}{2}\mathbb{1}\{v_j = -1\} + (\frac{1}{2} + \delta)\mathbb{1}\{v_j = 1\})}{2(1-\gamma)} \cdot d^\star(s_1^j),$$

$$d^\star(s_\ominus^j) = \frac{\gamma(\frac{1}{2}\mathbb{1}\{v_j = 1\} + (\frac{1}{2} - \delta)\mathbb{1}\{v_j = -1\})}{2(1-\gamma)} \cdot d^\star(s_1^j), \qquad d^\star(s_{-1}) = 2 - C^\star.$$

This allows us to construct the behavior distribution $\mu_0$ as follows

$$\mu_0(s_0^j) = \frac{d^\star(s_0^j)}{C^\star}, \qquad \mu_0(s_1^j, a_1) = \mu_0(s_1^j, a_2) = \frac{d^\star(s_1^j)}{C^\star}$$

$$\mu_0(s_\oplus^j) = \frac{3}{4} \cdot \frac{\gamma}{2(1-\gamma)} \cdot d^\star(s_1^j), \qquad \mu_0(s_\ominus^j) = \frac{1}{2} \cdot \frac{\gamma}{2(1-\gamma)} \cdot d^\star(s_1^j),$$

$$\mu_0(s_{-1}) = 1 - \sum_j (\mu_0(s_0^j) + \mu_0(s_1^j) + \mu_0(s_\oplus^j) + \mu_0(s_\ominus^j))$$

Again, one can check that for any $v_j \in \{-1, 1\}$ and $\delta \in [0, 1/4]$, we have $\mu_0(s_{-1}) > 0$ and

$$(\rho_0, \mu_0, P, R) \in \mathsf{MDP}(C^\star).$$

We use the same family $\mathcal{P}$ of transition probabilities as before. Following the same proof as Lemma 7 and noting that the initial distribution is multiplied by an extra $C^\star - 1$ factor, we know that for any policy $\pi$, and any two different distributions $P_1, P_2 \in \mathcal{P}$,

$$\mathcal{L}(\pi; P_1) + \mathcal{L}(\pi; P_2) \geq \frac{(C^\star - 1)\delta}{32(1 - \gamma)}.$$

**Application of Fano's inequality.** Now we are ready to apply Fano's inequality, that is

$$\inf_{\hat{\pi}} \sup_{P \in \mathcal{P}} \mathbb{E}[\mathcal{L}(\hat{\pi}; P)] \geq \frac{\delta}{64(1 - \gamma)} \left( 1 - \frac{N \max_{i \neq j} \mathsf{KL}\left(\mu_0 \otimes P_i \| \mu_0 \otimes P_j\right) + \log 2}{\log |\mathcal{P}|} \right).$$

Now the KL divergence satisfies

$$\mathsf{KL}(\mu_0 \otimes P_i \| \mu_0 \otimes P_j) \leq \frac{4(1 - \gamma)(C^\star - 1)}{SC^\star} \cdot \frac{S}{4} \cdot 16\delta^2 = \frac{16(1 - \gamma)(C^\star - 1)\delta^2}{C^\star}.$$

Here the first inequality comes from that $\mu_0(s_1^j) = \frac{c_2(1 - \gamma)(C^\star - 1)}{SC^\star}$ for some constant $c_2 \in [1, 4]$. As a result, we conclude that as long as

$$\frac{c_3(1 - \gamma)(C^\star - 1)N\delta^2}{SC^\star} \leq 1$$

for some universal constant $c_3$, one has

$$\inf_{\hat{\pi}} \sup_{P \in \mathcal{P}} \mathbb{E}[\mathcal{L}(\pi; P)] \gtrsim \frac{(C^\star - 1)\delta}{1 - \gamma}.$$

To finish the proof, we can set $\delta = \sqrt{\frac{SC^\star}{c_3(1 - \gamma)(C^\star - 1)N}}$ when $\sqrt{\frac{SC^\star}{c_3(1 - \gamma)(C^\star - 1)N}} < \frac{1}{4}$, and $\delta = \frac{1}{4}$ otherwise. This yields the desired lower bound (41).

**Putting the pieces together.** Now we are in position to summarize and simplify the three established lower bounds (39), (40), and (41).

When $C^\star = 1$, the claim in Theorem 7 is identical to the bound (39).

When $C^\star \geq 2$, we have from the bound (40) that

$$\inf_{\hat{\pi}} \sup_{P} \mathbb{E}[\mathcal{L}(\hat{\pi}; P)] \gtrsim \min \left( \frac{1}{1 - \gamma}, \sqrt{\frac{SC^\star}{(1 - \gamma)^3 N}} \right) \asymp \min \left( \frac{1}{1 - \gamma}, \sqrt{\frac{S(C^\star - 1)}{(1 - \gamma)^3 N}} \right).$$

Further notice that

$$\sqrt{\frac{S(C^\star - 1)}{(1 - \gamma)^3 N}} \geq \sqrt{\frac{S}{(1 - \gamma)^4 N}} \geq \min \left( \frac{1}{1 - \gamma}, \frac{S}{(1 - \gamma)^2 N} \right).$$

The claimed lower bound in Theorem 7 arises.

In the end, when $C^\star \in (1, 2)$, we know from the bounds (39) and (41) that

$$\inf_{\hat{\pi}} \sup_{P} \mathbb{E}[\mathcal{L}(\hat{\pi}; P)] \gtrsim \max \left\{ \min \left( \frac{1}{1 - \gamma}, \frac{S}{(1 - \gamma)^2 N} \right), \min \left( \frac{C^\star - 1}{1 - \gamma}, \sqrt{\frac{S(C^\star - 1)}{(1 - \gamma)^3 N}} \right) \right\}$$

$$\asymp \min \left( \frac{1}{1 - \gamma}, \frac{S}{(1 - \gamma)^2 N} + \sqrt{\frac{S(C^\star - 1)}{(1 - \gamma)^3 N}} \right),$$

which completes the proof.

### D.7.1 Proof of Lemma 7

By definition, one has

$$\mathcal{L}(\pi; P_1) + \mathcal{L}(\pi; P_2) = J_{P_1}(\pi^\star) - J_{P_1}(\pi) + J_{P_2}(\pi^\star) - J_{P_2}(\pi)$$

$$= \sum_{j=1}^{S/4} \rho_0(s_0^j) \left( V_{P_1}^\star(s_0^j) - V_{P_1}^\pi(s_0^j) + V_{P_2}^\star(s_0^j) - V_{P_2}^\pi(s_0^j) \right),$$

where we have ignored the state $s_{-1}$ since it has zero rewards. Our proof consists of three steps. We first connect the value difference $V_{P_1}^\star(s_0^j) - V_{P_1}^\pi(s_0^j)$ at $s_0^j$ to that $V_{P_1}^\star(s_1^j) - V_{P_1}^\pi(s_1^j)$ at $s_1^j$. Then, we further link the value difference at $s_1^j$ to the difference in transition probabilities, i.e., $\delta$ in our design. In the end, we use the property of the set $\mathcal{V}$ to conclude the lower bound.

**Step 1.** Since at state $s_0^j$, we only have one action $a_1$ with $r(s_0^j, a_1) = 0$, from the definition of value function one has

$$V_{P_1}^\pi(s_0^j) = \sum_{i=0}^\infty \gamma^{i+1}(1-p)p^i V_{P_1}^\pi(s_1^j),$$

for any policy $\pi$. Thus we have

$$V_{P_1}^\star(s_0^j) - V_{P_1}^\pi(s_0^j) = \sum_{i=0}^\infty \gamma^{i+1}(1-p)p^i \left( V_{P_1}^\star(s_1^j) - V_{P_1}^\pi(s_1^j) \right) > \frac{1}{4} \left( V_{P_1}^\star(s_1^j) - V_{P_1}^\pi(s_1^j) \right),$$

where we have used the fact that (assuming $\gamma \geq 1/2$)

$$\sum_{i=0}^\infty \gamma^{i+1}(1-p)p^i = \frac{1}{2}\gamma \geq \frac{1}{4}.$$

The same conclusion holds for $P_2$. Therefore we can obtain the following lower bound

$$\mathcal{L}(\pi; P_1) + \mathcal{L}(\pi; P_2) \geq \frac{1}{S} \sum_{j=1}^{S/4} \left( V_{P_1}^\star(s_1^j) - V_{P_1}^\pi(s_1^j) + V_{P_2}^\star(s_1^j) - V_{P_2}^\pi(s_1^j) \right).$$

**Step 2.** Without loss of generality, we assume that under $P_1$, $\mathbb{P}(s_\oplus^j \mid s_1^j, a_2) = \frac{1}{2} + \delta$, i.e., $v_j = +1$. Clearly, in this case, $a_2$ is the optimal action at $s_1^j$. If the policy $\pi$ chooses the sub-optimal action (i.e., $a_1$) at $s_1^j$, then we have

$$V_{P_1}^\star(s_1^j) - V_{P_1}^\pi(s_1^j) = \gamma \left( \left( \frac{1}{2} + \delta \right) V_{P_1}^\star\left( s_\oplus^j \right) + \left( \frac{1}{2} - \delta \right) V_{P_1}^\star\left( s_\ominus^j \right) - \frac{1}{2}V_{P_1}^\pi\left( s_\oplus^j \right) - \frac{1}{2}V_{P_1}^\pi\left( s_\ominus^j \right) \right)$$

$$\geq \gamma\delta \left( V_{P_1}^\star\left( s_\oplus^j \right) - V_{P_1}^\star\left( s_\ominus^j \right) \right)$$

$$\geq \gamma\delta \sum_{i=0}^\infty \gamma^i q^i = \frac{\gamma\delta}{1 - \gamma q} = \frac{\gamma\delta}{2(1 - \gamma)}.$$

On the other hand, if $\pi(s_1^j)$ is not the optimal action ($a_1$ in this case), we have the trivial lower bound $V_{P_1}^\star(s_1^j) - V_{P_1}^\pi(s_1^j) \geq 0$. As a result, we obtain

$$V_{P_1}^\star(s_1^j) - V_{P_1}^\pi(s_1^j) \geq \frac{\gamma\delta}{2(1 - \gamma)} \mathbf{1}\left\{ \pi(s_1^j) \neq \pi_{P_1}^\star(s_1^j) \right\},$$

which implies

$$\mathcal{L}(\pi; P_1) + \mathcal{L}(\pi; P_2) \geq \frac{1}{S} \cdot \frac{\gamma\delta}{2(1-\gamma)} \sum_{j=1}^{S/4} \left( 1\left\{\pi(s_1^j) \neq \pi_{P_1}^\star(s_1^j)\right\} + 1\left\{\pi(s_1^j) \neq \pi_{P_2}^\star(s_1^j)\right\} \right)$$

$$\geq \frac{1}{S} \cdot \frac{\gamma\delta}{2(1-\gamma)} \sum_{j=1}^{S/4} 1\left\{\pi_{P_1}^\star(s_1^j) \neq \pi_{P_2}^\star(s_1^j)\right\}.$$

**Step 3.** In the end, we use the second property of the set $\mathcal{V}$, namely for any $\boldsymbol{v}_i \neq \boldsymbol{v}_j$ in $\mathcal{V}$, one has $\|\boldsymbol{v}_i - \boldsymbol{v}_j\|_1 \geq S/8$. An immediate consequence is that

$$\sum_{j=1}^{S/4} 1\left\{\pi_{P_1}^\star(s_1^j) \neq \pi_{P_2}^\star(s_1^j)\right\} = \|\boldsymbol{v}_{P_1} - \boldsymbol{v}_{P_2}\|_1 \geq \frac{S}{8}.$$

Taking the previous three steps collectively completes the proof.

### D.7.2  Proof of Lemma 6

In the case of $C^\star = 1$, we have $d^\star = \mu$ which is the imitation learning setting. We adapt the information-theoretic lower bound for the episodic MDPs given in the work Rajaraman et al. (2020, Theorem 6) to the discounted setting.

**Notations and Setup:** Let $\mathcal{S}(\mathcal{D})$ be the set of all states that are observed in dataset $\mathcal{D}$. When $C^\star = 1$, we know the optimal policy $\pi^\star(s)$ at all states $s \in \mathcal{S}(\mathcal{D})$ visited in the dataset $\mathcal{D}$. We define $\Pi_{\mathrm{mimic}}(\mathcal{D})$ as the family of deterministic policies which always take the optimal action on each state visited in $\mathcal{D}$, namely,

$$\Pi_{\mathrm{mimic}}(\mathcal{D}) := \left\{\forall s \in \mathcal{S}(\mathcal{D}), \ \pi(s) = \pi^\star(s)\right\}, \tag{42}$$

Informally, $\Pi_{\mathrm{mimic}}(\mathcal{D})$ is the family of policies which are "compatible" with the dataset collected by the learner.

Define $\mathbb{M}_{\mathcal{S},\mathcal{A}}$ as the family of MDPs over state space $\mathcal{S}$ and action space $\mathcal{A}$. We proceed by by lower bounding the Bayes expected suboptimality. That is, we aim at finding a distribution $\mathcal{P}$ over MDPs supported on $\mathbb{M}_{\mathcal{S},\mathcal{A}}$ such that,

$$\mathbb{E}_{\mathsf{MDP}\sim\mathcal{P}}\left[J(\pi^\star) - \mathbb{E}_{\mathcal{D}}\left[J(\hat{\pi})\right]\right] \gtrsim \min\left\{\frac{1}{1-\gamma}, \frac{S}{(1-\gamma)^2 N}\right\},$$

where $\hat{\pi}$ is a function of dataset $\mathcal{D}$.

**Construction of the distribution $\mathcal{P}$:** We first determine the distribution of the optimal policy, and then we design $\mathcal{P}$ such that conditioned on the optimal policy, the distribution is deterministic. We let the distribution of the optimal policy be uniform over all deterministic policies. That is, for each $s \in \mathcal{S}$, $\pi^\star(s) \sim \mathrm{Unif}(\mathcal{A})$. For every $\pi^\star$, we construct an MDP instance in in Figure 2. Hence the distribution over MDPs comes from the randomness in $\pi$.

For a fixed optimal policy $\pi^\star$, the MDP instance $\mathsf{MDP}[\pi^\star]$ is determined as follows: we initialize with a fixed initial distribution over states $\rho = \{\zeta, \cdots, \zeta, 1-(S-2)\zeta, 0\}$ where $\zeta = \frac{1}{N+1}$. Let the last state be a special state $b$ which we refer to as the "bad state". At each state $s \in \mathcal{S} \setminus \{b\}$, choosing the optimal action renews the state in the initial distribution $\rho$ and gives a reward of 1, while any other

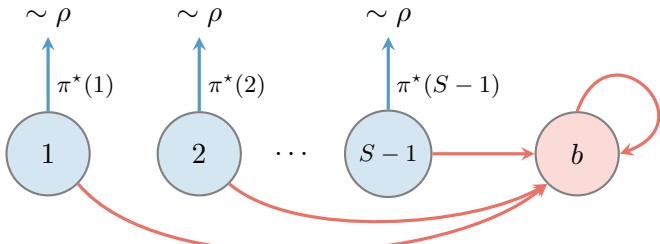

Figure 2: The hard MDP instance for the case $C^\star = 1$. Upon playing the optimal (blue) action at any state except $b$, the learner returns to a new state according to initial distribution $\rho = \{\zeta, \cdots, \zeta, 1-(S-2)\zeta, 0\}$ where $\zeta = \frac{1}{N+1}$. Any other choice of action (red) deterministically transitions the state to $b$.

choice of action deterministically induces a transition to the bad state $b$ and offers zero reward. In addition, the bad state is absorbing and dispenses no reward regardless of the choice of action. That is,

$$P(\cdot \mid s, a) = \begin{cases} \rho, & s \in \mathcal{S} \setminus \{b\}, \ a = \pi^\star(s) \\ \delta_b, & \text{otherwise,} \end{cases} \tag{43}$$

and the reward function of the MDP is given by

$$r(s, a) = \begin{cases} 1, & s \in \mathcal{S} \setminus \{b\}, \ a = \pi^\star(s), \\ 0, & \text{otherwise.} \end{cases} \tag{44}$$

Under this construction, it is easy to see that $J_{\mathsf{MDP}}(\pi^\star(\mathsf{MDP})) = 1/(1-\gamma)$ since the optimal action always acquires reward 1 throughout the trajectory. Thus the Bayes risk can be written as

$$\mathbb{E}_{\mathsf{MDP} \sim \mathcal{P}}\left[\frac{1}{1-\gamma} - \mathbb{E}\left[J_{\mathsf{MDP}}(\widehat{\pi}(\mathcal{D}))\right]\right]. \tag{45}$$

**Understanding the conditional distribution.** Now we study the conditional distribution of the MDP given the observed dataset $\mathcal{D}$. We start from the conditional distribution of the optimal policy. We present the following lemma without proof.

**Lemma 8** (Rajaraman et al. (2020, Lemma A.14)). *Conditioned on the dataset $\mathcal{D}$ collected by the learner, the optimal policy $\pi^\star$ is distributed $\sim \mathrm{Unif}(\Pi_{\mathrm{mimic}}(\mathcal{D}))$. In other words, at each state visited in the dataset, the optimal action is fixed. At the remaining states, the optimal action is sampled uniformly from $\mathcal{A}$.*

Now we define the conditional distribution of the MDPs given the dataset $\mathcal{D}$ collected by the learner as below.

**Definition 1.** *Define $\mathcal{P}(\mathcal{D})$ as the distribution of $\mathsf{MDP}$ conditioned on the observed dataset $\mathcal{D}$. In particular, $\pi^\star \sim \mathrm{Unif}(\Pi_{\mathrm{mimic}}(\mathcal{D}))$ and $\mathsf{MDP} = \mathsf{MDP}[\pi^\star]$.*

From Lemma 8 and the definition of $\mathcal{P}(\mathcal{D})$ in Definition 1, applying Fubini's theorem gives

$$\mathbb{E}_{\mathsf{MDP} \sim \mathcal{P}}\left[\frac{1}{1-\gamma} - \mathbb{E}_{\mathcal{D}}\left[J(\widehat{\pi})\right]\right] = \mathbb{E}_{\mathcal{D}}\left[\mathbb{E}_{\mathsf{MDP} \sim \mathcal{P}}\left[\frac{1}{1-\gamma} - J(\widehat{\pi})\right]\right]. \tag{46}$$

**Lower bounding the Bayes Risk.** Next we relate the Bayes risk to the first time the learner visits a state unobserved in $\mathcal{D}$.

**Lemma 9.** *In the trajectory induced by the infinite-horizon MDP and policy, define the stopping time $\tau$ as the first time that the learner encounters a state $s \neq b$ that has not been visited in $\mathcal{D}$ at time $t$. That is,*

$$\tau = \begin{cases} \inf\{t : s_t \notin \mathcal{S}(\mathcal{D}) \cup \{b\}\} & \exists t : s_t \notin \mathcal{S}(\mathcal{D}) \cup \{b\} \\ +\infty & \textit{otherwise.} \end{cases} \tag{47}$$

*Then, conditioned on the dataset $\mathcal{D}$ collected by the learner,*

$$\mathbb{E}_{\mathsf{MDP}\sim\mathcal{P}(\mathcal{D})}\Big[J(\pi^\star) - \mathbb{E}\left[J(\widehat{\pi})\right]\Big] \geq \left(1 - \frac{1}{|\mathcal{A}|}\right) \mathbb{E}_{\mathsf{MDP}\sim\mathcal{P}(\mathcal{D})}\left[\mathbb{E}_{\widehat{\pi}(\mathcal{D})}\left[\frac{\gamma^\tau}{1-\gamma}\right]\right] \tag{48}$$

We defer the proof to the end of this section.

Plugging the result of Lemma 9 into equality (46), we obtain

$$\mathbb{E}_{\mathsf{MDP}\sim\mathcal{P}}\Big[J(\pi^\star) - \mathbb{E}\left[J(\widehat{\pi})\right]\Big] \geq \left(1 - \frac{1}{|\mathcal{A}|}\right) \mathbb{E}_{\mathcal{D}}\left[\mathbb{E}_{\mathsf{MDP}\sim\mathcal{P}(\mathcal{D})}\left[\mathbb{E}_{\widehat{\pi}(\mathcal{D})}\left[\frac{\gamma^\tau}{1-\gamma}\right]\right]\right],$$

$$\overset{(i)}{\geq} \left(1 - \frac{1}{|\mathcal{A}|}\right) \frac{1}{2(1-\gamma)} \mathbb{E}_{\mathcal{D}}\left[\mathbb{E}_{\mathsf{MDP}\sim\mathcal{P}(\mathcal{D})}\left[\mathrm{Pr}_{\widehat{\pi}(\mathcal{D})}\left[\tau \leq \lfloor\frac{1}{\log(1/\gamma)}\rfloor\right]\right]\right],$$

$$= \left(1 - \frac{1}{|\mathcal{A}|}\right) \frac{1}{2(1-\gamma)} \mathbb{E}_{\mathsf{MDP}\sim\mathcal{P}}\left[\mathbb{E}_{\mathcal{D}}\left[\mathrm{Pr}_{\widehat{\pi}(\mathcal{D})}\left[\tau \leq \lfloor\frac{1}{\log(1/\gamma)}\rfloor\right]\right]\right],$$

where $(i)$ uses Markov's inequality. Lastly we bound the probability that we visit a state unobserved in the dataset before time $\lfloor\frac{1}{\log(1/\gamma)}\rfloor$. For any policy $\widehat{\pi}$, from a similar proof as Rajaraman et al. (2020, Lemma A.16) we have

$$\mathbb{E}_{\mathsf{MDP}\sim\mathcal{P}}\left[\mathbb{E}_{\mathcal{D}}\left[\mathrm{Pr}_{\widehat{\pi}}\left[\tau \leq \lfloor\frac{1}{\log(1/\gamma)}\rfloor\right]\right]\right] \gtrsim \min\left\{1, \frac{S}{\log(1/\gamma)N}\right\}. \tag{49}$$

Therefore,

$$\mathbb{E}_{\mathsf{MDP}\sim\mathcal{P}}\Big[J(\pi^\star) - \mathbb{E}\left[J(\widehat{\pi})\right]\Big] \gtrsim \left(1 - \frac{1}{|\mathcal{A}|}\right) \frac{1}{\log(1/\gamma)} \min\left\{1, \frac{S}{(1-\gamma)N}\right\}$$

$$\geq \left(1 - \frac{1}{|\mathcal{A}|}\right) \frac{\gamma}{1-\gamma} \min\left\{1, \frac{S}{(1-\gamma)N}\right\}$$

Here we use the fact that $\log(x) \leq x - 1$. Since $1 - \frac{1}{|\mathcal{A}|} \geq 1/2$ for $|\mathcal{A}| \geq 2$, the final result follows.

**Proof of Lemma 9.** To facilitate the analysis, we define an auxiliary random variable $\tau_b$ to be the first time the learner encounters the state $b$. If no such state is encountered, $\tau_b$ is defined as $+\infty$. Formally,

$$\tau_b = \begin{cases} \inf\{t : s_t = b\}, & \exists t : s_t = b, \\ +\infty, & \textit{otherwise.} \end{cases}$$

Conditioned on the observed dataset $\mathcal{D}$, we have

$$\frac{1}{1-\gamma} - \mathbb{E}_{\mathsf{MDP}\sim\mathcal{P}(\mathcal{D})}\left[J(\widehat{\pi})\right] = \frac{1}{1-\gamma} - \mathbb{E}_{\mathsf{MDP}\sim\mathcal{P}(\mathcal{D})}\left[\mathbb{E}_{\widehat{\pi}}\left[\sum_{t=0}^{\infty} \gamma^t r(s_t, a_t)\right]\right] \tag{50}$$

$$\geq \mathbb{E}_{\mathsf{MDP}\sim\mathcal{P}(\mathcal{D})}\left[\mathbb{E}_{\widehat{\pi}}\left[\frac{\gamma^{\tau_b - 1}}{1-\gamma}\right]\right] \tag{51}$$

where the last inequality follows from the fact that $r$ is bounded in $[0, 1]$, and the state $b$ is absorbing and always offers $0$ reward. Fixing the dataset $\mathcal{D}$ and the optimal policy $\pi^\star$ (which determines the MDP $\mathsf{MDP}[\pi^\star]$), we study $\mathbb{E}_{\widehat{\pi}(\mathcal{D})}\left[\frac{\gamma^{\tau_b-1}}{1-\gamma}\right]$ and try to relate it to $\mathbb{E}_{\widehat{\pi}(\mathcal{D})}\left[\frac{\gamma^\tau}{1-\gamma}\right]$. Note that for any $t$ and state $s \in \mathcal{S}$,

$$
\begin{aligned}
\Pr_{\widehat{\pi}}[\tau_b = t+1, \tau = t, s_t = s] &= \Pr_{\widehat{\pi}}[\tau_b = t+1 \mid \tau = t, s_t = s] \Pr_{\widehat{\pi}}[\tau = t, s_t = s] \\
&= \left(1 - \mathbb{1}\{\widehat{\pi}(s) = \pi^\star(s)\}\right)\Pr_{\widehat{\pi}}[\tau = t, s_t = s].
\end{aligned}
$$

In the last equation, we use the fact that the learner must play an action other than $\pi^\star(s_t)$ to visit $b$ at time $t+1$. Next we take an expectation with respect to the randomness of $\pi^\star$ which conditioned on $\mathcal{D}$ is drawn from $\mathrm{Unif}(\Pi_{\mathrm{mimic}}(\mathcal{D}))$. Note that $\mathsf{MDP}[\pi^\star]$ is also determined conditioning on $\pi^\star$. Observe that the dependence of the second term $\Pr_{\widehat{\pi}}[\tau = t, s_t = s]$ on $\pi^\star$ comes from the probability computed with the underlying MDP chosen as $\mathsf{MDP}[\pi^\star]$. However it only depends on the characteristics of $\mathsf{MDP}[\pi^\star]$ on the observed states in $\mathcal{D}$. On the other hand, the first term $(1 - \mathbb{1}\{\widehat{\pi}(s) = \pi^\star(s)\})$ depends only on $\pi^\star(s)$, where $s$ is an unobserved state. Thus the two terms are independent. By taking expectation with respect to the randomness of $\pi^\star \sim \mathrm{Unif}(\Pi_{\mathrm{mimic}}(\mathcal{D}))$ and $\mathsf{MDP} = \mathsf{MDP}[\pi^\star]$, we have

$$
\begin{aligned}
&\mathbb{E}_{\mathsf{MDP}\sim\mathcal{P}(\mathcal{D})}\left[\Pr_{\widehat{\pi}(\mathcal{D})}[\tau_b = t+1, \tau = t, s_t = s]\right] \\
&= \mathbb{E}_{\mathsf{MDP}\sim\mathcal{P}(\mathcal{D})}\left[1 - \mathbb{1}\{\widehat{\pi}(s) = \pi^\star(s)\}\right] \mathbb{E}_{\mathsf{MDP}\sim\mathcal{P}(\mathcal{D})}\left[\Pr_{\widehat{\pi}}[\tau = t, s_t = s]\right] \\
&= \left(1 - \frac{1}{|\mathcal{A}|}\right) \mathbb{E}_{\mathsf{MDP}\sim\mathcal{P}(\mathcal{D})}\left[\Pr_{\widehat{\pi}}[\tau = t, s_t = s]\right]
\end{aligned}
$$

where in the last equation, we use the fact that conditioned on $\mathcal{D}$ either $(i)$ $s = b$, in which case $\tau \neq t$ and both sides are $0$, or (ii) if $s \neq b$, then $\tau = t$ implies that the state $s$ visited at time $t$ must not be observed in $\mathcal{D}$, so $\pi^\star(s) \sim \mathrm{Unif}(\mathcal{A})$. Using the fact that $\Pr_{\widehat{\pi}}[\tau_b = t+1, \tau = t, s_t = s] \leq \Pr_{\widehat{\pi}}[\tau_b = t+1, s_t = s]$ and summing over $s \in \mathcal{S}$ results in the inequality,

$$
\mathbb{E}_{\mathsf{MDP}\sim\mathcal{P}(\mathcal{D})}\left[\Pr_{\widehat{\pi}}[\tau_b = t+1]\right] \geq \left(1 - \frac{1}{|\mathcal{A}|}\right) \mathbb{E}_{\mathsf{MDP}\sim\mathcal{P}(\mathcal{D})}\left[\Pr_{\widehat{\pi}}[\tau = t]\right].
$$

Multiplying both sides by $\frac{\gamma^t}{1-\gamma}$ and summing over $t = 1, \cdots, \infty$,

$$
\mathbb{E}_{\mathsf{MDP}\sim\mathcal{P}(\mathcal{D})}\left[\mathbb{E}_{\widehat{\pi}}\left[\frac{\gamma^{\tau_b-1}}{1-\gamma}\right]\right] \geq \left(1 - \frac{1}{|\mathcal{A}|}\right) \mathbb{E}_{\mathsf{MDP}\sim\mathcal{P}(\mathcal{D})}\left[\mathbb{E}_{\widehat{\pi}}\left[\frac{\gamma^\tau}{1-\gamma}\right]\right].
$$

here we use the fact that the initial distribution $\rho$ places no mass on the bad state $b$. Therefore, $\Pr_{\widehat{\pi}(D)}[\tau_b = 1] = \rho(b) = 0$. This equation in conjunction with (51) completes the proof.

## D.8 Imitation learning in discounted MDPs

In Theorem **??**, we have shown that imitation learning has a worse rate than LCB even in the contextual bandit case when $C^\star \in (1, 2)$. In this section, we show that if we change the concentrability assumption from density ratio to conditional density ratio, behavior cloning continues to work in certain regime. This also shows that behavior cloning works when $C^\star = 1$ in the discounted MDP case.

**Theorem 1.** *Assume the expert policy $\pi^\star$ is deterministic and that $\max \frac{(1-\gamma)d^*(a|s)}{\mu(a|s)} \leq C^\star$ for some $C^\star \in [1, 2)$. We consider a variant of behavior cloning policy:*

$$\Pi_{mimic} = \{\pi \in \Pi_{det} : \forall s \in \mathcal{D}, \pi(\cdot \mid s) = \arg\max_a N(s, a)\}. \tag{52}$$

*Here $\pi \in \Pi_{det}$ refers to the set of all deterministic policies. Then for any $\hat{\pi} \in \Pi_{mimic}$, we have*

$$\mathbb{E}_{\mathcal{D}}[J(\pi^*) - J(\hat{\pi})] \lesssim \frac{S}{C_0 N(1-\gamma)^2},$$

*where $C_0 = 1 - \exp\left(-\mathsf{KL}\left(\frac{1}{2}\|\frac{1}{C^\star}\right)\right)$.*

*Proof.* Define the following population loss:

$$\mathcal{L}(\hat{\pi}, \pi^*) = \mathbb{E}_{\mathcal{D}}[\mathbb{E}_{s\sim d_\star}[1\{\hat{\pi}(s) \neq \pi^*(s)\}]]. \tag{53}$$

From Lemma 5, we know that it suffices to control the population loss $\mathcal{L}(\hat{\pi}, \pi^\star)$. From a similar argument as in Rajaraman et al. (2020), we know that when $C^\star = 1$, the expected suboptimality of $\hat{\pi}$ is upper bounded by $\min(\frac{1}{1-\gamma}, \frac{S}{(1-\gamma)^2 N})$.

When $C^\star \in (1, 2)$, the contribution to the indicator loss can be decomposed into two parts: (1) the loss incurred due to the states not included in $\mathcal{D}$ whose expected value is upper bounded by $S/N$; (2) the loss incurred due to states the states for which the optimal action is not the most frequent in $\mathcal{D}$. Conditioned on $N(s)$ and from $\mu(\pi^\star(s)|s) \geq d^\star(\pi^\star(s)|s)/C^\star = 1/C^\star$ the probability of not picking the optimal action is upper bounded by $\exp(-N(s) \cdot \mathsf{KL}\left(\mathrm{Bern}\left(\frac{1}{2}\right) \| \mathrm{Bern}\left(\frac{1}{C^\star}\right)\right))$ using Chernoff's inequality. We have

$$\mathbb{E}[\mathcal{L}(\hat{\pi}, \pi^\star)] \tag{54}$$
$$= \mathbb{E}_{s\sim d_\star, \mathcal{D}}[1\{\hat{\pi}(s) \neq \pi^\star(s)\}]$$
$$\leq \mathbb{E}_{s\sim d_\star, \mathcal{D}}[\mathbb{P}(N(s) = 0)] + \mathbb{E}_{s\sim d_\star} \mathbb{E}_{\mathcal{D}}[\mathbb{P}(\hat{\pi}(s) \neq \pi^\star(s)) \mid N(s) \geq 1]$$
$$\lesssim \frac{S}{N} + \mathbb{E}_{s\sim d_\star} \mathbb{E}_{\mathcal{D}}\left[\exp\left(-N(s) \cdot \mathsf{KL}\left(\mathrm{Bern}\left(\frac{1}{2}\right) \| \mathrm{Bern}\left(\frac{1}{C^\star}\right)\right)\right) \mid N(s) \geq 1)\right]$$
$$\lesssim \frac{S}{N} + \sum_s p(s) \sum_{n=1}^{N} \binom{N}{n} \exp\left(-n \cdot \mathsf{KL}\left(\mathrm{Bern}\left(\frac{1}{2}\right) \| \mathrm{Bern}\left(\frac{1}{C^\star}\right)\right)\right) p(s)^n (1 - p(s))^{N-n}$$
$$\leq \frac{S}{N} + \sum_s p(s) \left(1 - p(s)\left(1 - \exp\left(-\mathsf{KL}\left(\mathrm{Bern}\left(\frac{1}{2}\right) \| \mathrm{Bern}\left(\frac{1}{C^\star}\right)\right)\right)\right)\right)^N. \tag{55}$$

Denote $C_0 = 1 - \exp\left(-\mathsf{KL}\left(\mathrm{Bern}\left(\frac{1}{2}\right) \| \mathrm{Bern}\left(\frac{1}{C^\star}\right)\right)\right)$. Note that $\max_{x\in[0,1]} x(1 - C_0 x)^N \leq \frac{1}{C_0(N+1)}(1 - \frac{1}{N+1})^N \leq \frac{4}{9C_0 N}$. Thus we have $\mathbb{E}[\mathcal{L}(\hat{\pi}, \pi^\star)] \leq \frac{4S}{9C_0 N}$. We then use Lemma 5 to conclude that the final sub-optimality is upper bounded by $\frac{S}{C_0 N(1-\gamma)^2}$. $\square$

# E   LCB in episodic Markov decision processes

The aim of this section is to illustrate the validity of Conjecture **??** in episodic MDPs. In Section E.1, we give a brief review of episodic MDPs, describing the batch dataset and offline RL objective in this setting, and introducing additional notation. We then present a variant of the VI-LCB algorithm (Algorithm 3) for episodic MDPs and state its sub-optimality guarantees in Section E.2. In Section E.3, we show that the proposed penalty captures a confidence interval and prove a value

difference lemma for Algorithm 3. Section E.4 is devoted to the proof of the sub-optimality upper bound. In Section E.5, we give an alternative sub-optimality decomposition as an attempt to obtain a tight dependency on $C^\star$ in the regime $C^\star \in [1, 2)$. We analyze the sub-optimality in this regime in a special example provided in Section E.6.

## E.1 Model and notation

**Episodic MDP.** We consider an episodic MDP described by a tuple $(\mathcal{S}, \mathcal{A}, \mathcal{P}, \mathcal{R}, \rho, H)$, where $\mathcal{S} = \{\mathcal{S}_h\}_{h=1}^{H}$ is the state space, $\mathcal{A}$ is the action space, $\mathcal{P} = \{P_h\}_{h=1}^{H}$ is the set of transition kernels with $P_h : \mathcal{S}_h \times \mathcal{A} \mapsto \Delta(\mathcal{S}_{h+1})$, $\mathcal{R} = \{R_h\}_{h=1}^{H}$ is the set of reward distributions $R_h : \mathcal{S}_h \times \mathcal{A} \to \Delta([0, 1])$ with $r : \mathcal{S} \times \mathcal{A} \mapsto [0, 1]$ as the expected reward function, $\rho : \mathcal{S}_1 \to \Delta(\mathcal{S}_1)$ is the initial distribution, and $H$ is the horizon. To streamline our analysis, we assume that $\{\mathcal{S}_h\}_{h=1}^{H}$ partition the state space $\mathcal{S}$ and are disjoint.

**Policy and value functions.** Similar to the discounted case, we consider deterministic policies $\pi : \mathcal{S} \mapsto \mathcal{A}$ that map each state to an action. For any $h \in \{1, \ldots, H\}$, $s \in \mathcal{S}_h$, and $a \in \mathcal{A}_h$, the value function $V_h^\pi : \mathcal{S} \mapsto \mathbb{R}$ and Q-function $Q_h^\pi : \mathcal{S} \times \mathcal{A} \mapsto \mathbb{R}$ are respectively defined as

$$V_h^\pi(s) := \mathbb{E}\left[\sum_{i=h}^{H} r_i \,\middle|\, s_h = s, a_i = \pi(s_i) \text{ for } i \geq h\right],$$

$$Q_h^\pi(s, a) := \mathbb{E}\left[\sum_{i=h}^{H} r_i \,\middle|\, s_h = s, a_h = a, a_i = \pi(s_i) \text{ for } i \geq h+1\right].$$

Since we assume that the set of state in different levels are disjoint, we drop the subscript $h$ when it is it clear from the context. The expected value of a policy $\pi$ is defined analogously to the discounted case:

$$J(\pi) := \mathbb{E}_{s \sim \rho}[V_1^\pi(s)].$$

It is well-known that a deterministic policy $\pi^\star$ exists that maximizes the value function from any state.

**Episodic occupancy measures.** We define the (normalized) state occupancy measure $d_\pi : \mathcal{S} \mapsto [0, H]$ and state-action occupancy measure $d^\pi : \mathcal{S} \times \mathcal{A} \mapsto [0, H]$ as

$$d_\pi(s) := \frac{1}{H} \sum_{h=1}^{H} \mathbb{P}_h(s_h = s; \pi), \quad \text{and} \quad d^\pi(s, a) := \frac{1}{H} \sum_{h=1}^{H} \mathbb{P}_h(s_h = s, a_h = a; \pi), \tag{56}$$

where we overload notation and write $\mathbb{P}_h(s_h = s; \pi)$ to denote the probability of visiting state $s_h = s$ (and similarly $s_h = s, a_h = a$) at level $h$ after executing policy $\pi$ and starting from $s_1 \sim \rho(\cdot)$.

**Batch dataset.** The batch dataset $\mathcal{D}$ consists of tuples $(s, a, r, s')$, where $r = r(s, a)$ and $s' \sim P(\cdot \mid s, a)$. As in the discounted case, we assume that $(s, a)$ pairs are generated i.i.d. according to a data distribution $\mu$, unknown to the agent. We denote by $N(s, a) \geq 0$ the number of times a pair $(s, a)$ is observed in $\mathcal{D}$ and by $N = |\mathcal{D}|$ the total number of samples.

---
**Algorithm 3** Episodic value iteration with LCB
---
1: **Inputs:** Batch dataset $\mathcal{D}$.
2: $\hat{V}_{H+1} \leftarrow 0$.
3: **for** $h = H - 1, \ldots, 1$ **do**
4:      **for** $s \in \mathcal{S}_h, a \in \mathcal{A}$ **do**
5:          **if** $N(s,a) = 0$ **then**
6:              Set $r(s,a) = 0$.
7:              Set the empirical transition vector $\hat{P}_{s,a}$ randomly.
8:              Set the penalty $b(s,a) = H\sqrt{L}$.
9:          **else**
10:             Set $r(s,a)$ according to dataset.
11:             Compute the empirical transition vector $\hat{P}_{s,a}$ according to dataset.
12:             Set the penalty $b(s,a) = H\sqrt{L/N(s,a)}$, where $L = 2000 \log(2S|\mathcal{A}|/\delta)$.
13:          Compute $\hat{Q}_h(s,a) \leftarrow r(s,a) - b(s,a) + \hat{P}_{s,a} \cdot \hat{V}_{h+1}$.
14:          Compute $\hat{V}_h(s) \leftarrow \max_a \hat{Q}_h(s,a)$ and $\hat{\pi}(s) \in \arg\max_a \hat{Q}_h(s,a)$.
15: **Return:** $\hat{\pi}$.
---

**The learning objective.** Fix a deterministic policy $\pi$. The expected sub-optimality of policy $\hat{\pi}$ computed based on dataset $\mathcal{D}$ competing with policy $\pi$ is defined as

$$\mathbb{E}_{\mathcal{D}}\left[J(\pi) - J(\hat{\pi})\right]. \tag{57}$$

**Assumption on dataset coverage.** Equipped with the definitions for occupancy densities in episodic MDPs, we define the concentrability coefficient in the episodic case analogously: given a deterministic policy $\pi$, $C^\pi$ is the smallest constant satisfying

$$\frac{d^\pi(s,a)}{\mu(s,a)} \leq C^\pi \qquad \forall s \in \mathcal{S}, a \in \mathcal{A}. \tag{58}$$

**Matrix notation.** We adopt a matrix notation similar to the one described in Section **??**.

**Bellman equations.** Given any value function $V : \mathcal{S}_{h+1} \mapsto \mathbb{R}$, the Bellman value operator at each level $h \in \{1, \ldots, H\}$

$$\mathcal{T}_h V = r_h + P_h V. \tag{59}$$

We write $(\mathcal{T}_h V)(s,a) = r_h(s,a) + (P_h V)(s,a)$ for $\mathcal{S} \in \mathcal{S}_h, a \in \mathcal{A}$.

## E.2    Episodic value iteration with LCB

Algorithm 3 presents a pseudocode for value iteration with LCB in the episodic setting. As in the classic value iteration in episodic MDPs, this algorithm computes values and policy through a backward recursion starting at $h = H$ with the distinction of subtracting penalties when computing the Q-function. This algorithm can be viewed as an instance of Algorithm 3 of Jin et al. (2020).

In the following theorem, we provide an upper bound on the expected sub-optimality of the policy returned by Algorithm 3. The proof is presented in Appendix E.4.

**Theorem 9** (LCB sub-optimality, episodic MDP). *Consider an episodic MDP and assume that*

$$\frac{d^\pi(s,a)}{\mu(s,a)} \le C^\pi \qquad \forall s \in \mathcal{S}, a \in \mathcal{A}$$

*holds for an arbitrary deterministic policy $\pi$. Set $\delta = 1/N$ in Algorithm 3. Then, for all $C^\pi \ge 1$, one has*

$$\mathbb{E}_\mathcal{D}[J(\pi) - J(\hat\pi)] \lesssim \min\left\{H, \widetilde{O}\left(H^2\sqrt{\frac{SC^\pi}{N}}\right)\right\}.$$

*In addition, if $1 \le C^\pi \le 1 + L/(200N)$, then we have a tighter performance guarantee*

$$\mathbb{E}_\mathcal{D}[J(\pi) - J(\hat\pi)] \lesssim \min\left\{H, \widetilde{O}\left(H^2\frac{S}{N}\right)\right\}.$$

We make the following conjecture that the sub-optimality rate smoothly transitions from $1/N$ to $1/\sqrt{N}$ as $C^\pi$ increases from 1 to 2.

**Conjecture 1.** *Assume as in Theorem 9. If $1 \le C^\pi \le 2$, then policy $\hat\pi$ returned by Algorithm 3 obeys*

$$\mathbb{E}_\mathcal{D}[J(\pi) - J(\hat\pi)] \lesssim \min\left\{H, \widetilde{O}\left(H^2\sqrt{\frac{S(C^\pi - 1)}{N}}\right)\right\}.$$

We present our attempt in proving the above conjecture in part in Appendix E.5 followed by an example in Appendix E.6.

## E.3   Properties of Algorithm 3

In this section, we prove two properties of Algorithm 3. We first prove that the penalty captures the Q-function lower confidence bound. Then, we prove a value difference lemma.

**Clean event in episodic MDPs.**   Define the following clean event

$$\mathcal{E}_{\text{EMDP}} := \left\{\forall h, \forall s \in \mathcal{S}_h, \forall a : \left|r(s,a) + P_{s,a} \cdot \hat{V}_{h+1} - \hat{r}(s,a) - \hat{P}_{s,a} \cdot \hat{V}_{h+1}\right| \le b_h(s,a)\right\}, \qquad (60)$$

where $\hat{V}_{H+1} = 0$. In the following lemma, we show that the penalty used in Algorithm 3 captures the confidence interval of the empirical expectation of the Q-function.

**Lemma 10** (Clean event probability, episodic MDP). *One has $\mathbb{P}(\mathcal{E}_{EMDP}) \ge 1 - \delta$.*

*Proof.* The proof is analogous to the proof of Lemma 3. Fix a tuple $(s, a, h)$. If $N(s, a) = 0$, it is immediate that

$$|r(s,a) + P_{s,a} \cdot \hat{V}_{h+1} - \hat{r}(s,a) - \hat{P}_{s,a} \cdot \hat{V}_{h+1}| \le H\sqrt{L}.$$

When $N(s, a) \ge 1$, we exploit the independence of $\hat{V}_{h+1}$ and $\hat{P}_{s,a}$ (thanks to the disjoint state space at each step $h$) and conclude by Hoeffding's inequality that for any $\delta_1 \in (0, 1)$

$$\mathbb{P}\left(|r(s,a) + P_{s,a} \cdot \hat{V}_{h+1} - \hat{r}(s,a) + P_{s,a} \cdot \hat{V}_{h+1}| \ge H\sqrt{\frac{2\log(2/\delta_1)}{N(s,a)}}\right) \le \delta_1.$$

The claim follows by taking a union bound over $s \in \mathcal{S}_h, a \in \mathcal{A}, h \in [H]$ and setting $\delta_1 = \delta/(S|\mathcal{A}|)$. $\qquad\square$

**Value difference lemma.** The following lemma bounds the sub-optimality of Algorithm 3 by expected bonus. This result is similar to Theorem 4.2 in Jin et al. (2020). We present the proof for completeness.

**Lemma 11** (Value difference for Algorithm 3). *Let $\pi$ be an arbitrary policy. On the event $\mathcal{E}_{EMDP}$, the policy $\hat{\pi}$ returned by Algorithm 3 satisfies*

$$J(\pi) - J(\hat{\pi}) \leq 2H \, \mathbb{E}_{d^\pi} \left[ b(s,a) \right].$$

*Proof.* Define the following self-consistency error

$$\iota_h(s,a) = \mathcal{T}_h \hat{V}_{h+1}(s,a) - \hat{Q}_h(s,a),$$

where $\mathcal{T}_h$ is the Bellman value operator defined in (59). Let $\pi'$ be an arbitrary policy. By Jin et al. (2020, Lemma A.1), one has

$$\hat{V}_1(s) - V_1^{\pi'}(s) = \sum_{h=1}^{H} \mathbb{E}[\hat{Q}_h(s_h, \hat{\pi}(s_h)) - \hat{Q}_h(s_h, \pi'(s_h)) \mid s_1 = s] \tag{61}$$
$$- \sum_{h=1}^{H} \mathbb{E}[\iota_h(s_h, \pi'(s_h)) \mid s_1 = s]$$

Setting $\pi' \leftarrow \pi$ in (61) gives

$$V_1^\pi(s) - \hat{V}_1(s) = \sum_{h=1}^{H} \mathbb{E}[\iota_h(s_h, \pi(s_h)) \mid s_1 = s] - \sum_{h=1}^{H} \mathbb{E}[\hat{Q}_h(s_h, \hat{\pi}(s_h)) - \hat{Q}_h(s_h, \pi(s_h)) \mid s_1 = s]$$
$$\leq \sum_{h=1}^{H} \mathbb{E}[\iota_h(s_h, \pi(s_h)) \mid s_1 = s], \tag{62}$$

where the last line uses the fact that $\hat{\pi}(s)$ maximizes $\hat{Q}_h(s,a)$.

We apply (61) once more, this time setting $\pi' \leftarrow \hat{\pi}$:

$$\hat{V}_1(s) - V_1^{\hat{\pi}}(s) = \sum_{h=1}^{H} \mathbb{E}[\hat{Q}_h(s_h, \hat{\pi}(s_h)) - \hat{Q}_h(s_h, \hat{\pi}(s_h)) \mid s_1 = s] - \sum_{h=1}^{H} \mathbb{E}[\iota_h(s_h, \hat{\pi}(s_h)) \mid s_1 = s]$$
$$\leq - \sum_{h=1}^{H} \mathbb{E}[\iota_h(s_h, \pi'(s_h)) \mid s_1 = s]. \tag{63}$$

Adding (62) and (63), we have

$$V_1^\pi(s) - V_1^{\hat{\pi}}(s) = V_1^\pi(s) - \hat{V}_1(s) + \hat{V}_1(s) - V_1^{\hat{\pi}}(s)$$
$$\leq \sum_{h=1}^{H} \mathbb{E}[\iota_h(s_h, \pi(s_h)) \mid s_1 = s] - \sum_{h=1}^{H} \mathbb{E}[\iota_h(s_h, \pi'(s_h)) \mid s_1 = s]. \tag{64}$$

By Jin et al. (2020, Lemma 5.1), conditioned on $\mathcal{E}_{EMDP}$, we have

$$0 \leq \iota_h(s,a) \leq 2b_h(s,a) \quad \forall s, a, h.$$

The proof is completed by applying the above bound in (64) and taking an expectation with respect to $\rho$

$$\mathbb{E}_\rho[V_1^\pi(s) - V_1^{\hat\pi}(s)] \leq 2\sum_{h=1}^H \mathbb{E}[b_h(s_h, \pi(s_h))]$$

$$= 2\sum_{h=1}^H P_h(s_h; \pi)b_h(s_h, \pi(s_h)) = 2H\,\mathbb{E}_{d^\pi}[b(s,a)],$$

where the last equation hinges on the definition of occupancy measure for episodic MDPs given in (56). $\qquad\square$

### E.4 Proof of Theorem 9

The proof follows a similar decomposition argument as in Theorem 6. Nonetheless, we present a complete proof for the reader's convenience.

We divide the proof into two parts and separately analyze the general case $C^\pi \geq 1$ and $C^\star \leq 1 + L/(200N)$ since the techniques used in the proof of these two claims are rather distinct.

**The general case when $C^\pi \geq 1$.** We decompose the expected sub-optimality into two terms

$$\mathbb{E}_\mathcal{D}\left[\sum_s \rho(s)[V_1^\pi(s) - V_1^{\hat\pi}(s)]\right] = \mathbb{E}_\mathcal{D}\left[\sum_s \rho(s)[V_1^\pi(s) - V_1^{\hat\pi}(s)]\,\mathbb{1}\{\mathcal{E}_{\mathrm{EMDP}}\}\right] =: T_1$$

$$+ \mathbb{E}_\mathcal{D}\left[\sum_s \rho(s)[V_1^\pi(s) - V_1^{\hat\pi}(s)]\,\mathbb{1}\{\mathcal{E}_{\mathrm{EMDP}}^c\}\right] =: T_2. \tag{65}$$

The first term $T_1$ captures the sub-optimality under the clean event $\mathcal{E}_{\mathrm{EMDP}}$ whereas $T_2$ represents the sub-optimality suffered when the constructed confidence interval via the penalty function falls short of containing the empirical Q-function estimate. We will prove in subsequent sections that $T_1$ and $T_2$ are bounded according to:

$$T_1 \leq 32H^2\sqrt{\frac{SC^\pi L}{N}} \tag{66a}$$

$$T_2 \leq H\delta. \tag{66b}$$

Taking the above bounds as given for the moment and setting $\delta = 1/N$, we conclude that

$$\mathbb{E}_\mathcal{D}[J(\pi) - J(\hat\pi)] \lesssim \min\left(H, 32H^2\sqrt{\frac{SC^\pi L}{N}}\right).$$

**The case when $C^\pi \leq 1 + L/(200N)$.** To obtain faster rates in this regime, we resort to directly analyzing the policy sub-optimality instead of bounding the value sub-optimality (such as by Lemma 11). It is useful to connect the sub-optimality of a policy to whether it disagrees with the optimal policy at each state. The following lemma due to Ross and Bagnell (2010, Theorem 2.1) provides such a connection.

**Lemma 12.** *For any deterministic policies $\pi, \hat\pi$, one has*

$$J(\pi) - J(\hat\pi) \leq H^2\,\mathbb{E}_{s\sim d_\pi}[\mathbb{1}\{\pi(s) \neq \hat\pi(s)\}].$$

We apply Lemma 12 to bound the sub-optimality and further decompose it based on whether any samples are observed on each state $s$.

$$\mathbb{E}_{\mathcal{D}}[\rho(s)[V_1^\pi(s) - V_1^{\hat\pi}(s)]]$$
$$\leq H^2 \, \mathbb{E}_{\mathcal{D}} \, \mathbb{E}_{d_\pi}[\mathbb{1}\{\pi(s) \neq \hat\pi(s)\}]$$
$$= H^2 \, \mathbb{E}_{\mathcal{D}} \, \mathbb{E}_{d_\pi}[\mathbb{1}\{\pi(s) \neq \hat\pi(s)\} \, \mathbb{1}\{N(s, \pi(s)) = 0\}] =: T_1'$$
$$+ H^2 \, \mathbb{E}_{\mathcal{D}} \, \mathbb{E}_{d_\pi}[\mathbb{1}\{\pi(s) \neq \hat\pi(s)\} \, \mathbb{1}\{N(s, \pi(s)) \geq 1\}] =: T_2'.$$

In a similar manner to the proof of Theorem 6, we prove the following bounds on $T_1'$ and $T_2'$:

$$T_1' \leq H^2 \frac{4C^\pi}{N}; \tag{67a}$$
$$T_2' \lesssim \frac{2SC^\pi H^2 L}{N} + H^2 \frac{|\mathcal{A}|}{N^9}. \tag{67b}$$

### E.4.1   Proof the bound (66a) on $T_1$

By the value difference Lemma 11, one has

$$\mathbb{E}_{\mathcal{D}}[\sum_s \rho(s)[V^\pi(s) - V^{\hat\pi}(s)] \, \mathbb{1}\{\mathcal{E}_{\text{EMDP}}\}] \leq 2H \sum_{s,a} d^\pi(s,a) \, \mathbb{E}_{\mathcal{D}}[b(s,a)]$$

$$\leq 2H \sum_{s,a} d^\pi(s,a) H \, \mathbb{E}_{\mathcal{D}} \left[ \sqrt{\frac{L}{N(s,a) \vee 1}} \right]$$

$$\leq 32H^2 \sum_{s,a} d^\pi(s,a) \left[ \sqrt{\frac{L}{N\mu(s,a)}} \right],$$

where the last inequality uses the bound on inverse moments of binomial random variables given in 14 with $c_{1/2} \leq 16$. We then apply the concentrability assumption and the Cauchy-Schwarz inequality to conclude that

$$T_1 \leq 32H^2 \sum_{s,a} \sqrt{d^\pi(s,a)} \sqrt{HC^\pi \mu(s,a)} \left[ \sqrt{\frac{L}{N\mu(s,a)}} \right]$$

$$\leq 32H^2 \sqrt{\frac{C^\pi L H}{N}} \sum_s \sqrt{d^\pi(s,\pi(s))} \leq 32H^2 \sqrt{\frac{SC^\pi L}{N}}.$$

### E.4.2   Proof of the bound (66b) on $T_2$

We use a argument similar to that in the proof of (32c). First, observe that $\sum_s \rho(s)[V_1^\pi(s) - V^{\hat\pi}(s)] \leq H$. Consequently, in light of Lemma 10 one can conclude

$$T_3 \leq H \, \mathbb{E}_{\mathcal{D}}[\mathbb{1}\{\mathcal{E}_{\text{EMDP}}^c\}] = H \, \mathbb{P}(\mathcal{E}_{\text{EMDP}}^c) \leq H\delta.$$

### E.4.3   Proof of the bound (67a) on $T_1'$

We have

$$T_1' \leq H^2 \, \mathbb{E}_{d_\pi} \, \mathbb{E}_{\mathcal{D}}[\mathbb{1}\{N(s, \pi(s)) = 0\}] \leq H^2 \, \mathbb{E}_{d_\pi} \, \mathbb{P}(N(s, \pi(s)) = 0).$$

It follows from the concentrability assumption $d^\pi(s, \pi(s))/\mu(s, \pi(s)) \le C^\pi$ that

$$T_1 \le H^2 \sum_s C^\pi \mu(s, \pi(s)) \, \mathbb{P}(N(s, \pi(s)) = 0) = H^2 C^\pi \sum_s \mu(s, \pi(s))(1 - \mu(s, \pi(s)))^N.$$

Note that $\max_{x \in [0,1]} x(1-x)^N \le 4/(9N)$. We thus conclude that

$$T_1 \le H^2 C^\pi \sum_s \mu(s, \pi(s))(1 - \mu(s, \pi(s)))^N \le H^2 \frac{4C^\pi}{9N}.$$

### E.4.4   Proof of the bound (67b) on $T_2'$

We prove the bound on $T_2'$ by partitioning the states based on how much they are occupied under the target policy. Define the following set:

$$\mathcal{O}_1 := \left\{ s \mid d_\pi(s) < \frac{2C^\pi L}{N} \right\}. \tag{68}$$

We can then decompose $T_2'$ according to whether state $s$ belongs to $\mathcal{O}_1$:

$$T_2' = H^2 \sum_{s \in \mathcal{O}_1} d_\pi(s) \, \mathbb{E}_{\mathcal{D}}[\mathbb{1}\{\hat\pi(s) \ne \pi(s)\} \, \mathbb{1}\{N(s, \pi(s)) \ge 1\}] =: T_{2,1}$$

$$+ H^2 \sum_{s \notin \mathcal{O}_1} d_\pi(s) \, \mathbb{E}_{\mathcal{D}}[\mathbb{1}\{\hat\pi(s) \ne \pi(s)\} \, \mathbb{1}\{N(s, \pi(s)) \ge 1\}] =: T_{2,2}.$$

Here, $T_{2,1}$ captures the sub-optimality due to the less important states under the target policy. We will shortly prove the following bounds on these two terms:

$$T_{2,1} \le \frac{2SC^\pi H^2 L}{N} \qquad \text{and} \qquad T_{2,2} \lesssim H^2 \frac{|\mathcal{A}|}{N^9}.$$

**Proof of the bound on $T_{2,1}$.** Since $\mathbb{E}_{\mathcal{D}}[\mathbb{1}\{\hat\pi(s) \ne \pi(s)\} \, \mathbb{1}\{N(s, \pi(s)) \ge 1\}] \le 1$, it follows immediately that

$$T_{2,1} \le H^2 \sum_{s \in \mathcal{S}_1} d_\pi(s) \le \frac{2SC^\pi H^2 L}{N},$$

where the last inequality relies on the definition of $\mathcal{O}_1$ provided in (68).

**Proof of the bound on $T_{2,2}$.** The term $T_{2,2}$ is equal to

$$T_{2,2} = H^2 \sum_{s \notin \mathcal{O}_1} d_\pi(s) \mathbb{P}\left( \hat\pi(s) \ne \pi(s), \ N(s, \pi(s)) \ge 1 \right).$$

We subsequently show that the probability $\mathbb{P}\left( \hat\pi(s) \ne \pi(s), \ N(s, \pi(s)) \ge 1 \right)$ is small. Fix a state $s \notin \mathcal{O}_1$ and let $h$ be the level to which $s$ belongs. The concentrability assumption along with the constraint on $d_\pi(s)$ implies the following lower bound on $\mu(s, \pi(s))$:

$$\mu(s, \pi(s)) \ge \frac{1}{C^\pi} d_\pi(s) \ge \frac{1}{C^\pi} \frac{2C^\pi L}{N} = \frac{2L}{N}. \tag{69}$$

On the other hand, by the concentrability assumption and using $C^\pi \leq 1 + \frac{L}{200N}$, the following upper bound holds for $\mu(s, a \neq \pi(s))$:

$$\mu(s, a) \leq \sum_{a \neq \pi(s)} \mu(s, a) \leq 1 - \frac{1}{C^\pi} \leq \frac{L}{200N}, \tag{70}$$

The above bounds suggest that the target action is likely to be included in the dataset more frequently than the rest of the actions for $s \notin \mathcal{O}_1$. We will see shortly that in this scenario, the LCB algorithm picks the target action with high probability. The bounds (69) and (70) together with Chernoff's bound give

$$\mathbb{P}\left(N(s, a \neq \pi(s)) \leq \frac{5L}{200}\right) \geq 1 - \exp\left(-\frac{L}{200}\right);$$

$$\mathbb{P}\left(N(s, \pi(s)) \geq L\right) \geq 1 - \exp\left(-\frac{L}{4}\right).$$

We can thereby write an upper bound $\hat{Q}_h(s, a \neq \pi(s))$ and a lower bound on $\hat{Q}_h(s, \pi(s))$. In particular, when $N(s, a) \leq \frac{5L}{200}$, one has

$$\hat{Q}_h(s, a) = r_h(s, a) - b_h(s, a) + \hat{P}_{s,a} \cdot \hat{V}_{h+1}$$

$$= r_h(s, a) - H\sqrt{\frac{L}{N(s, a) \vee 1}} + \hat{P}_{s,a} \cdot \hat{V}_{h+1}$$

$$\leq 1 - H\sqrt{\frac{L}{5L/200}} + H \leq -4H,$$

where we used the fact that $L \geq 70$. When $N(s, \pi(s)) \geq L$, one has

$$\hat{Q}_h(s, \pi(s)) = r_h(s, \pi(s)) - H\sqrt{\frac{L}{N(s, \pi(s))}} + \hat{P}_{s,\pi(s)} \cdot V_{h+1} \geq -H.$$

Note that if both $N(s, a \neq \pi(s)) \leq \frac{5L}{200}$ and $N(s, \pi(s)) \geq L$ hold, we must have $\hat{Q}_h(s, a \neq \pi(s)) < \hat{Q}_h(s, \pi(s))$. Therefore, we deduce that

$$\mathbb{P}\left(\hat{\pi}(s) \neq \pi(s), \ N(s, \pi(s)) \geq 1\right) \leq (|\mathcal{A}| - 1)\exp\left(-\frac{L}{200}\right) + \exp\left(-\frac{1}{4}L\right) \leq |\mathcal{A}|\exp\left(-\frac{L}{200}\right),$$

which further implies

$$T_{2,2} \leq H^2 \sum_{s \notin \mathcal{O}_1} d_\pi(s)|\mathcal{A}|\exp\left(-\frac{L}{200}\right) \leq H^2|\mathcal{A}|\exp\left(-\frac{L}{200}\right) \lesssim H^2|\mathcal{A}|N^{-9}.$$

## E.5   The case of $C^\pi \in [1, 2)$

In this section, we present an attempt in obtaining tight bounds on the LCB algorithm for episodic MDPs in the regime $C^\pi \in [1, 2)$. We start with a decomposition similar to the one given in (65).

$$\mathbb{E}_\mathcal{D}\left[\sum_s \rho(s)[V_1^\pi(s) - V^{\hat{\pi}}(s)]\right] = \mathbb{E}_\mathcal{D}\left[\sum_s \rho(s)[V_1^\pi(s) - V^{\hat{\pi}}(s)] \, \mathbb{1}\{\mathcal{E}_{\text{EMDP}}\}\right] =: T_1$$

$$+ \mathbb{E}_\mathcal{D}\left[\sum_s \rho(s)[V_1^\pi(s) - V^{\hat{\pi}}(s)] \, \mathbb{1}\{\mathcal{E}_{\text{EMDP}}^c\}\right] =: T_2.$$

An upper bound on the term $T_2$ is already proven in (66b). We follow a different route for bounding the term $T_1$. For any state $s \in \mathcal{S}$, define

$$\bar{\mu}(s) := \sum_{a \neq \pi(s)} \mu(s, a) \tag{71}$$

to be the total mass on actions not equal to the target policy $\pi(s)$. Consider the following set:

$$\mathcal{B} := \{s \mid \mu(s, \pi(s)) \leq 9\bar{\mu}(s)\}. \tag{72}$$

The set $\mathcal{B}$ includes the states for which the expert action is drawn more frequently under the data distribution. We then decompose $T_1$ based on whether state $s$ belongs to $\mathcal{B}$

$$T_2 = \mathbb{E}_{\mathcal{D}}\left[\sum_{s \in \mathcal{B}} \rho(s)[V_1^\pi(s) - V^{\hat{\pi}}(s)]\, \mathbb{1}\{\mathcal{E}_{\text{EMDP}}\}\right] =: \beta_1 \tag{73}$$

$$+ \mathbb{E}_{\mathcal{D}}\left[\sum_{s \notin \mathcal{B}} \rho(s)[V_1^\pi(s) - V^{\hat{\pi}}(s)]\, \mathbb{1}\{\mathcal{E}_{\text{EMDP}}\}\right] =: \beta_2. \tag{74}$$

We prove the following bound on $\beta_1$:

$$\beta_1 \leq 136H^2\sqrt{\frac{S(C^\pi - 1)L}{N}}. \tag{75}$$

We *conjecture* that $\beta_2$ is bounded similarly:

$$\beta_2 \lesssim H^2\sqrt{\frac{S(C^\pi - 1)L}{N}}. \tag{76}$$

We demonstrate our conjecture on $\beta_2$ in a special episodic MDP case with $H = 3, |\mathcal{S}_h| = 2$, and $|\mathcal{A}| = 2$ in Appendix E.6.

**Proof of the bound (75) on $\beta_1$.** By Lemma 10, it follows that

$$\beta_1 = \mathbb{E}_{\mathcal{D}}[\sum_{s \in \mathcal{B}} \rho(s)[V^\pi(s) - V^{\hat{\pi}}(s)]\, \mathbb{1}\{\mathcal{E}_{\text{EMDP}}\}]$$

$$\leq 2H \sum_{s \in \mathcal{B}} d^\pi(s, \pi(s))\, \mathbb{E}_{\mathcal{D}}[b(s, \pi(s))]$$

$$\leq 2H \sum_{s \in \mathcal{B}} d^\pi(s, \pi(s)) H\, \mathbb{E}_{\mathcal{D}}\left[\sqrt{\frac{L}{N(s, \pi(s)) \vee 1}}\right]$$

$$\leq 32H^2 \sum_{s \in \mathcal{B}} d^\pi(s, \pi(s))\left[\sqrt{\frac{L}{N\mu(s, \pi(s))}}\right]$$

In the first inequality, we substituted the definition of penalty and the second inequality arises from Lemma 14 with $c_{1/2} \leq 16$. We then apply the concentrability assumption to bound $d^\pi(s, \pi(s)) \leq$

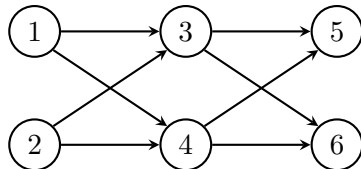

Figure 3: An episodic MDP with $H = 3$, two states per level, and two actions $\mathcal{A} = \{1, 2\}$ available from every state. The rewards are assumed to be deterministic and bounded. Action 1 is assumed to be optimal in all states and that $\mu(s, 1) \geq 9\mu(s, 2)$.

$C^\pi \mu(s, \pi(s))$ and thereby conclude

$$\beta_1 \leq 32H^2 \sum_{s \in \mathcal{B}} C^\pi \mu(s, \pi(s)) \left[ \sqrt{\frac{L}{N\mu(s, \pi(s))}} \right]$$

$$= 32C^\pi H^2 \sqrt{\frac{L}{N}} \sum_{s \in \mathcal{B}} \sqrt{\mu(s, \pi(s))}$$

$$\leq 32C^\pi H^2 \sqrt{\frac{LS}{N}} \sqrt{\sum_{s \in \mathcal{B}} \mu(s, \pi(s))},$$

where the last line is due to Cauchy-Schwarz inequality. We continue the bound relying on the definition of $\mathcal{B}$

$$\beta_1 \leq 32C^\pi H^2 \sqrt{\frac{LS}{N}} \sqrt{\sum_{s} \mu(s, \pi(s)) \, \mathbb{1}\{\mu(s, \pi(s)) \leq 9\bar{\mu}(s)\}} \leq 32C^\pi H^2 \sqrt{\frac{LS}{N}} \sqrt{\sum_{s} 9\bar{\mu}(s)}. \quad (77)$$

It is easy to check that the concentrability assumption implies the following bound on the total mass over the actions not equal to $\pi(s)$

$$\sum_{s} \bar{\mu} \leq \frac{C^\pi - 1}{C^\pi}.$$

Substituting the above bound to (77) and bounding $C^\pi \leq 2$ yields

$$\beta_1 \leq 136H^2 \sqrt{\frac{S(C^\pi - 1)L}{N}}.$$

## E.6   Analysis of LCB for a simple episodic MDP

We consider an episodic MDP with $H = 3$, $\mathcal{S}_1 = \{1, 2\}$, $\mathcal{S}_2 = \{3, 4\}$, $\mathcal{S}_3 = \{5, 6\}$, and $\mathcal{A} = \{1, 2\}$, where we assume without loss of generality that action 1 is optimal in all states. We are interested in bounding the $\beta_2$ term defined in (74) when $C^\pi \in [1, 2)$:

$$\beta_2 = \mathbb{E}_{\mathcal{D}} \left[ \sum_{s:\mu(s, \pi^\star(s)) \geq 9\bar{\mu}(s)} \rho(s)[V_1^\pi(s) - V_1^{\hat{\pi}}(s)] \, \mathbb{1}\{\mathcal{E}_{\mathrm{EMDP}}\} \right]. \quad (78)$$

Note that $\beta_2$ captures sub-optimality in states for which $\mu(s, \pi(s)) > 9\bar{\mu}(s)$. To illustrate the key ideas and avoid clutter, we consider the following setting:

1. Competing with the optimal policy $\pi(s) = \pi^\star(s) = 1$ and thus the concentrability assumption $d^\star(s,a) \leq C^\star \mu(s,a)$ for all $s \in \mathcal{S}, a \in \mathcal{A}$;

2. $\mu(s,1) \geq 9\mu(s,2)$ for all $s \in \mathcal{S}$;

3. $N(s,a) = N\mu(s,a) \geq 1$ for all $s \in \mathcal{S}, a \in \mathcal{A}$.

4. We assume that the rewards are deterministic and consider an implementation of Algorithm 3 with deterministic rewards. In particular, at level $H$ this implementation of VI-LCB sets $\hat{Q}_H$ according to

$$\hat{Q}_H(s,a) = \begin{cases} 0 & N(s,a) = 0; \\ r(s,a) & N(s,a) \geq 1. \end{cases}$$

**Outline of the proof.** Let us first give an outline for the sub-optimality analysis of the episodic VI-LCB Algorithm 3 in this example. We begin by showing that the concentrability assumption in conjunction with $\mu(s,1) \geq 9\mu(s,2)$ dictates certain bounds on the penalties. Afterward, we argue that the episodic VI-LCB algorithm finds the optimal policy at levels 2 and 3 with high probability. This result allows writing the sub-optimality as an expectation over the product of the gap $g_1(s) = Q_1^\star(s,1) - Q_1^\star(s,2)$ and the probability that the agent chooses the wrong action, i.e., $\mathbb{P}(\hat{\pi}(s) \neq 1)$. Consequently, if for state $s$ the gap $g_1(s)$ is small, the sub-optimality incurred by that state is also small. On the other hand, when the gap is large, we prove via Hoeffding's inequality that $\mathbb{P}(\hat{\pi}(s) \neq 1)$ is negligible.

**Bounds on penalties.** The setting introduced above dictates the following bounds on penalties

$$b_h(s,2) - b_h(s,1) \geq \frac{1}{3} b_h(s,2) + b_h(s,1), \tag{79a}$$

$$3\sqrt{\frac{LC^\star}{N(\bar{d}^\star(s,1) + C^\star - 1)}} \leq b_h(s,1) \leq 3\sqrt{\frac{LC^\star}{N\bar{d}^\star(s,1)}}, \tag{79b}$$

whose proofs can be found at the end of this subsection.

**VI-LCB policy in each level.** The main idea for a tight sub-optimality bound is to directly compare $\hat{Q}_h(s,1)$ to $\hat{Q}_h(s,2)$ at every level. Specifically, we first determine the conditions under which $\mathbb{E}[\hat{Q}_h(s,1) - \hat{Q}_h(s,2)] > 0$ and then show $\hat{Q}_h(s,1) > \hat{Q}_h(s,2)$ with high probability via a concentration argument. It turns out that these conditions depend on the value of the sub-optimality gap associated with a state defined as

$$g_h(s) := Q_h^\star(s,1) - Q_h^\star(s,2) \geq 0 \quad \forall s \in \mathcal{S}, \forall h \in \{1,2,3\}. \tag{80}$$

We start the analysis at level 3 going backwards to level 1.

- **Level 3.** Since $N(s,a) \geq 1$ and the rewards are deterministic, the value function computed by VI-LCB algorithm is equal to $V_3^\star$ and action 1 is selected for both states 5 and 6, i.e.,

$$\hat{V}_3 = V_3^\star. \tag{81}$$

- **Level 2.** We first show that $\hat{Q}_2(s,1)$ is greater than $\hat{Q}_2(s,2)$ in expectation

$$\mathbb{E}[\hat{Q}_2(s,1) - \hat{Q}_2(s,2)] = \mathbb{E}[r(s,1) - b_2(s,1) + \hat{P}_{s,1} \cdot V_3^\star - r(s,2) + b_2(s,2) - \hat{P}_{s,2} \cdot V_3^\star]$$
$$= b_2(s,2) - b_2(s,1) + g_2(s)$$
$$\geq \frac{1}{3}b_2(s,2) + b_2(s,1) + g_2(s) \geq \frac{1}{3}b_2(s,2) \geq 0, \tag{82}$$

where we used the bound on $b_2(s,2) - b_2(s,1)$ given in (79a). By the concentration inequality in Lemma 13 we then show $\hat{Q}_2(s,1) \geq \hat{Q}_2(s,2)$ with high probability:

$$\mathbb{P}(\hat{Q}_2(s,2) - \hat{Q}_2(s,1) \geq 0) \leq \exp\left(-6\frac{N(s,1)N(s,2)\,\mathbb{E}^2[\hat{Q}_2(s,1) - \hat{Q}_2(s,2)]}{N(s,1) + N(s,2)}\right)$$
$$\leq \exp\left(-1.8N(s,2)\left(\frac{1}{3}\right)^2 b_2^2(s,2)\right)$$
$$= \exp\left(-0.8N(s,2)\frac{L}{N(s,2)}\right) \lesssim \frac{1}{N^{160}}, \tag{83}$$

where in the second inequality we used $N(s,2) \leq 1/9N(s,1)$ as well as the bound given in (82) and the last inequality holds for $c_1 \geq 1$ and $\delta = 1/N$.

- **Level 1.** Define the following event

$$\mathcal{E}_o = \{\hat{\pi}(s) = 1, \ \forall s \in \mathcal{S}_2\}, \tag{84}$$

which refers to the event that action 1 is chosen for all states at level 2. Conditioned on $\mathcal{E}_o$, the Q-function computed by VI-LCB in level 1 is given by

$$\hat{Q}_1(s,a) = r(s,a) - b_1(s,a) + \hat{P}(3 \mid s,a)[r(3,1) - b_2(3,1) + \hat{P}_{3,1}V_3^\star]$$
$$\qquad\qquad + \hat{P}(4 \mid s,a)[r(4,1) - b_2(4,1) + \hat{P}_{4,1}V_3^\star]. \qquad \forall s \in \mathcal{S}_1, a \in \mathcal{A}.$$

Taking the expectation with respect to the data randomness, one has for any $s \in \mathcal{B}$ that

$$\mathbb{E}[\hat{Q}_1(s,1) - \hat{Q}_1(s,2)]$$
$$= [b_1(s,2) - b_1(s,1)] + [P(3|s,2) - P(3|s,1)]b_2(3,1) + [P(4|s,2) - P(4|s,1)]b_2(4,1) + g_1(s)$$
$$= [b_1(s,2) - b_1(s,1)] + [P(3|s,1) - P(3|s,2)][b_2(4,1) - b_2(3,1)] + g_1(s),$$

where the last equation uses $P(3 \mid s,a) = 1 - P(4 \mid s,a)$. We continue the analysis assuming that $p := P(3 \mid s,1) - P(3 \mid s,2) \geq 0$; the other case can be shown similarly. Using $p \geq 0$ and $b_2(4,1) \geq 0$ together with the penalty bound of (79a), we see that

$$\mathbb{E}[\hat{Q}_1(s,1) - \hat{Q}_1(s,2)] \geq \frac{1}{3}b_1(s,2) + b_1(s,1) - pb_2(3,1) + g_1(s).$$

We proceed by applying (79b) on $b_1(s,1)$ and $b_1(3,1)$

$$\mathbb{E}[\hat{Q}_1(s,1) - \hat{Q}_1(s,2)] \geq \frac{1}{3}b_1(s,2) + 3\sqrt{\frac{LC^\star}{N(d^\star(s,1) + C^\star - 1)}} - 3p\sqrt{\frac{LC^\star}{Nd^\star(3,1)}} + g_1(s). \tag{85}$$

Note that $d^\star(s,1) = \rho(s)/3$ and $3d^\star(3,1) = \rho(s)P(3|s,1) + \rho(2)P(3|s,2) \geq \rho(s)P(3|s,1) \geq \rho(s)p$. Substituting these quantities into (85), we obtain

$$\mathbb{E}[\hat{Q}_1(s,1) - \hat{Q}_1(s,2)] \geq \frac{1}{3}b_1(s,2) + 3\sqrt{\frac{LC^\star}{N(\rho(s)/3 + C^\star - 1)}} - 3p\sqrt{\frac{LC^\star}{N\rho(s)p/3}} + g_1(s)$$

$$\geq \frac{1}{3}b_1(s,2) + 3\sqrt{\frac{LC^\star}{N(\rho(s)/3 + C^\star - 1)}} - 3\sqrt{\frac{LC^\star}{N\rho(s)/3}} + g_1(s),$$

where the last inequality uses $p \leq 1$. Observe that

$$\frac{1}{\sqrt{\rho(s)/3}} - \frac{1}{\sqrt{\rho(s)/3 + C^\star - 1}} = \frac{\sqrt{\rho/3 + C^\star - 1} - \sqrt{\rho/3}}{\sqrt{\rho(s)/3(\rho(s)/3 + C^\star - 1)}} \leq 3\frac{\sqrt{C^\star - 1}}{\rho(s)}.$$

This implies

$$\rho(s)g_1(s) \geq 9\sqrt{\frac{2(C^\star - 1)L}{N}} \quad \Rightarrow \quad \mathbb{E}[\hat{Q}_1(s,1) - \hat{Q}_1(s,2)] \geq \frac{1}{3}b_1(s,2). \tag{86}$$

Then, a similar argument to (83) proves that $\hat{Q}(s,1) > \hat{Q}(s,2)$ with high probability:

$$\mathbb{P}(\hat{Q}_1(s,2) - \hat{Q}_1(s,1) \geq 0) \lesssim \frac{1}{N^{160}}. \tag{87}$$

**Sub-optimality bound.** We are now ready to compute the sub-optimality. Decompose the sub-optimality based on whether event $\mathcal{E}_o$ defined in (84) has occurred and use the fact that we assumed $\mu(s,1) \geq 9\mu(s,2)$ for all $s \in \mathcal{S}$

$$\beta_2 = \mathbb{E}_\mathcal{D}\left[\sum_{s:\mu(s,\pi^\star(s))\geq 9\bar{\mu}(s)} \rho(s)[V_1^\pi(s) - V_1^{\hat{\pi}}(s)]\,\mathbb{1}\{\mathcal{E}_{\mathrm{EMDP}}\}\right]$$

$$\leq \mathbb{E}_{\mathcal{D},\rho}\left[[V^\star(s) - V^{\hat{\pi}}(s)]\,\mathbb{1}\{\mathcal{E}_o\}\right] + \mathbb{E}_{\mathcal{D},\rho}\left[[V^\star(s) - V^{\hat{\pi}}(s)]\,\mathbb{1}\{\mathcal{E}_o^c\}\right]$$

$$\lesssim \mathbb{E}_{\mathcal{D},\rho}\left[[V^\star(s) - V^{\hat{\pi}}(s)]\,\mathbb{1}\{\mathcal{E}_o\}\right] + \frac{3}{N^{160}}.$$

Here, the second line is by $\mathbb{1}\{\mathcal{E}_{\mathrm{EMDP}}\} \leq 1$ and the last line follows from $V^\star(s) - V^{\hat{\pi}}(s) \leq 3$ and the probability of the complement event $\mathcal{E}_o^c$ given in (83).

Conditioned on the event $\mathcal{E}_o$, LCB-VI algorithm chooses the optimal action from every state at levels 2 and 3 and hence $V_2^{\hat{\pi}} = V_2^\star$ and we get

$$\mathbb{E}_{\mathcal{D},\rho}\left[[V^\star(s) - V^{\hat{\pi}}(s)]\,\mathbb{1}\{\mathcal{E}_o\}\right]$$

$$= \sum_s \rho(s)\,\mathbb{E}_\mathcal{D}[[Q^\star(s,1) - Q^{\hat{\pi}}(s,\hat{\pi}(s))]\,\mathbb{1}\{\mathcal{E}_o\}]$$

$$= \sum_s \rho(s)\,\mathbb{E}_\mathcal{D}[r(s,1) + P_{s,1} \cdot V_2^\star - r(s,\hat{\pi}(s)) - P_{s,\hat{\pi}(s)} \cdot V_2^\star]$$

$$= \sum_s \rho(s)\,\mathbb{E}_\mathcal{D}[(r(s,1) + P_{s,1} \cdot V_2^\star - r(s,2) - P_{s,2} \cdot V_2^\star)\,\mathbb{1}\{\hat{\pi}(s) \neq 1\}].$$

By definition, we have $g_1(s) = r(s,1) + P_{s,1} \cdot V_2^\star - r(s,2) - P_{s,2} \cdot V_2^\star$. Therefore,

$$\mathbb{E}_{\mathcal{D},\rho}\left[V^\star(s) - V^{\hat{\pi}}(s)\,\mathbb{1}\{\mathcal{E}_o\}\right] \leq \sum_s \rho(s)g(s)\,\mathbb{E}_{\mathcal{D}}[\mathbb{1}\{\hat{\pi}(s) \neq 1\}]$$

$$= \sum_s \rho(s)g(s)\,\mathbb{P}(\hat{Q}(s,2) - \hat{Q}(s,1) \geq 0).$$

We decompose the sub-optimality based on whether $\rho(s)g_1(s)$ is large

$$\mathbb{E}_{\mathcal{D}}[J(\pi^\star) - J(\hat{\pi})] \leq \sum_s \rho(s)g(s)\,\mathbb{P}(\hat{Q}(s,2) - \hat{Q}(s,1) \geq 0)\,\mathbb{1}\left\{\rho(s)g(s) \leq 9\sqrt{\frac{2(C^\star-1)L}{N}}\right\} =: \tau_1$$

$$+ \sum_s \rho(s)g_1(s)\,\mathbb{P}(\hat{Q}(s,2) - \hat{Q}(s,1) \geq 0)\,\mathbb{1}\left\{\rho(s)g_1(s) > 9\sqrt{\frac{2(C^\star-1)L}{N}}\right\} =: \tau_2$$

$$+ \frac{3}{N^{160}}.$$

The first term is bounded by

$$\tau_1 \leq \sum_s 9\sqrt{\frac{2(C^\star-1)L}{N}} = 18\sqrt{\frac{2(C^\star-1)L}{N}}.$$

The second term is bounded using (87)

$$\tau_2 \lesssim \frac{3}{N^{160}}.$$

Combining the bounds yields the following sub-optimality bound

$$\beta_2 \lesssim \sqrt{\frac{(C^\star-1)L}{N}} + \frac{1}{N^{160}}.$$

**Proof of inequality** (79a). From $\mu(s,1) \geq 9\mu(s,2)$, one has $N(s,1) \geq 9N(s,2)$ implying $b_h(s,2) \geq 3b_h(s,1)$. Therefore, we conclude that

$$b_h(s,2) - b_h(s,1) = \frac{1}{2}(b_h(s,2) - b_h(s,1)) + \frac{1}{2}(b_h(s,2) - b_h(s,1)) \geq \frac{1}{3}b_h(s,2) + b_h(s,1).$$

**Proof of inequality** (79b). The concentrability assumption implies the following bound on $\mu(s,1)$

$$\frac{\bar{d}(s,1)}{C^\star} \leq \mu(s,1) \leq \frac{\bar{d}(s,1)}{C^\star} + 1 - \frac{1}{C^\star},$$

The upper bound is based on the fact that the probability mass of at least $1/C^\star$ is distributed on the optimal actions with a remaining mass of $1 - 1/C^\star$. Applying the above bounds to $b_h(s,1)$, gives

$$3\sqrt{\frac{LC^\star}{N(\bar{d}(s,1) + C^\star - 1)}} \leq b_h(s,1) = 3\sqrt{\frac{L}{N\mu(s,1)}} \leq 3\sqrt{\frac{LC^\star}{N\bar{d}^\star(s,1)}}.$$

# F   Auxiliary lemmas

This section collects a few auxiliary lemmas that are useful in the analysis of LCB.

We begin with a simple extension of the conventional Hoeffding bound to the two-sample case.

**Lemma 13.** *Let $X_1, \ldots, X_n$ be i.i.d. in range $[0, 1]$ with average $\mathbb{E}[X]$ and $Y_1, \ldots, Y_m$ be i.i.d. in range $[0, 1]$ with average $\mathbb{E}[Y]$. Further assume that $\{X_i\}$ and $\{Y_j\}$ are independent. Then for any $\epsilon$ such that $\epsilon + \mathbb{E}[Y] - \mathbb{E}[X] \geq 0$, we have*

$$\mathbb{P}\left(\frac{1}{n}\sum_i X_i - \frac{1}{m}\sum_j Y_j > \epsilon\right) \leq \exp\left(-2\frac{(mn)(\epsilon + \mathbb{E}[Y] - \mathbb{E}[X])^2}{m + n}\right).$$

*Proof.* It is easily seen that

$$\mathbb{P}\left(\sum_{i=1}^n mX_i - \sum_{j=1}^m nY_j > mn\epsilon\right)$$

$$= \mathbb{P}\left(\sum_{i=1}^n (mX_i - m\,\mathbb{E}[X]) - \sum_{j=1}^m (nY_j - \mathbb{E}[Y]) > mn(\epsilon + \mathbb{E}[Y] - \mathbb{E}[X])\right)$$

$$\leq \exp\left(-2\frac{(mn)^2(\epsilon + \mathbb{E}[Y] - \mathbb{E}[X])^2}{nm(m + n)}\right)$$

$$= \exp\left(-2\frac{(mn)(\epsilon + \mathbb{E}[Y] - \mathbb{E}[X])^2}{m + n}\right),$$

where the inequality is based on Hoeffding's inequality on independent random variables. $\qquad \square$

The next lemma provides useful bounds for the inverse moments of a binomial random variable.

**Lemma 14** (Bound on binomial inverse moments). *Let $n \sim \text{Binomial}(N, p)$. For any $k \geq 0$, there exists a constant $c_k$ depending only on $k$ such that*

$$\mathbb{E}\left[\frac{1}{(n \vee 1)^k}\right] \leq \frac{c_k}{(Np)^k},$$

*where $c_k = 1 + k2^{k+1} + k^{k+1} + k\left(\frac{16(k+1)}{e}\right)^{k+1}$.*

*Proof.* The proof is adapted from that of Lemma 21 in Jiao et al. (2018).

To begin with, when $p \leq 1/N$, the statement is clearly true for $c_k = 1$. Hence we focus on the case when $p > 1/N$. We define a useful helper function $g_N(p)$ to be

$$g_N(p) := \begin{cases} \frac{1}{p^k}, & p \geq \frac{1}{N}, \\ N^k - kN^{k+1}(p - \frac{1}{N}), & 0 \leq p < \frac{1}{N}. \end{cases}$$

Further denote $\hat{p} := n/N$. The proof relies heavily on the following decomposition, which is an direct application of the triangle inequality:

$$\mathbb{E}\left[\frac{N^k}{(n \vee 1)^k}\right] \leq \left|\mathbb{E}\left[\frac{N^k}{(n \vee 1)^k} - g_N(\hat{p})\right]\right| + |\mathbb{E}[g_N(p) - g_N(\hat{p})]| + g_N(p). \tag{88}$$

This motivates us to take a closer look at the helper function $g_N(p)$. Simple algebra reveals that

$$g_N(p) \le \frac{1}{p^k} \quad \text{and} \quad g_N(\hat{p}) - \frac{N^k}{(n \vee 1)^k} = kN^k \mathbb{1}\{\hat{p} = 0\}.$$

Substitute these two facts back into the decomposition (88) to reach

$$\mathbb{E}\left[\frac{N^k}{(n \vee 1)^k}\right] \le kN^k(1-p)^N + \frac{1}{p^k} + |\mathbb{E}[g_N(p) - g_N(\hat{p})]|.$$

It remains to bound the term $|\mathbb{E}[(g_N(p) - g_N(\hat{p}))^2]|$. To this goal, one has

$$|\mathbb{E}[(g_N(p) - g_N(\hat{p}))^2]| \le |\mathbb{E}[(g_N(p) - g_N(\hat{p}))^2 1\{\hat{p} \ge p/2\}]| + |\mathbb{E}[(g_N(p) - g_N(\hat{p}))^2 1\{\hat{p} \ge p/2\}]|$$

$$\overset{(i)}{\le} \sup_{\xi \ge p/2} |g_N'(\xi)|^2 \mathbb{E}[(p - \hat{p})^2] + \sup_{\xi > 0} |g_N'(\xi)|^2 p^2 \, \mathbb{P}(\hat{p} \le p/2)$$

$$\overset{(ii)}{\le} \frac{k^2}{(p/2)^{2k+2}} \frac{p(1-p)}{N} + k^2 N^{2k+2} p^2 e^{-Np/8}.$$

Here the inequality (i) follows from the mean value theorem, and the last one (ii) uses the derivative calculation as well as the tail bound for binomial random variables; see e.g., Exercise 4.7 in Mitzenmacher and Upfal (2017). As a result, we conclude that

$$\mathbb{E}\left[\frac{N^k}{(n \vee 1)^k}\right] \le kN^k(1-p)^N + \frac{1}{p^k} + \sqrt{\mathbb{E}[(g_N(p) - g_N(\hat{p}))^2]}$$

$$\le kN^k(1-p)^N + \frac{1}{p^k} + \frac{k}{(p/2)^{k+1}} \sqrt{\frac{p(1-p)}{N}} + kN^{k+1}pe^{-Np/16}$$

$$\le kN^k(1-p)^N + \frac{1}{p^k} + \frac{k2^{k+1}}{p^k} + kN^{k+1}pe^{-Np/16},$$

where the last inequality holds since $p \ge 1/N$. Consequently, we have

$$\mathbb{E}\left[\frac{(Np)^k}{(n \vee 1)^k}\right] \le 1 + k2^{k+1} + k(Np)^k(1-p)^N + k(Np)^{k+1}e^{-Np/16}.$$

Note that the following two bounds hold:

$$\max_p k(Np)^k(1-p)^N \le k\left(N\frac{k}{N+k}\right)^k\left(1 - \frac{k}{k+N}\right)^N \le k^{k+1},$$

$$(Np)^k e^{-Np/16} \le \left(\frac{16k}{e}\right)^k.$$

The proof is now completed. $\qquad\square$

The last lemma, due to Gilbert and Varshamov (Gilbert, 1952; Varshamov, 1957), is useful for constructing hard instances in various minimax lower bounds.

**Lemma 15.** *There exists a subset $\mathcal{V}$ of $\{-1,1\}^S$ such that (1) $|\mathcal{V}| \ge \exp(S/8)$ and (2) for any $v_i, v_j \in \mathcal{V}$, $v_i \ne v_j$, one has $\|v_i - v_j\|_1 \ge \frac{S}{2}$.*