# OpenReview forum: "Bridging Offline Reinforcement Learning and Imitation Learning: A Tale of Pessimism"
_NeurIPS.cc/2021/Conference — NeurIPS 2021 Poster_

### Official Review · Reviewer_M9C5 · 2021-07-12

**Rating:** 8
**Confidence:** 3

**Summary:**

This work presents a simple algorithm for offline reinforcement learning based on optimizing the sum of lower confidence bounds of rewards for each transition (where the confidence bound is constricted using the offline dataset of transitions).  Theoretical bounds are given on the suboptimality of the resulting policy, based on the quantity $C^*$, which bounds the proportion of non-optimal transitions in the dataset.  Conditional on there being a finite ratio of non-optimal to optimal actions in each state, the presented algorithm has good performance in both the "expert data" case, in which the behavioral policy that generated the data is close to optimal, and the "uniform coverage" case, in which the behavioral policy covers all possible transitions.


**Limitations And Societal Impact:**

The authors have done a good job of specifying the technical limitations of their work.  I do not see any societal impacts that require further comment.

**Main Review:**

Overall, the paper is clearly written, and the problem is important.  I did not carefully check the proofs, but clear intuitions are provided for the results.  The paper does a good job of focusing on the important details, and of expressing why the results are significant.

My main concern is with the novelty of the results; the following paragraph on p.3 is very worrying:

> A similar algorithm design has recently appeared in [16]. It turns out that such a simple algorithm—fully agnostic to the data composition—achieves almost optimal performance in multi-armed bandits and MDPs, and optimally solves the offline learning problem in contextual bandits.

This sounds like more or less exactly the results of the current paper.  Could the authors comment on the differences between [16] and this work?

I am going to rate the paper lower than I otherwise would in light of this question about novelty, but I am willing to increase my rating if the novelty question is addressed to my satisfaction.

#### Minor comments:

* p.2: "Assuming a finite $C^*$ is the weakest concentrability requirement [39, 12, 48] that is currently enjoyed only by some online algorithms such as CPI [17]": It's not clear what this means.

* p.6 L227: The subscripts on the expectation operator seem inconsistent; $\mathcal{D}$ is a random variable, and $\rho$ is a distribution, so what do these signify?

* p.7 L248-255: These paragraphs are very helpful and give exactly the right amount of detail.

* alg.2: Are rewards assumed to have a maximum magnitude of 1 here?  I can't make sense of $V_{max} = (1-\gamma)^{-1}$ otherwise.

* p.5: Should the inequality in equation (3) be $\lesssim$?

* p.7: Similarly, should the inequality in prop.3 actually be $\gtrsim$?

* p.9 L323: $a$ is free in the definition of $g(s)$; what is it meant to bind to?

**Time Spent Reviewing:**

2.5

---

> ### Author Response · Authors · 2021-08-08
> **Response**
>
> We thank the reviewer for the time spent reading our paper, their thorough review, and thoughtful comments. We respond to the reviewer's comments below.
>
> ### Immediate clarification of the sentences in the introduction
>
> Regarding the following sentences in our paper
>
> > A similar algorithm design has recently appeared in [16]. It turns out that such a simple algorithm—fully agnostic to the data composition—achieves almost optimal performance in multi-armed bandits and MDPs, and optimally solves the offline learning problem in contextual bandits,
>
> we think our writing might have caused confusion. We would like to clarify that *only* our algorithmic design is similar to [16] with some important differences which we point out below. However, the second sentence refers to our contributions and not those in [16]. Indeed, the characterization of the offline data coverage, analysis of adaptive optimality of LCB, and connections to imitation learning are not studied in [16] or other prior work.
>
>
> ### Novelty and further comparison with [16]
>
> **1. Characterization of offline data coverage and connections to imitation learning.** We propose the first characterization of the offline dataset via the single-policy concentrability coefficient $C^\star$. Our formulation naturally unifies imitation learning and vanilla offline RL and sets the stage for studying the adaptive optimality of conservative algorithms with respect to the data coverage. [16] does not provide any characterization of offline dataset nor unifies vanilla offline RL and imitation learning. Furthermore, in the past, imitation learning has been usually studied using reduction methods (Ross and Bagnell, 2010) whereas offline RL has been studied using statistical rates. To further unify the two, we study the statistical rates of both methods.
>
> **2. Adaptive optimality of LCB and tight bounds for near-expert datasets.** [16] does not study adaptive optimality of the LCB algorithm with respect to data coverage $C^\star$. In [16], the sub-optimality is bounded by an expectation over the uncertainty quantifier, which only leads to the loose bound of $1/\sqrt{N}$ and does not recover the tight rate of $1/N$ for near-expert datasets.
>
> **3. Novelty in proof techniques.** We use a novel sub-optimality decomposition technique to prove rates faster than $1/\sqrt{N}$ and show adaptive optimality in the CB setting. Such techniques are not used in [16] or other prior work. We refer to our response to Reviewer ivrh under the heading “Originality in proofs and high-level description” for further details.
>
> **4. Differences in algorithms.** Our VI-LCB algorithm has two additional components of data splitting and monotonic update that are not included in [16]. These two components result in removing an extra $S$ factor in the sample complexity.
>
> **5. Surprising results regarding adaptive optimality of LCB in bandits vs. contextual bandits with at least two contexts.**
> Our analysis reveals that LCB is not adaptively optimal in bandits (Theorem 3). However, in contextual bandits with at least two contexts, we prove that LCB is adaptively optimal (Theorems 4 and 5). This separation has not been reported in [16] or any prior work.
>
> **6. Lower bounds in terms of data coverage.**  In contextual bandits and MDPs, we prove information-theoretic lower bounds showing the dependency on $C^\star$ is of the form $C^\star - 1$, which results in a sub-optimality lower bound of $1/N$ when $C^\star \approx 1$ instead of $1/\sqrt{N}$.
>
> **7. Reducing the gap between upper and lower bounds.** [16] focuses on episodic MDPs in the function approximation setting and the authors instantiate their bounds in linear MDPs. When translated to the tabular setting, there is a gap of $SAH$ between their upper and lower bounds whereas in our episodic MDP analysis, the gap is only $H$.
>
> ### Response to the minor comments
>
> > "Assuming a finite $C^\star$ is the weakest concentrability requirement [39, 12, 48] that is currently enjoyed only by some online algorithms such as CPI [17]": It's not clear what this means.
>
>
> We mean that a definition of concentrability related to our single-policy concentrability has been used to characterize sample complexity of online RL algorithms such as CPI (Kakade and Langford, 2002) or some policy gradient methods (Agarwal et al., 2020). However, the sample complexity of previous offline RL algorithms often depends on *uniform* concentrability definitions, as discussed in lines 46-48.
>
> Scherrer, 2014 shows that there exists a hierarchy among existing concentrability definitions. This hierarchy (Figure 1 in Scherrer, 2014) has the following form: concentrability definition $A$ is considered better than $B$ when an algorithm that has a guarantee with respect to $A$ can be arbitrarily better than an algorithm that has a guarantee with respect to $B$. Among all existing concentrability definitions, Scherrer, 2014 shows that the single-policy concentrability is the best.
>
> > The subscripts on the expectation operator seem inconsistent; is a random variable, and is a distribution, so what do these signify?
>
>  In the paper, we use a distribution subscript on the expectation operator to denote $E_p [x] = E_{x \sim p} [x] = \int_{x \in \mathcal{X}} x dp$. We overload the notation and write $E_{\mathcal{D}}$ to denote an expectation taken with respect to the random draw of the dataset $\mathcal{D}$.
>
> > Are rewards assumed to have a maximum magnitude of 1 here?  I can't make sense of $V_{\max} = (1-\gamma)^{-1}$ otherwise.
>
>  We consider stochastic rewards with an expected reward function in $[0,1]$ (defined in Section 2.1).
>
> > Should the inequality in equation (3) be $\lesssim$?
>
>  No, this inequality uses Chernoff bound and holds exactly as proved in Appendix B.4.
>
> >  Similarly, should the inequality in prop 3 be $\gtrsim$?
>
>  No, this inequality is also exact as proved in Appendix C.3.
>
> > $a$ is free in the definition of $g(s)$; what is it meant to bind to?
>
>  We thank the reviewer for pointing this out. The gap should be denoted as $g(s,a) = Q^\star(s,\pi^\star(s)) - Q^\star(s, a)$ with a minimum gap defined as $g(s) = \min_{a \not =  \pi^\star(s)} g(s,a)$. This notation change does not affect our analysis in the example of Appendix E.6 as the MDP considered there only involves two actions per state.
>
>
> ### References:
>
> Ross, Stéphane, and Drew Bagnell. "Efficient reductions for imitation learning." Proceedings of the thirteenth international conference on artificial intelligence and statistics. JMLR Workshop and Conference Proceedings, 2010.
>
> Kakade, Sham, and John Langford. "Approximately optimal approximate reinforcement learning." In Proc. 19th International Conference on Machine Learning. 2002.
>
> Agarwal, Alekh, et al. "Optimality and approximation with policy gradient methods in Markov decision processes." Conference on Learning Theory. PMLR, 2020.
>
> Scherrer, Bruno. "Approximate policy iteration schemes: a comparison." International Conference on Machine Learning. PMLR, 2014.
>
> ____
> We hope that we have addressed the reviewer’s concerns. We would be happy to answer any questions and discuss further in case the reviewer believes there are any missing details.

---

> > ### Comment · Reviewer_M9C5 · 2021-08-23
> > **Review update**
> >
> > Thank you for your informative response.  My concerns about novelty are fully addressed and I will update my score to recommend acceptance.  I strongly recommend clarifying the quoted passage; even opening the second sentence with "As we show in this work, ..." or something similar would be helpful.

---

### Official Review · Reviewer_eXEz · 2021-07-16

**Rating:** 8
**Confidence:** 2

**Summary:**

The authors present a new offline RL framework that builds the gap between imitation learning (data are generated by experts) and vanilla offline RL (data are generated with large coverage). Under this framework, the authors consider the Lower Confidence Bound (LCB) algorithm and obtain adaptive optimality on Contextual Bandit problem, sub-optimality for Multi-arm Bandit problem and conjecturely adaptive optimality for MDP.

**Limitations And Societal Impact:**

Yes

**Main Review:**

Strength:
- The paper is very clear and well-written. Though many theorems are proposed, the paper is easy to follow by only listing the theorems and their intuitive explanations.
- The new framework of constructing bounds on single policy concentrability assumption is neat and clean: it covers the entire data spectrum and is applicable to all three cases (MAB, CB and MDP)

Weakness:
- The adaptive optimality of LCB on MDPs is not proved completely.




**Time Spent Reviewing:**

4

---

> ### Author Response · Authors · 2021-08-08
> **Response**
>
> We thank the reviewer for the time spent reviewing our work, their thorough review, and positive feedback. Regarding the adaptive optimality of LCB in the MDP setting, we proved our conjecture in an episodic MDP example in Appendix E.6 and explained the difficulty in proving the smooth transition from the $1/N$ rate to the $1/\sqrt{N}$ rate in Section 6. We plan to resolve this question in a future work.

---

### Official Review · Reviewer_9pjr · 2021-07-20

**Rating:** 6
**Confidence:** 4

**Summary:**

This paper aims to bridge the gap between offline reinforcement learning algorithms that are “vanilla” (which perform well in the “diverse data” regime) and ones based on imitation learning (which perform well in the “optimal data” regime). They propose and study an algorithm with pessimism that performs well both in the diverse data regime and in the optimal data regime, and prove bounds on the error of their approach.

**Limitations And Societal Impact:**

Yes.

**Main Review:**

Strengths
- Important problem
- Interesting theoretical analysis
- Upper and lower bounds

Weaknesses
- Doesn’t compare to MOPO, even though the algorithms are similar
- Motivation is model-based vs. model-free, but they are purely studying model-based approaches

This paper is studying an important problem. There are significant differences between the two classes of algorithms (vanilla vs. imitation learning), and understanding which of these approaches is ideal or how to combine them is an important research direction.

However, I’m not sure this paper really addresses this problem. My main concern is that their algorithm closely resembles the MOPO algorithm [51], which uses a pessimism term to penalize parts of the state space with high uncertainty. In particular, for finite MDPs, this pessimism term can be instantiated with rigorous bounds based on the state-action visitation counts, which is similar to how the pessimism term in this paper is defined. Furthermore, the MOPO paper includes an error guarantee that appears to be quite similar to the one achieved by the authors (in particular, by using the total variation norm on the transition probability estimates). Thus, it is not clear to me that the authors’ algorithm and analysis are novel.

A related issue is that in their related work, the authors claim that there is a gap between model-free and model-based approaches in terms of pessimism, and that their work aims to clarify this gap. However, since the authors’ paper is model-based, I’m not sure how it achieves this goal.

Finally, the authors do not provide any experimental results to validate their approach. Given that the performance tradeoffs are primarily of interest from an empirical point of view, I think experiments would be very helpful in clarifying this paper’s insights.



**Time Spent Reviewing:**

1

---

> ### Author Response · Authors · 2021-08-08
> **Response**
>
> We thank the reviewer for the time spent reviewing our paper and for their thoughtful feedback. We provide our response to the reviewer's comments below.
>
> ### On bridging vanilla offline RL and imitation learning
> Regarding the reviewer's comment:
>
> > There are significant differences between the two classes of algorithms (vanilla vs. imitation learning), and understanding which of these approaches is ideal or how to combine them is an important research direction. However, I’m not sure this paper really addresses this problem.
>
> We discuss how our work addresses this question below.
>
> 1. The first step that we take is formulating offline learning problems so that we can study imitation learning and vanilla offline RL under the same framework. We accomplish this by characterizing the offline data composition via single-policy concentrability $C^\star$. In prior work, imitation learning has been usually studied using reduction methods (Ross and Bagnell, 2010) whereas offline RL has been studied using statistical rates. To further unify the two, we study the statistical rates of both methods.
>
> 2. We compute tight upper bounds for our LCB algorithm with a specific design of penalty, without the knowledge of data composition. In contextual bandits and MDPs, we prove that when $C^\star \approx 1$, LCB achieves a $1/N$ sub-optimality, the same rate as imitation learning (Rajaraman et al. 2020). When $C^\star = 1 + \Omega(1)$, LCB achieves a $1/\sqrt{N}$ sub-optimality, the same rate as vanilla offline RL (e.g. Xie and Jiang, 2020). In contextual bandits, we also prove a smooth transition as $C^\star$ increases. Thus, we have answered the question on which of these approaches is ideal and how to combine them in the tabular setting and the answer is to use LCB to achieve the optimal rates in near-expert or uniform datasets.
>
>
> ### On novelty and comparison with MOPO
>
> We do not claim novelty on using pessimism as many forms of pessimism have been used in prior work such as auction design (Guo et al., 2019) and RL (Jin et al., 2021; Yin et al., 2021; Kumar et al. 2020; Buckman et al., 2020; Dai et al., 2020; Kidambi et al., 2020) including MOPO (Yu et al., 2020).
>
> As discussed in the introduction, our contributions are
>
> 1. A new formulation that unifies imitation learning and offline RL;
>
> 2. The study of adaptive optimality of pessimism and obtaining tight sub-optimality bounds;
>
> 3. Novel analysis techniques.
>
> We compare our work with MOPO in line 75. Below, we provide a detailed comparison to MOPO and will include further discussion in the camera-ready version per reviewer's suggestion.
>
> **1. Differences in algorithm design.** There are several differences between MOPO and our VI-LCB algorithm, highlighted below.
>
> * **Different uncertainty characterization.** As pointed out by the reviewer, MOPO characterizes uncertainty by bounding the TV distance between true and learned transitions. However, we characterize the uncertainty of the Q-functions; see e.g. line 100 and equation (27) in Appendix D.2. This difference is critical in achieving a minimax optimal dependency on the state size $S$. Bounding the TV distance suffers from an additional factor of $S$ in the sample complexity.
>
> * **Data splitting and monotonic updates.** VI-LCB uses data splitting and monotonic update which are important for obtaining a tight dependency on $S$. Such steps are not included in MOPO.
>
> **2. Novelty in the analysis.** Our analysis is novel compared to MOPO as explained below.
>
> * The guarantee presented in Theorem 4.4 of MOPO depends on the expectation of penalty w.r.t. the occupancy measure $d^\pi_{\hat{M}}$ in the *approximate* MDP  $\hat M$ and not the true MDP. However, our result depends on the true MDP $M^\star$ (Lemma 4). This difference is important as it allows for using single-policy concentrability $\frac{d^\pi_{M^\star}(s,a)}{\mu(s,a)} \leq C^\pi$, which naturally captures imitation learning (with $\pi$ being the expert policy) when $\mu(s,a) = d^\pi_{M^\star}(s,a)$. In contrast, using $d^\pi_{\hat{M}}$ in the $C^\star$ definition does not recover imitation learning.
>
>
> * More importantly, the analysis in MOPO cannot recover the tight $1/N$ rate in near-expert datasets (even when instantiated with our penalty design) or prove/disprove adaptive optimality of pessimism. This is because first, MOPO does not instantiate the uncertainty quantifier to characterize statistical rates and second, Theorem 4.4 in MOPO only provides a loose bound and will not give a rate tighter than $1/\sqrt{N}$. To prove sharp bounds, we develop a novel policy decomposition technique that characterizes the probability of choosing a wrong action. We refer to our response to Reviewer ivrh under the heading "Originality in proofs and high-level description", Section 6 in the paper, and our decomposition technique in Appendix C.1 for further details.
>
> **3. Offline RL formulation and adaptive optimality.** As mentioned earlier, we presented a new formulation to unify offline RL and imitation learning and study the adaptive optimality of the LCB algorithm. These problems are not studied in MOPO or other prior work.
>
> **4. Several new results on pessimism.**
> We also prove several new results that are not presented in MOPO or other prior work.
>
> * In contextual bandits and MDPs, we prove information-theoretic lower bounds showing the dependency on $C^\star$ is of the form $C^\star - 1$, which results in a lower bound of $1/N$ when $C^\star \approx 1$ instead of $1/\sqrt{N}$.
> * We prove the failure of the best empirical arm and theoretically compare LCB with the most played arm in bandits.
> * We prove a surprising result that LCB cannot be adaptively optimal in bandits but it is adaptively optimal in contextual bandits with at least two contexts.
>
>
> ### Model-based vs. model-free
> Regarding the reviewer's comment:
>
> >A related issue is that in their related work, the authors claim that there is a gap between model-free and model-based approaches in terms of pessimism, and that their work aims to clarify this gap.
>
> we emphasize that our work is not focused on understanding the differences between pessimistic model-based and model-free algorithms. Rather, we focus on the two questions presented in our introduction:
>
> 1. Is there a framework for offline RL that captures the entire data composition?
> 2. Is there a single algorithm that is adaptively rate-optimal w.r.t. the unknown data composition?
>
> The reviewer might be referring to lines 86-90 of the paper:
>
> > it is observed empirically that existing model-free methods perform better when the dataset is nearly expert-driven whereas existing model-based methods perform better when the dataset is randomly-collected [51, 5, 52]. It remains unclear whether a single algorithm exists that performs well regardless of data composition.
>
> We clarify that here we are saying that empirical studies suggest that some pessimistic algorithms work better for uniform datasets (and they happen to be model-based) while others work better for near-expert datasets (and they happen to be model-free). This motivates us to ask: is there a single algorithm (model-based or model-free) that is optimal regardless of the data composition, be it uniform, near-expert, or somewhere in between? To answer this question, we study whether a particular algorithm (VI-LCB) is optimal regardless of data composition.
>
> ### Experimental results
>
> Our paper is theoretical with the goal of formulating offline RL problems and providing theoretical foundations and insights for pessimism by studying tight upper bounds and information-theoretic limits under the $C^\star$ formulation. We agree with the reviewer that empirical evaluations can further strengthen the work. We plan to include such experiments in the camera-ready version as well as future work where we study the function approximation setting.
>
> ### References
>
> Ross, Stéphane, and Drew Bagnell. "Efficient reductions for imitation learning." Proceedings of the thirteenth international conference on artificial intelligence and statistics. JMLR Workshop and Conference Proceedings, 2010.
>
> Rajaraman, Nived, et al. "Toward the fundamental limits of imitation learning." arXiv preprint arXiv:2009.05990 (2020).
>
>
> Xie, Tengyang, and Nan Jiang. "Q* approximation schemes for batch reinforcement learning: A theoretical comparison." Conference on Uncertainty in Artificial Intelligence. PMLR, 2020.
>
> Guo, Chenghao, Zhiyi Huang, and Xinzhi Zhang. "Sample complexity of single-parameter revenue maximization." ACM SIGecom Exchanges 17.2 (2020): 62-70.
>
> Jin, Ying, Zhuoran Yang, and Zhaoran Wang. "Is Pessimism Provably Efficient for Offline RL?." International Conference on Machine Learning. PMLR, 2021.
>
> Yin, Ming, Yu Bai, and Yu-Xiang Wang. "Near-optimal offline reinforcement learning via double variance reduction." arXiv preprint arXiv:2102.01748 (2021).
>
> Kumar, Aviral, et al. "Conservative q-learning for offline reinforcement learning." arXiv preprint arXiv:2006.04779 (2020).
>
> Buckman, Jacob, Carles Gelada, and Marc G. Bellemare. "The Importance of Pessimism in Fixed-Dataset Policy Optimization." International Conference on Learning Representations. 2020.
>
> Dai, Bo, et al. "Coindice: Off-policy confidence interval estimation." arXiv preprint arXiv:2010.11652 (2020).
>
> Kidambi, Rahul, et al. "Morel: Model-based offline reinforcement learning." arXiv preprint arXiv:2005.05951 (2020).
>
> Yu, Tianhe, et al. "Mopo: Model-based offline policy optimization." arXiv preprint arXiv:2005.13239 (2020).
>
> ____
> We hope that we have addressed the reviewer’s concerns. We would be happy to answer any questions and discuss further in case the reviewer believes there are any missing details.

---

> > ### Comment · Reviewer_9pjr · 2021-09-01
> > **Respons**
> >
> > Thank you for addressing my concerns; I will update my score accordingly.

---

### Official Review · Reviewer_ivrh · 2021-08-01

**Rating:** 8
**Confidence:** 4

**Summary:**

This is a strong paper that is well-motivated and well-written. I enjoy reading the paper, thinking about the questions and learning the new results.

Contribution on problem formulation: The paper provides a new offline RL framework that accommodates the entire data composition range, smoothly interpolating between two regimes: expert data and data with uniform coverage.

Contribution on algorithm design and the upper bounds: The paper provides a lower-confidence-bound algorithm that achieves almost optimal performance in multi-armed bandits and MDPs, and optimally solves the offline learning problem in contextual bandits. It is interesting to see that in all the three settings, LCB achieves a faster rate of 1/N for nearly-expert datasets compared to the usual rate of 1/\sqrt{N} in offline RL.

Contribution on the lower bounds: The paper provides information-theoretical lower bounds to capture the fundamental hardness for this new offline RL framework. The paper also has interesting discussion on whether the knowledge of C* is necessary to design optimal algorithm.

**Limitations And Societal Impact:**

I like what the authors honestly point out in the checklist question 1(b). For me, a major limitation of the work is that all the results are limited to the tabular setting, which is fairly well-studied in both the realm of online RL and the realm of off-policy evaluation. It is not clear whether the new concept of C* is still fundamental and meaningful when the cardinality of states is very large. But this is still a clear accept given that the paper is already full and addresses the tabular setting throughly.

**Main Review:**

This paper is really well-written. The authors think about the questions throughly and do a very complete study. I particularly like the fact that the proposed LCB algorithm is very simple and intuitive. I think the paper is self-contained and addresses its two proposed questions very well.

The paper make contributions in both the problem formulation and a bunch of new theoretical claims (some of them are insightful). On the other hand, it is not clear whether the existing theoretical claims have significant technical challenges that require very original methodology. If there are, it may be good to add some high-level descriptions in Section 6 and discuss whether the proof techniques can be used to analyze other problems. I appreciate that the author point out the technical hurdle to improve their current results and close the gap.

I did not read the proofs but the results looked solid (based on the writing and the organization of their theoretical claims).

**Time Spent Reviewing:**

4.5

---

> ### Author Response · Authors · 2021-08-08
> **Response**
>
> We thank the reviewer for the time spent reviewing our work, their thorough review, and positive feedback. We discuss the points raised by the reviewer below.
>
> ### Originality in proofs and high-level description
>
> We discuss the main challenges in the proofs and our original proof techniques below. Per reviewer's suggestion, we will add a discussion of these points to the camera-ready version.
>
> Our **most important technical contributions** are:
>
> 1. Going from the loose sub-optimality bound to the tight bound;
>
> 2. Proving the tight $1/N$ rate for the near-expert datasets.
>
> For instance, in contextual bandits the loose rate is $\sqrt{SC^\star/N}$ and the tight minimax optimal rate is $\sqrt{S(C^\star - 1)/N} + S/N$. To show the loose rate, conditioned on penalty capturing the confidence interval and $N(s,\pi^\star(s)) > 0$, one can bound the expected sub-optimality by an expectation over penalty, followed by our technical result on inverse moments of binomial random variables (Lemma 5 in Appendix F), and applying the definition of $C^\star$:
>
> $E_{\mathcal{D, \rho}} [r(s,\pi^\star(s)) - r(s, \hat{\pi}(s))] \lesssim E_{\mathcal{D, \rho}} \left[ \sqrt{\frac{L}{N(s,\pi^\star(s))}} \right] \lesssim E_\rho \left[ \sqrt{\frac{L}{N \mu(s,\pi^\star(s))}} \right] \lesssim \sqrt{\frac{SC^\star L}{N}}$
>
> This approach, however, fails to show the $C^\star - 1$ dependency and thus the fast rate of $1/N$ when $C^\star \approx 1$. To prove the fast rate, we carefully decompose the policy sub-optimality and directly characterize the probability that LCB chooses a wrong action.
>
> Our **sub-optimality decomposition technique** is as follows.
>
> 1. First, we partition the states based on whether at least one sample on the optimal action is included in the dataset. By a missing mass analysis, we show that the contribution of the states with $N(s,\pi^\star(s)) = 0$ to total sub-optimality is bounded by $S/N$.
>
> 2. Second, we decompose the sub-optimality based on whether the penalty captures the LCB of the Q-function.
>
> 3. Third, we partition the states according to whether they significantly affect the final performance, i.e. whether $\rho(s) \gtrsim \frac{C^\star L}{N}$.
>
> 4. In the last step, we partition the states based on whether the optimal action is selected with a dominant probability. We prove that if the optimal action is dominant in a state, then the sub-optimality suffered from that state is small. Interestingly, we show that the complement case has a sub-optimality bounded by $\sqrt{S(C^\star - 1)/N}$.
>
> In Section 6, we provide intuition on why such direct analysis yields the tight rate. The detailed proof is given in Appendix C.1.
>
> ### Analysis takeaway: The hierarchy of sub-optimality surrogates
> Our analysis of LCB in an episodic MDP example (Appendix E.6) reveals challenges involved in proving the adaptive optimality of LCB due to error propagation in MDP. A takeaway from our technical contribution is the following hierarchy of sub-optimality surrogates:
>
> `policy sub-optimality`
>
> $\quad \quad \quad \quad \bigwedge $
>
> `policy and gap analysis:` probability of choosing a wrong action $\times$ loss suffered (gap)
>
> $\quad \quad \quad \quad \bigwedge $
>
> `policy analysis:` probability of choosing a wrong action $\times$ maximum loss
>
> $\quad \quad \quad \quad \bigwedge $
>
> `value-based analysis:` bounding the expectation of penalty
>
> As we move up in this hierarchy, we achieve a tighter bound on sub-optimality but the analysis becomes more complex. In the LCB analysis, the value-based analysis only gives a $1/\sqrt{N}$ rate. However, policy-based analysis can give a tighter bound achieving a rate faster than $1/\sqrt{N}$. Our episodic example suggests that proving our conjecture may require a policy and gap analysis.
>
> **Proof techniques can be used to analyze other problems.** Our decomposition technique (policy analysis) and our policy and gap analysis in Appendix E.6 provide a way to go beyond the value-based analysis and may be useful in obtaining tighter rates, closing the gap between upper and lower bounds in RL problems, and performing instance-dependent analysis.
>
> ### Going beyond tabular
> We expect to see our characterization of offline RL via single-policy concentrability to be extended to the function approximation setting and used in the development of new offline RL algorithms that only require partial coverage. Relevant notions of distribution shift are already used in linear MDPs in online RL algorithms.
> Below, we give examples of possible definitions of $C^\star$ beyond tabular parameterization.
>
> * **Low rank MDPs.** In low rank MDPs with features $\phi(s,a)$, one may define $C^\star$ as follows: $\max_x \frac{x^\top E_{s,a \sim d^\star}[\phi(s,a) \phi(s,a)^\top]x}{x^\top E_{s,a \sim \rho}[\phi(s,a) \phi(s,a)^\top]x} \leq C^\star$.
>
> * **Q-function parameterization.** For a general Q-function class $\mathcal{Q}$, one can define $C^\star$ based on the Bellman error:
> $\sup_{Q \in \mathcal{Q}} \frac{E_{s,a \sim d^\star}[(Q - \mathcal{T}Q)^2(s,a)]}{E_{s,a \sim \rho}[(Q - \mathcal{T}Q)^2(s,a)]} \leq C^\star$.
>
> * **Model-based with function approximation.** For a class of transition dynamics $\mathcal{F}$, one can define $C^\star$ according to:
> $\sup_{P \in \mathcal{F}} \frac{ E_{s,a \sim d^\star} [|| P(. | s,a) - P^\star(. | s,a)||^2_1] }{E_{s,a \sim \rho} [|| P(. | s,a)- P^\star(. | s,a)||_1^2 ]} \leq C^\star.$
> Here, $P^\star$ denotes the true transition dynamics.

---

> > ### Comment · Reviewer_ivrh · 2021-08-22
> > **Review Update**
> >
> > Thank you for the update. The part about going beyond tabular should be added to the paper, with clear explanation on the definition of the dynamics.  It seems that you already have a systematical way to generalize your results, covering the general Q-function parametrization and the general model-based setting. Then, how strong is the assumption that C^* is bounded in those general settings? How are these assumptions related to assumptions (that lead to learnability) in online RL (e.g., Jiang et al 2017, Sun et al. 2019)?
> >
> > I am satisfied with your explanation on your two technical contributions. I recommend you to add a high level description pointing out those two technical contributions in Section 6, and add an independent section in the appendix to further explain your techniques as what you write here. Is it correct that you use the sub-optimality decomposition technique several times for different settings in the paper? If so, having a unifying description will be good for the readers to learn your proof.
> >
> > I appreciate the author's efforts to address my questions and other reviewers' questions. I think the pessimistic penalty idea behind LCB is standard (that may be the reason why the paper is titled ''a tale of Pessimism"), but the paper provides a interesting scale-sensitive analysis to the LCB algorithm. In learning theory, refined understanding of "standard" algorithmic idea is quite important, and obtaining more adaptive guarantee is always a popular theme. I recommend for acceptance and maintain my score.
> >
> >
> >
> > Jiang, Nan, et al. "Contextual decision processes with low Bellman rank are PAC-learnable." International Conference on Machine Learning. PMLR, 2017.
> >
> > Sun, Wen, et al. "Model-based rl in contextual decision processes: Pac bounds and exponential improvements over model-free approaches." Conference on learning theory. PMLR, 2019.

---

> > > ### Author Response · Authors · 2021-09-02
> > > **Response**
> > >
> > > We thank the reviewer for providing further feedback and suggestions for improving the quality and clarity of the paper. We are happy that the reviewer is satisfied with our answers. We agree with the reviewer's suggestions and will include further discussion on going beyond the tabular setting and our decomposition technique to the camera-ready version. We respond to the reviewer's questions below.
> > >
> > > >Then, how strong is the assumption that $C^\star$ is bounded in those general settings? How are these assumptions related to assumptions (that lead to learnability) in online RL (e.g., Jiang et al 2017, Sun et al. 2019)?
> > >
> > > The generalized definitions of $C^\star$ to the function approximation setting are weaker than the tabular definition $\frac{d^\star(s,a)}{\mu(s,a)} \leq C^\star$ and thus also weaker than prior assumptions in offline RL. In offline RL with function approximation, $C^\star$ definition depends on the following three aspects of the problem:
> > > 1. Properties of the environment;
> > > 2. Properties of the function class;
> > > 3. Properties of the data distribution.
> > >
> > > Overall, generalized $C^\star$ definitions are not strong since (1) they do not require coverage over non-competing policies, and (2) they do not impose assumptions related to individual states and actions.
> > >
> > > The learnability assumptions in online RL, which characterize the possibility of efficient exploration and depend on the environment and function class such as in the Bellman rank definition, seem fundamentally different from offline RL. One corner case of offline RL is imitation learning for which realizability is sufficient and no additional learnability assumptions are needed.
> > >
> > > > Is it correct that you use the sub-optimality decomposition technique several times for different settings in the paper? If so, having a unifying description will be good for the readers to learn your proof.
> > >
> > > Yes, the sub-optimality decomposition technique is used in both CB and MDP settings, although some details are different. We thank the reviewer for their suggestion on unifying the description of the proofs. We will include that in the camera-ready version.

---

### Decision · Program_Chairs · 2021-09-27

**Decision:**

Accept (Poster)

**Comment:**

Authors show novel and interesting theoretical results that provide scale-sensitive minimax upper and lower bounds to off-policy learning, which correctly quantify the difficulty of the problem as the behavior policy becomes near-optimal. I agree with the reviewers that the contributions are significant and substantive.